# RODIMUS*: BREAKING THE ACCURACY-EFFICIENCY TRADE-OFF WITH EFFICIENT ATTENTIONS

**Zhihao He**[1,2]*, **Hang Yu**[2]*, **Zi Gong**[2], **Shizhan Liu**[2], **Jianguo Li**[2]†, **Weiyao Lin**[1]†
[1]Shanghai Jiao Tong University, [2]Ant Group

## ABSTRACT

Recent advancements in Transformer-based large language models (LLMs) have set new standards in natural language processing. However, the classical softmax attention incurs significant computational costs, leading to a $\mathcal{O}(T)$ complexity for per-token generation, where $T$ represents the context length. This work explores reducing LLMs' complexity while maintaining performance by introducing Rodimus and its enhanced version, Rodimus+. Rodimus employs an innovative data-dependent tempered selection (DDTS) mechanism within a linear attention-based, purely recurrent framework, achieving significant accuracy while drastically reducing the memory usage typically associated with recurrent models. This method exemplifies semantic compression by maintaining essential input information with fixed-size hidden states. Building on this, Rodimus+ combines Rodimus with the innovative Sliding Window Shared-Key Attention (SW-SKA) in a hybrid approach, effectively leveraging the complementary semantic, token, and head compression techniques. Our experiments demonstrate that Rodimus+-1.6B, trained on 1 trillion tokens, achieves superior downstream performance against models trained on more tokens, including Qwen2-1.5B and RWKV6-1.6B, underscoring its potential to redefine the accuracy-efficiency balance in LLMs. Model code and pre-trained checkpoints are open-sourced at `https://github.com/codefuse-ai/rodimus`.

## 1 INTRODUCTION

Recent advancements have positioned Transformer-based large language models (LLMs) at the forefront of natural language processing, establishing them as state-of-the-art. Their strong capabilities stem from the softmax attention mechanism, which selectively focuses on relevant tokens stored in the key-value (KV) cache when predicting the next token. By maintaining a historical record of tokens, the KV cache allows all pertinent data to remain accessible during inference. However, the demand to store this extensive historical information incurs notable computational costs, leading to $\mathcal{O}(T)$ complexity for per-token generation, where $T$ represents the length of the context preceding the generated token. Indeed, a 7B Llama (Touvron et al., 2023)), without inference optimization, takes over a minute to generate 2K-length sequences on an A10-24G (Ilyas Moutawwakil, 2023).

This has sparked research into next-generation foundation models aimed at efficient alternatives to attention. Among them, three main categories stand out that aim to compress the KV cache in the original softmax attention from distinct perspectives: semantic, token, and head compression.

**Semantic Compression**: The first category, also known as linear attention (Katharopoulos et al., 2020), or linear state space models (SSMs) (Gu & Dao, 2023; Dao & Gu, 2024), substitutes the exponential kernel in softmax attention with a simplified inner product of (transformed) query and key vectors. This approach results in a recurrently updated hidden state of fixed size that retains historical semantic information like linear RNNs (Qin et al., 2024b; Peng et al., 2023), successfully reducing the per-token generation complexity from $\mathcal{O}(T)$ to $\mathcal{O}(1)$. However, compressing the softmax attention—characterized by its unlimited capacity—into linear attention with fixed capacity inevitably leads to some information loss. To address this, current methods strive to (i) increase the recurrent state size to enhance memory capacity (Qin et al., 2024a) and (ii) utilize the fixed-sized states more effectively (Yang et al., 2024b). Despite these advancements, there remains a trade-off between complexity and performance (refer to Figure 1). Blindly expanding the hidden states compromises

---

*Equal contribution. This work was done when Zhihao He was a research intern at Ant Group.
†Corresponding authors.

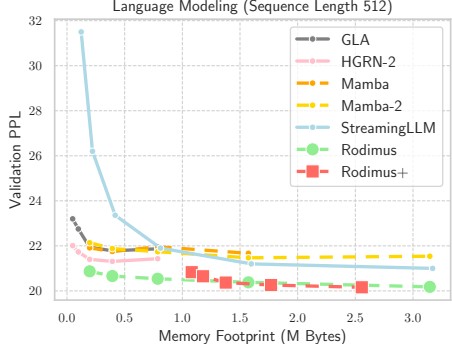 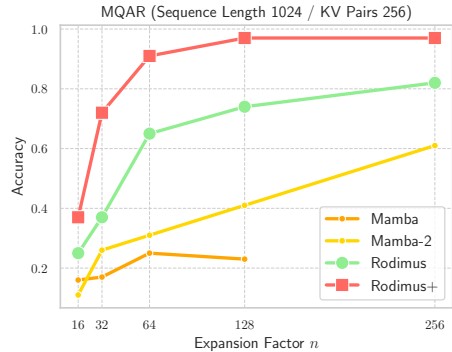

(a) LM Task on the WikiText-103 Dataset.      (b) Multi-Query Associative Recall (MQAR) Task.

Figure 1: Memory Footprint vs. Performance: (a) This experiment is conducted on the WikiText-103 dataset (Details in Appendix D.3), and focuses on the recorded best perplexity (PPL). The memory footprint is adjusted by modifying the expansion factor. (b) The model's recall capability is assessed using the MQAR Task (See Appendix D.7), as described in Arora et al. (2024a). Among all models evaluated, Rodimus* achieves the optimal balance between space complexity and performance.

time and space efficiency, while incorporating data-dependent decay or gating mechanisms to filter out irrelevant past information may hinder parallel training efficiency, as noted in (Yang et al., 2024c).

**Token Compression**: This second category introduces sparsity into the attention mask, allowing it to follow predefined patterns and focus on strategically chosen tokens, such as those at the beginning of a sequence or close to the answer (Xiao et al., 2024; Han et al., 2024). Consequently, this method can also achieve the $\mathcal{O}(1)$ complexity for per-token generation. However, the elimination of masked tokens can lead to a complete loss of information and potentially degrade the overall performance.

**Head Compression**: The third category reshapes attention by modifying the design of attention heads. This involves grouping heads and sharing keys and values within each group, as seen in multi-query attention (MQA) (Shazeer, 2019) and grouped-query attention (GQA) (Ainslie et al., 2023). While these methods reduce the cache size by a constant factor, they are inherently lossy compared to multi-head attention (MHA) (Vaswani, 2017) as the same values are used within each group.

Ultimately, none of these methods can fully supplant the original softmax attention, prompting an essential question: *Is it feasible to reduce the complexity of LLMs while preserving performance?* Our research affirms this possibility, showing that a linear attention model enhanced with a well-designed data-dependent tempered selection (DDTS) mechanism—an improvement within the first category—and its hybridization with our proposed sliding window shared-key attention (SW-SKA), which integrates all three types of attention compressions, offers viable solutions. We designate the former model as Rodimus and the latter as Rodimus+. Remarkably, **Rodimus** maintains a much smaller hidden state size than most current SOTA recurrent models (e.g., **half** of the size in Mamba2 (Dao & Gu, 2024)), but outperforms softmax attention-based Transformers. Moreover, **Rodimus**+-1.6B (trained on 1 trillion tokens) achieves an average performance that is 0.31% higher than Qwen2-1.5B (trained on 7 trillion tokens) and 2.3% higher than RWKV6-1.6B (trained on 1.4 trillion tokens) in downstream task evaluations. Our contributions can be summarized as follows:

- We introduce a linear attention-based, purely recurrent model, Rodimus, effectively overcoming the accuracy-efficiency trade-off found in existing recurrent models. By incorporating the innovative DDTS, Rodimus can autonomously filter out irrelevant information during semantic compression, resulting in improved validation perplexity (PPL) with much smaller memory footprints compared to other SOTA methods, as illustrated by the green curves in Figure 1.

- We present a hybrid model, Rodimus+, which combines Rodimus with the innovative SW-SKA. While the recurrent hidden state in Rodimus offers a comprehensive semantic view of the historical context, the sliding window attention highlights the nearby tokens that are most influential in predicting the next token, and SKA further compresses cache size in a lossless manner. As shown by the red curves in Figure 1, Rodimus+ continues to extend the accuracy-efficiency frontier of existing methods, achieving superior performance with reduced complexity.

- We validate the effectiveness of Rodimus* (encompassing both Rodimus and Rodimus+) through thorough experimentation. In downstream evaluations with model sizes from 130M to 1.3B, Rodimus* demonstrates enhanced language modeling performance relative to current SOTA models of similar sizes. Furthermore, Rodimus* achieves a performance improvement of up to 7.21% over Mamba2, and outperforms even softmax attention-based Pythia on NeedleBench—a suite of recall-intensive tasks where recurrent models typically underperform (Waleffe et al., 2024).

## 2 PRELIMINARIES

In this section, we provide a brief introduction to softmax attention in Vaswani (2017) and derive its recurrent form. We will then derive the aforementioned three attention compression methods, including linear attention for semantic compression, sparse attention for token compression, and sharing-based attention for head compression. The first two methods simplify attention by reducing context length, while the last one makes modifications along the dimension of attention heads.

**Softmax Attenion and its Recurrent Form**: Suppose that $\boldsymbol{X} = \{\boldsymbol{x}_1, \ldots, \boldsymbol{x}_T\} \in \mathbb{R}^{T \times d}$ denotes the input sequence with length $T$ and dimension $d$. Standard autoregressive Transformers (Vaswani, 2017) utilize a softmax attention mechanism to generate the output $\boldsymbol{O} \in \mathbb{R}^{T \times d}$, that is,

$$\boldsymbol{Q}, \boldsymbol{K}, \boldsymbol{V} = \boldsymbol{X}\boldsymbol{W}_Q, \boldsymbol{X}\boldsymbol{W}_K, \boldsymbol{X}\boldsymbol{W}_V, \quad \boldsymbol{O} = \mathrm{softmax}\left((\boldsymbol{Q}\boldsymbol{K}^\top) \odot \boldsymbol{M}\right)\boldsymbol{V}, \tag{1}$$

where $\boldsymbol{W}_Q \in \mathbb{R}^{d \times n}, \boldsymbol{W}_K \in \mathbb{R}^{d \times n}, \boldsymbol{W}_V \in \mathbb{R}^{d \times m}$ are learnable weight matrices associated with the queries $\boldsymbol{Q}$, the keys $\boldsymbol{K}$, and values $\boldsymbol{V}$, respectively. The matrix $\boldsymbol{M} \in \{-\infty, 1\} \in \mathbb{R}^{T \times T}$ denotes the upper-triangular attention mask, ensuring that the model does not attend to future tokens. Multi-Head Attention (MHA) further splits the dimensionality $d$ into $h$ heads, computing attention for each head in $\mathbb{R}^{d_h}$ individually with distinct weights. This approach enhances the model's capacity to capture diverse sequential information while reducing the computational costs for matrix operations.

The above parallel form allows for the computation of $\boldsymbol{O} = \{\boldsymbol{o}_1, \ldots, \boldsymbol{o}_T\}$ in parallel given the full input $\boldsymbol{X}$, facilitating efficient training. In contrast, during inference, Transformers rely on the following recurrent formulation (Katharopoulos et al., 2020):

$$\boldsymbol{q}_t, \boldsymbol{k}_t, \boldsymbol{v}_t = \boldsymbol{x}_t\boldsymbol{W}_Q, \boldsymbol{x}_t\boldsymbol{W}_K, \boldsymbol{x}_t\boldsymbol{W}_V, \quad \boldsymbol{o}_t = \frac{\sum_{i=1}^t \exp(\boldsymbol{q}_t\boldsymbol{k}_i^\top)\boldsymbol{v}_i}{\sum_{i=1}^t \exp(\boldsymbol{q}_t\boldsymbol{k}_i^\top)}. \tag{2}$$

Here, the query $\boldsymbol{q}_t \in \mathbb{R}^{1 \times n}$, key $\boldsymbol{k}_t \in \mathbb{R}^{1 \times n}$, and value $\boldsymbol{v}_t \in \mathbb{R}^{1 \times m}$ vectors are computed based on the representation of current token $\boldsymbol{x}_t \in \mathbb{R}^{1 \times d}$. Attention is subsequently performed over the evolving collection of keys $\{\boldsymbol{k}_1, \ldots, \boldsymbol{k}_t\}$ and values $\{\boldsymbol{v}_1, \ldots, \boldsymbol{v}_t\}$ (i.e., the KV cache). Thus, the time and space complexity for generating the next token at time stamp $t$ is $\mathcal{O}(t)$.

**Linear Attention for Semantic Compression**: To optimize efficiency, linear attention mechanisms replace the exponential kernel $\exp(\boldsymbol{q}_t\boldsymbol{k}_i^\top)$ in Eq. (2) by a kernel $k(\boldsymbol{q}_t, \boldsymbol{k}_i)$ paired with an associated feature map $\phi$, i.e., $k(\boldsymbol{q}_t, \boldsymbol{k}_i) = \phi(\boldsymbol{q}_t)\phi(\boldsymbol{k}_i)^\top$. As a result, the calculation of $\boldsymbol{o}_t$ can be simplified as:

$$\boldsymbol{o}_t = \frac{\sum_{i=1}^t \phi(\boldsymbol{q}_t)\phi(\boldsymbol{k}_i)^\top \boldsymbol{v}_i}{\sum_{i=1}^t \phi(\boldsymbol{q}_t)\phi(\boldsymbol{k}_i)^\top} = \frac{\phi(\boldsymbol{q}_t)\sum_{i=1}^t \phi(\boldsymbol{k}_i)^\top \boldsymbol{v}_i}{\phi(\boldsymbol{q}_t)\sum_{i=1}^t \phi(\boldsymbol{k}_i)^\top}. \tag{3}$$

Letting $\boldsymbol{S}_t = \sum_{i=1}^t \phi(\boldsymbol{k}_i)^\top \boldsymbol{v}_i \in \mathbb{R}^{n \times m}$ and $\boldsymbol{z}_t = \sum_{i=1}^t \phi(\boldsymbol{k}_i)^\top \in \mathbb{R}^{n \times 1}$ be the KV state and the K state respectively, we can rewrite the previous equation as a linear state-space model (SSM) or RNN:

$$\boldsymbol{S}_t = \boldsymbol{S}_{t-1} + \phi(\boldsymbol{k}_t)^\top \boldsymbol{v}_t, \quad \boldsymbol{z}_t = \boldsymbol{z}_{t-1} + \phi(\boldsymbol{k}_t)^\top, \quad \boldsymbol{o}_t = \frac{\phi(\boldsymbol{q}_t)\boldsymbol{S}_t}{\phi(\boldsymbol{q}_t)\boldsymbol{z}_t}. \tag{4}$$

We emphasize that $\boldsymbol{S}_t$ provides a semantic compression of the historical context up to $t$. The denominator $\phi(\boldsymbol{q}_t)\boldsymbol{z}_t \in \mathbb{R}$ may introduce numerical instabilities and hinder the optimization (Qin et al., 2022a), prompting many recent studies to replace it with a normalization (Qin et al., 2022a; Sun et al., 2023a). Moreover, it is common to use the identity mapping for $\phi$ (Sun et al., 2023a; Yang et al., 2024b), and so Eq. (4) amounts to

$$\boldsymbol{S}_t = \boldsymbol{S}_{t-1} + \boldsymbol{k}_t^\top \boldsymbol{v}_t, \quad \boldsymbol{o}_t = \boldsymbol{q}_t\boldsymbol{S}_t. \tag{5}$$

More generally, when framed within the context of linear **SSMs** (Gu & Dao, 2023; Dao & Gu, 2024) or **RNNs** (De et al., 2024), these equations can be reformulated as:

$$\boldsymbol{S}_t = \boldsymbol{A}_t \odot \boldsymbol{S}_{t-1} + \boldsymbol{B}_t \odot \boldsymbol{u}_t, \quad \boldsymbol{o}_t = \boldsymbol{C}_t\boldsymbol{S}_t, \tag{6}$$

where $\boldsymbol{A}_t = 1$, $\boldsymbol{B}_t = 1$, and $\boldsymbol{C}_t = \boldsymbol{q}_t$ represent the state transition, input, and output matrix respectively, while $\boldsymbol{u}_t = \boldsymbol{k}_t^\top \boldsymbol{v}_t$ signifies the input or control. In stark contrast to softmax attention, the per-token generation complexity at time stamp $t$ is $\mathcal{O}(1)$, enabling efficient inference. Meanwhile, linear attention can be expressed in a parallel form similar to Eq. (1), allowing for training parallelism.

However, this additive formulation of updating the hidden states $\boldsymbol{S}_t$ with new key-value pairs at each time step $t$ does not possess the capability to forget irrelevant information, which leads to the phenomenon known as attention dilution (Qin et al., 2022b;a). To address this concern, recent works (i) increase the state size (either $n$ or $d_h$) to retain more information and (ii) incorporate decay

factors or gating mechanisms that enable the state $S_t$ to discard irrelevant past information. The first approach typically sacrifices efficiency for performance, whereas the second method focuses on designing $A_t$ and $B_t$ to manage memory retention and forgetting behaviors within the hidden states.

We summarize the functional forms of $A_t$ and $B_t$ used in the SOTA methods in Table 4. Three major trends can be gleaned from this table. Firstly, it is beneficial for $A_t$ and $B_t$ to be negatively correlated, as shown in Mamba (Gu & Dao, 2023), Mamba2 (Dao & Gu, 2024), and HGRN2 (Qin et al., 2024a), allowing them to collaboratively regulate the deletion and addition of information in the hidden state $S_t$. This contrasts with methods like RetNet (Sun et al., 2023a), gRetNet (Sun et al., 2024), and GLA (Yang et al., 2024b), which use $A_t$ as a decay factor while keeping $B_t$ constant at 1. Secondly, allowing $A_t$ and $B_t$ to be functions of the input $u_t$ enables dynamic adjustments over time in a data-dependent manner. This approach has been validated by the superior performance of GLA (Yang et al., 2024b), gRetNet (Sun et al., 2024) compared to RetNet(Sun et al., 2023a). Lastly, designs for gating mechanisms must be compatible with GPU acceleration, ensuring that the recurrent expression (6) aligns with a parallel format, similarly to (1).

In Section 3.1, we further analyze these functional forms in relation to their equivalence with linear attention and propose a novel data-dependent tempered selection (DDTS) mechanism. This mechanism is capable of succinctly compressing historical information within a recurrent hidden state of fixed capacity, all while facilitating parallel training.

**Sparse Attention for Token Compression**: This group of methods aim to sparsify the attention mask $M$ in Eq. (1). The goal is to compute only the $(q_t, k_i)$ pairs associated with nonzero elements of $M_{ti}$. Various types of attention masks have been proposed in the literature, including window attention (Child et al., 2019), dilated attention (Beltagy et al., 2020), and bridge attention (Guo et al., 2023), among others. Research shows that the softmax attention maps in vanilla Transformer models often exhibit a localized behavior (Qin et al., 2022a;b; Xiao et al., 2024). Thus, we exploit a sliding window (SW) attention to enhance the local context understanding in Rodimus+.

**Sharing-based Attention for Head Compression**: In the original MHA, each head has distinct linear transformations $W_Q, W_K, W_V$ for the input $X$, allowing for diverse representations and attention maps across heads. Sharing-based attention seek to compress the MHA by allowing key and value heads to be shared among multi-query heads. Specifically, in MQA (Shazeer, 2019), all heads use a single set of key and value weights, which reduces parameters and memory usage but risks diminishing the attention mechanism's expressiveness. As a better alternative, GQA assigns one key and value head for each group of query heads (Ainslie et al., 2023). However, it still limits the expressiveness of the original MHA by overly constraining learned relationships. As a remedy, we propose the Shared-Key Attention (SKA) in Section 3.2.1, which compresses the MHA while preserving its expressiveness.

## 3 METHODOLOGY

In this section, we introduce two models: Rodimus and Rodimus+, whose architecture is depicted in Figure 2. Rodimus is a purely recurrent model that iteratively compresses historical context into a fixed-size hidden state (i.e., semantic compression) and further exploits this state for the next token prediction. With the equipment of the newly proposed DDTS mechanism, Rodimus effectively filters out irrelevant information, thereby enhancing performance while reducing the size of the hidden state. On the other hand, Rodimus+ builds upon Rodimus by integrating the proposed SW-SKA technique. This enhancement improves performance without sacrificing efficiency, allowing for a seamless combination of semantic, token, and head compression methods.

### 3.1 RODIMUS: OVERCOMING PERFORMANCE BOTTLENECK IN SEMANTIC COMPRESSION

#### 3.1.1 ANALYSIS OF EXISTING LINEAR ATTENTION MODELS

As can be observed from Table 4, the state transition equation in all existing linear attention models can be expressed recurrently as:

$$S_t = (\alpha_t^\top \beta_t) \odot S_{t-1} + (\hat{\alpha}_t^\top \hat{\beta}_t) \odot (k_t^\top v_t) = (\alpha_t^\top \beta_t) \odot S_{t-1} + (\hat{\alpha}_t \odot k_t)^\top (\hat{\beta}_t \odot v_t), \quad (7)$$

where $\alpha_t \in \mathbb{R}^{1 \times n}$ and $\beta_t \in \mathbb{R}^{1 \times m}$ denotes the gating mechanism along the dimension of $n$ and $m$ respectively, and $\hat{\alpha}_t$ and $\hat{\beta}_t$ are negatively correlated with $\alpha_t$ and $\beta_t$, allowing them to select between the current input $u_t = k_t^\top v_t$ and the previous state $S_{t-1}$. Substituting Eq. (7) into Eq. (6)

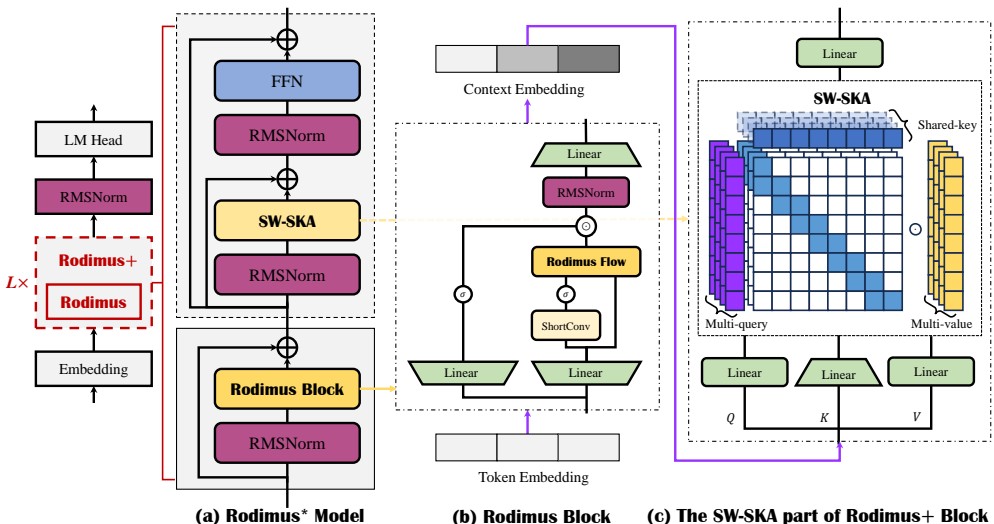

Figure 2: Overview of the Proposed Models. The **Rodimus\* Model** serves as a template for both Rodimus and Rodimus+. When modules within the gray dashed box are included, it becomes the Rodimus+ Model; otherwise, it is the Rodimus Model. The architecture comprises $L$ layers of stacked **Rodimus\* Blocks** along with essential modules for language modeling (e.g., Embedding, RMSNorm, LM Head). Rodimus Flow (11) denotes our proposed recurrent computation method. The purple arrows depict information flow between layers (ignoring RMSNorm). In the Rodimus+ Model, the Rodimus block compresses the Token Embedding into a global semantic embedding, enhancing SW-SKA's ability to perceive long contexts.

and unfolding the recurrent relation yields:

$$o_t = \sum_{i=1}^{t} \left[ q_t \big( \prod_{j=i+1}^{t} \alpha_j \odot \hat{\alpha}_i \odot k_i \big)^\top \right] \big( \prod_{j=i+1}^{t} \beta_j \odot \hat{\beta}_i \odot v_i \big). \tag{8}$$

Here, $\prod_{j=i+1}^{t} \alpha_j$ captures the relative positional information between time stamp $t$ and $i$, specifically between the query $q_t$ and the key $k_i$. Indeed, this component acts as a more flexible version of relative positional embeddings for softmax attention (e.g., xPos (Sun et al., 2023b) and ALiBi (Press et al., 2022)). In addition, the elements within the vector $\alpha_j$ vary from one another, enabling the capture of high- and low-frequency information akin to sinusoidal positional encoding (Vaswani, 2017) and RoPE (Su et al., 2021). Notably, $\prod_{j=i+1}^{t} \alpha_j$ also incorporates absolute positional information since it is also a function of the absolute positions $j = i + 1, \ldots, t$. Consequently, linear attention models expressed in the form of Eq. (7) capture first-order dependencies within sequences in a position-aware manner. In contrast, the original softmax attention (1) captures only second-order dependencies in a position-agnostic framework, necessitating the addition of positional embeddings (Ren et al., 2023).

On the other hand, $\prod_{j=i+1}^{t} \beta_j$ also regulates the relative positional information between the time stamp $t$ and $i$, but between the query $q_t$ and the value $v_i$. This aspect has not been witnessed in softmax attention. Moreover, earlier studies (Yang et al., 2024b) and our ablation study (see Table 16) indicate that incorporating $\prod_{j=i+1}^{t} \beta_j$ cannot result in improvements compared to setting $\beta_j = 1_m$, probably because such positional information has already been described by $\prod_{j=i+1}^{t} \alpha_j$. Moreover, including $\beta_j$ can impede training efficiency, as the dimensionality of $\beta_t \in \mathbb{R}^{1 \times m}$ is significantly larger than that of $\alpha_t \in \mathbb{R}^{1 \times n}$. This discrepancy can lead to redundant I/O operations when $\beta_j$ are frequently transferred between high bandwidth memory (HBM) and on-chip SRAM during chunkwise parallelism (cf. Appendix B.3).

As a consequence, our focus is on optimizing the design of $\alpha_t$ while setting $\beta_t = 1_m$ for all $t$. We retain $\hat{\beta}_t$ to allow for flexible selection among the various elements in the value vector $v_t$.

### 3.1.2 DESIGN OF $\alpha_t$, $\hat{\alpha}_t$ AND $\hat{\beta}_t$

We propose the following formulations for $\alpha_t$ and $\hat{\alpha}_t$:

$$\alpha_t = \exp(-g_t \odot \tau_t), \quad \hat{\alpha}_t = g_t^{\tau_t}, \tag{9}$$

where $g_t = \zeta(x_t W_g + b_g)$ and $\tau_t = \sigma(x_t W_\tau + b_\tau)$. In this design, $g_t$ serves as a selection gate, determining whether to retain the previous state $S_{t-1}$ or incorporate the current input $k_t$. Note that $\exp(-g_t) = 1 - \sigma(x_t W_g + b_g)$, thus ensuring that $\alpha_t \in [0, 1]$ and preventing complete oblivion of the previous state. In contrast, $\hat{\alpha}_t$ can assume values greater than 1 due to the softplus function. This asymmetry between discarding the previous state and retaining the current introduces greater flexibility into the state transition equation (Gu & Dao, 2023).

Furthermore, different from all existing works, we introduce a temperature gate $\tau_t$ that governs the sharpness or sensitivity of the selection gate $g_t$. As visualized in Figure 7 in the appendix, as $\tau_t$ decreases, both $\alpha_t$ and $\hat{\alpha}_t$ changes more slowly with $g_t$. Note that $\tau_t$ is a function of the input $x_t$, as opposed to being a fixed constant as is the case in gRetNet (Sun et al., 2024) and GLA (Yang et al., 2024b). This provides an additional degree of freedom in the selection process. Indeed, recent findings indicate that tempered losses can improve robustness against noise during training (Papernot et al., 2021; Wang et al., 2023). In our case, we find that $\tau_t$ helps sharpen the original selection gate $g_t$ (see Figure 8 in the appendix), thereby facilitating more aggressive filtering of irrelevant information.

On the other hand, $\hat{\beta}_t$ can be expressed as:

$$\hat{\beta}_t = \sigma(x_t W_{\hat{\beta}}^1 W_{\hat{\beta}}^2 + b_{\hat{\beta}}), \tag{10}$$

where $W_{\hat{\beta}}^1 \in \mathbb{R}^{m \times \ell}$ and $W_{\hat{\beta}}^2 \in \mathbb{R}^{\ell \times m}$ are two low-rank matrices (i.e., $\ell < m$). This low-rank formulation helps mitigate noise in the input while keeping the overall increase in model parameters manageable, thus providing parameter efficiency (Hu et al., 2022). This comprehensive design is referred to as the data-dependent tempered selection (DDTS) mechanism and is proven to be a selection mechanism as demonstrated below:

**Proposition 1.** *Given the specifications for $A_t$ and $B_t$ in Eqs. (6), (7), (9), and (10), DDTS can realize the selection between the previous state $S_{t-1}$ and the current input $u_t$.*

*Proof.* See Appendix C.2. □

### 3.1.3 THE OVERALL RODIMUS BLOCK

The SSM formulation for the overall Rodimus block can be articulated as follows:

$$S_t = \left( \exp(-g_t \odot \tau_t)^\top \mathbf{1}_m \right) \odot S_{t-1} + \left( (g_t^{\tau_t})^\top \hat{\beta}_t \right) \odot (k_t^\top v_t), \quad o_t = q_t S_t + d_t \odot x_t'. \tag{11}$$

In this formulation, we define $q_t = x_t W_q$, $k_t = x_t W_k$, $v_t = x_t$. We follow the Mamba series (Dao & Gu, 2024) to set $v_t = x_t$, and the computation of the control $u_t = k_t^\top v_t$ can be interpreted as the state expansion operation within SSMs. Additionally, we employ ShortConv to process the original input $x_t$ into $x_t'$, from which we derive $g_t$ and $\tau_t$. Note that ShortConv is widely used in recent recurrent models (Gu & Dao, 2023; Dao & Gu, 2024; Yang et al., 2024b); it enhances local context aggregation and introduces nonlinearity for $g_t$ and $\tau_t$. The learnable weight $d_t$ functions as the feedthrough matrix in SSMs. To ensure stability during back-propagation, we implement post-normalization after activation. This entails normalizing $k_t$ and dividing $q_t$ by $\sqrt{n}$ (Dao & Gu, 2024).

The final configuration of the Rodimus block is depicted in Figure 2b. The aforementioned SSM merges into a Gated Linear Unit (GLU) (Dauphin et al., 2017), allowing simultaneous token and channel mixing compactly, akin to the GAU (Hua et al., 2022) and Mamba series (Gu & Dao, 2023; Dao & Gu, 2024). The initial linear layers within the GLU increase the size of the hidden states, thereby enhancing the performance of Rodimus. The last linear layer finally reduces the hidden dimension back to its original size $d$. In contrast to the original Transformer block, which mixes tokens and channels separately through softmax attention and Feed-Forward Network (FFN), this compact architecture demonstrates greater parameter efficiency. Additionally, the output gate of the GLU enriches the gating mechanisms of the Rodimus block.

During inference, the Rodimus block retains only a fixed-size hidden state, resulting in $\mathcal{O}(1)$ time and space complexity. In contrast, during training, the chunkwise parallelization of the Rodimus block ensures sub-quadratic computational complexity, which is explained in the Appendix B.3.

### 3.2 RODIMUS+: INTEGRATION WITH TOKEN COMPRESSION AND HEAD COMPRESSION

Although the hidden states $S_t$ in Rodimus provide a comprehensive semantic overview of historical context, their fixed capacity may still limit performance. On the other hand, it has been observed in the literature (Qin et al., 2022a;b; Xiao et al., 2024) that the attention map in vanilla Transformer

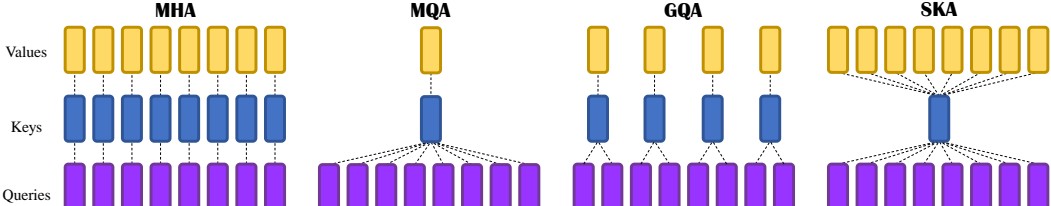

Figure 3: Comparison of Different Attention Mechanisms. In MQA and GQA, values are shared among the same group of query heads, resulting in lossy compression compared to the individual head-specific values used in MHA. SKA maintains the multi-value setting of MHA but uses a shared key across all heads. This approach produces a separate attention map for each head, preserving the expressiveness of MHA while reducing the memory footprint.

tends to be localized, indicating that nearby tokens are significantly more influential in generating responses. To leverage the strengths of both approaches, we introduce Rodimus+, which integrates the Rodimus block with sliding window attention (i.e., token compression). This integration enables the hidden states to act as a robust global representation of the context, incorporating information even from distant tokens relative to the answer, while the sliding window attention sharpens focus on the most influential nearby tokens. Notably, this enhancement occurs without increasing computational complexity. Furthermore, we optimize the KV cache of the sliding window attention by a constant factor by substituting the traditional MHA with our proposed SKA (i.e., head compression).

### 3.2.1 SHARED-KEY ATTENTION FOR LOSSLESS HEAD COMPRESSION

Recall from Section 2 that the MHA for head $h$ can be expressed as:

$$\boldsymbol{O}^h = \mathrm{softmax}\Big(\big(\boldsymbol{Q}^h {\boldsymbol{K}^h}^\top\big) \odot \boldsymbol{M}\Big)\boldsymbol{V}^h = \mathrm{softmax}\Big(\big(\boldsymbol{X}\boldsymbol{W}_Q^h {\boldsymbol{W}_K^h}^\top \boldsymbol{X}^\top\big) \odot \boldsymbol{M}\Big)\boldsymbol{X}\boldsymbol{W}_V^h$$

$$= \mathrm{softmax}\Big(\big((\boldsymbol{X}\boldsymbol{W}_Q^h {\boldsymbol{W}_K^h}^\top \tilde{\boldsymbol{W}}_K^{-\top})(\tilde{\boldsymbol{W}}_K^\top \boldsymbol{X}^\top)\big) \odot \boldsymbol{M}\Big)\boldsymbol{X}\boldsymbol{W}_V^h, \tag{12}$$

where $\tilde{\boldsymbol{W}}_K$ is a learnable weight matrix that is invariant with regard to head $h$, and $\tilde{\boldsymbol{W}}_K^{-\top}$ denotes the pseudo inverse of the transposed $\tilde{\boldsymbol{W}}_K$. Thus, we can redefine the query as $\tilde{\boldsymbol{Q}}^h = \boldsymbol{X}\boldsymbol{W}_Q^h {\boldsymbol{W}_K^h}^\top \tilde{\boldsymbol{W}}_K^{-\top}$ and the key as $\tilde{\boldsymbol{K}} = \boldsymbol{X}\tilde{\boldsymbol{W}}_K$, which is invariant across heads. This means we can employ a single key for all heads without diminishing the expressiveness of MHA. We refer to this mechanism as Shared-Key Attention (SKA), which achieves lossless compression of the KV cache in MHA by utilizing a single key for all heads. In contrast, approaches like MQA (Shazeer, 2019) and GQA (Ainslie et al., 2023) also share the values $\boldsymbol{V}$ among the same group of query heads, which typically differ for each head in the original MHA formulation. Consequently, MQA and GQA can only be seen as lossy compressions of MHA. Figure 3 illustrates the distinctions between MHA, MQA, GQA, and SKA.

We further combine SKA with sliding window attention to create SW-SKA, which introduces a local structure where each token focuses solely on its immediate contextual window, thereby maintaining a constant memory footprint during both training and inference.

### 3.2.2 THE OVERALL RODIMUS+ BLOCK

Recall that the Rodimus block builds upon the foundations established by the Mamba series (Gu & Dao, 2023; Dao & Gu, 2024) and GAU (Hua et al., 2022), enabling the simultaneous mixing of tokens and channels while offering a unified residual connection for both mixers (see Figure 2b). Specifically, it integrates the SSM block (token mixer) into the FFN layers (channel mixer). This structured arrangement allows for an increase in state size without necessitating additional parameters, as the FFN typically amplifies the dimension before subsequently reducing it. In contrast, a conventional Transformer block alternates between self-attention (token mixer) and FFN (channel mixer), with each component linked by its own residual connection. Notably, self-attention is generally not placed in between the FFN layers due to the linear growth of the KV cache with input sequence length, which can lead to significant memory consumption when key and value dimensions are increased.

When integrating the Rodimus block with the SW-SKA, it is beneficial to have the attention mechanism and FFN closely interdependent, as these components collaboratively deliver a localized understanding of context. However, they should maintain relative independence from the Rodimus block, which is responsible for providing a global perspective. To achieve this balance, we employ a two-hop residual approach, adapted from Ma et al. (2024), to effectively bind the attention and FFN

Table 1: Results on WikiText-103.

| Models | Test PPL | Params(M) |
|---|---|---|
| Rodimus | 21.90 | 46.68 |
| Rodimus+ | **21.56** | 48.45 |
| Mamba | 22.58 | 46.08 |
| Mamba2 | 22.78 | 46.38 |
| Transformer++ | 22.02 | 46.24 |
| Llama3 | 22.12 | 46.48 |
| GLA | 22.16 | 44.71 |
| HGRN2 | 22.00 | 46.20 |

- The lower the PPL, the better the language modeling effect. **Bold** indicates the best result, and underline indicates the second-best result.

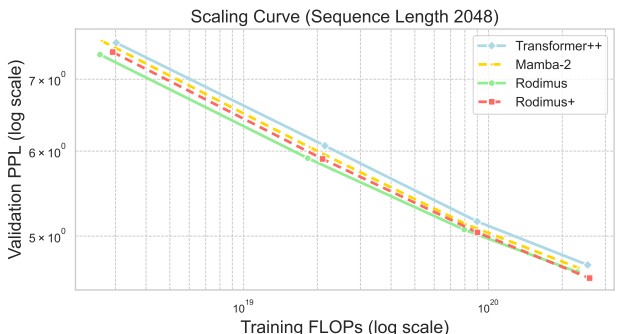

Figure 4: Scaling Curve based on Scaling Laws.

layers. This ensures that the information derived from the token mixer is cohesively integrated into the channel mixer, preserving the overall integrity of the system. The resulting Rodimus+ block is:

$$\boldsymbol{X}_{\text{state}} = \text{Rodimus}(\text{Norm}(\boldsymbol{X})) + \boldsymbol{X},$$

$$\hat{\boldsymbol{Y}} = \text{SW–SKA}(\text{Norm}(\boldsymbol{X}_{\text{state}})) + \boldsymbol{X}_{\text{state}}, \tag{13}$$

$$\boldsymbol{Y} = \text{FFN}(\text{Norm}(\hat{\boldsymbol{Y}})) + \boldsymbol{X}_{\text{state}},$$

where FFN represents the GLU, Norm denotes RMSNorm (Zhang & Sennrich, 2019), and Rodimus signifies the Rodimus block, generating state information for the subsequent SW-SKA. Taken together, the Rodimus block, SW-SKA, and FFN constitute the final Rodimus+ block. In Appendix B.4, we further discuss the relationship between Rodimus+ and existing hybrid models.

## 4 EXPERIMENTS

In this section, we compare Rodimus* with other SOTA methods across various benchmarks, including language modeling and recall benchmarks. A brief introduction to the SOTA methods is presented in Appendix B.5. We conclude this section with a series of ablation studies.

### 4.1 LANGUAGE MODELING

**WikiText-103**: The WikiText-103 language modeling dataset (Merity et al., 2022) consists of over 100 million tokens derived from verified good and featured articles on Wikipedia. In this study, we compare Rodimus* to six other models. Among these, Mamba (Gu & Dao, 2023) and Mamba2 (Dao & Gu, 2024) utilize purely recurrent architectures, similar to Rodimus. GLA (Yang et al., 2024b) and HGRN2 (Qin et al., 2024a) serve as linear attention counterparts to the Mamba series. We also include Transformer++ (Touvron et al., 2023) which employs multihead softmax attention, and Llama3 (Dubey et al., 2024), the grouped-query attention version of Transformer++. All models are trained from scratch on the training dataset, each with approximately 44 million parameters. To ensure a fair comparison, we set Rodimus*'s expansion factor $n$ to 32, aligning its parameter count with that of the other models. Other experimental configurations are also the same across all models, including batch size, learning rate, and training iterations (see Appendix D.1).

The results, summarized in Table 1, present the perplexity (PPL) of the testing data. Notably, Rodimus, a purely recurrent model, surpasses both Transformer++ and Llama3 in performance, despite having an inference complexity one power less than that of the latter two models. When compared to Mamba2, which has a state expansion factor of 128, Rodimus employs a smaller expansion factor of 32, leading to a significantly reduced hidden state size while achieving a lower PPL. Furthermore, the performance of Rodimus+ even exceeds that of Rodimus. These findings indicate that Rodimus* effectively challenges the traditional accuracy-efficiency trade-off in language modeling.

**Scaling Laws** (Hoffmann et al., 2022): We further expand the number of parameters in our models from approximately 125 million to 1.3 billion, adhering to the architectural guidelines of GPT-3 (Brown et al., 2020). For comparison, we used Mamba2 (Dao & Gu, 2024) and Transformer++ (Touvron et al., 2023) as baselines. Mamba2 represents the most advanced recurrent models, while Transformer++ exemplifies the leading softmax attention models. All models are trained on subsets of Pile (Gao et al., 2020). Detailed experimental setups are provided in Appendix D.4. Figure 4 illustrates that both Rodimus and Rodimus+ outperform Mamba2 and Transformer++ across all model sizes, ranging from 125 million to 1.3 billion parameters. Interestingly, Rodimus+, which integrates Rodimus with the SW-SKA, achieves even greater improvements over Rodimus with the

Table 2: Results of Downstream Tasks via Zero-Shot Evaluation.

| Model | Params (B) | Tokens (B) | Tokenizer | ARC-c acc_norm ↑ | ARC-e acc ↑ | HS acc_norm ↑ | LMB ppl ↓ | LMB acc ↑ | OBQA acc_norm ↑ | PIQA acc ↑ | WG acc ↑ | AVG acc* ↑ |
|---|---|---|---|---|---|---|---|---|---|---|---|---|
| Mamba£ | 0.13 | 100 | NeoX | 23.72 | **46.84** | 33.37 | 19.72 | 40.33 | 29.80 | 65.07 | **52.17** | 41.61 |
| Mamba2£ | 0.13 | 100 | NeoX | **23.81** | 45.50 | 34.13 | 20.42 | 39.55 | 29.20 | 64.47 | 51.85 | 41.22 |
| Rodimus£ | 0.13 | 100 | NeoX | 23.38 | 46.84 | 34.22 | **17.95** | 42.44 | 30.00 | 65.40 | 51.46 | **41.96** |
| GPT-Neo | 0.13 | 300 | GPT2 | 23.21 | 43.69 | 30.42 | 30.26 | 37.38 | 26.40 | 62.89 | 50.51 | 39.21 |
| OPT | 0.13 | 300 | OPT | 22.78 | 43.52 | 31.35 | 26.02 | 37.90 | 28.00 | 63.00 | 50.36 | 39.56 |
| Pythia | 0.16 | 300 | NeoX | 23.72 | 43.60 | 30.19 | 38.31 | 32.80 | 26.00 | 61.59 | 50.51 | 38.34 |
| Mamba | 0.13 | 300 | NeoX | **24.32** | 47.90 | 35.20 | 16.05 | 44.21 | 28.60 | 64.69 | 52.33 | 42.46 |
| Mamba2 | 0.13 | 300 | NeoX | 24.23 | 47.35 | 35.32 | 16.79 | 43.78 | **30.40** | **64.85** | 52.64 | 42.65 |
| RWKV4 | 0.17 | 300 | NeoX | 23.55 | 47.56 | 32.27 | 31.73 | 32.64 | 27.80 | 64.25 | 50.99 | 39.87 |
| Rodimus | 0.13 | 300 | NeoX | 23.72 | **49.07** | 35.45 | 15.75 | 45.00 | 29.20 | 64.36 | 53.35 | 42.88 |
| Rodimus+ | 0.13 | 300 | NeoX | 24.23 | 47.22 | 35.49 | **12.94** | **48.09** | 29.20 | 64.36 | **53.43** | **43.15** |
| OPT | 0.35 | 300 | OPT | 23.98 | 44.19 | 36.66 | 16.40 | 45.10 | 28.20 | 64.47 | 52.41 | 42.14 |
| Pythia | 0.41 | 300 | NeoX | 24.49 | 52.10 | 40.59 | 10.83 | 51.45 | 29.40 | 66.97 | 53.59 | 45.51 |
| BLOOM | 0.56 | 300 | BLOOM | 23.89 | 47.47 | 36.89 | 28.83 | 34.10 | 28.80 | 64.20 | 52.01 | 41.05 |
| Mamba | 0.37 | 300 | NeoX | 27.82 | 54.92 | 46.49 | **8.14** | 55.60 | 30.80 | 69.53 | 55.17 | 48.62 |
| Mamba2 | 0.37 | 300 | NeoX | 26.62 | 54.67 | 46.96 | **7.98** | **55.85** | 32.60 | 70.46 | 55.64 | 48.97 |
| RWKV4 | 0.43 | 300 | NeoX | 25.26 | 47.18 | 40.77 | 13.38 | 45.39 | 30.60 | 68.12 | 53.28 | 44.37 |
| Qwen2 | 0.5 | 7000 | Qwen2 | **28.84** | 54.97 | 49.03 | 11.66 | 50.05 | 33.20 | 69.42 | **57.62** | 49.02 |
| Rodimus | 0.46 | 150 | Rodimus | 27.65 | 55.77 | 48.78 | 10.17 | 50.65 | 32.60 | 70.73 | 55.41 | 48.80 |
| Rodimus+ | 0.47 | 150 | Rodimus | 28.58 | **57.28** | **52.13** | 10.22 | 51.14 | **35.00** | **72.91** | 53.59 | **50.09** |
| GPT-Neo | 1.3 | 300 | GPT2 | 25.77 | 56.14 | 48.91 | 7.50 | 57.21 | 33.60 | 71.22 | 55.01 | 49.69 |
| OPT | 1.3 | 300 | OPT | 29.52 | 56.94 | 53.75 | 6.65 | 57.89 | 33.20 | 71.60 | 59.59 | 51.78 |
| Pythia | 1.0 | 300 | NeoX | 27.05 | 56.94 | 47.15 | 7.92 | 56.22 | 31.40 | 70.67 | 53.51 | 48.99 |
| Pythia | 1.4 | 300 | NeoX | 28.41 | 60.48 | 51.98 | 6.08 | 61.60 | 33.20 | 70.84 | 57.30 | 51.97 |
| BLOOM | 1.1 | 300 | BLOOM | 25.51 | 51.47 | 42.97 | 17.28 | 42.64 | 29.40 | 67.19 | 55.01 | 44.88 |
| GLA | 1.3 | 100 | Mistral | 27.82 | 55.13 | 48.97 | 15.37 | 46.09 | 33.00 | 69.80 | 53.20 | 47.72 |
| RetNet | 1.3 | 100 | Mistral | 26.45 | 57.32 | 48.04 | 16.44 | 43.37 | 32.00 | 69.42 | 53.43 | 47.15 |
| HGRN | 1.3 | 100 | Mistral | 25.51 | 55.01 | 48.82 | 20.24 | 37.71 | 31.80 | 69.80 | 50.28 | 45.56 |
| HGRN2 | 1.3 | 100 | Mistral | 28.16 | 58.16 | 51.74 | 11.38 | 49.97 | 32.80 | 71.27 | 52.17 | 49.18 |
| Mamba | 1.3 | 300 | NeoX | 32.85 | 65.49 | 59.08 | **5.04** | 64.87 | 36.40 | 74.10 | 61.40 | 56.31 |
| Mamba2 | 1.4 | 300 | NeoX | 33.36 | 64.10 | 59.92 | 5.02 | 65.61 | 37.80 | 73.29 | 60.93 | 56.43 |
| RWKV4 | 1.5 | 300 | NeoX | 29.01 | 60.94 | 52.90 | 7.08 | 57.19 | 33.40 | 72.09 | 55.33 | 51.55 |
| RWKV6 | 1.6 | 1420 | RWKV6 | 33.36 | 60.69 | 61.42 | **4.61** | **67.26** | 37.40 | 73.67 | 60.38 | 56.31 |
| Rec-Gemma | 2.0 | 2000 | Gemma | 29.69 | 48.91 | 61.82 | 9.27 | 53.83 | 29.40 | 67.52 | 57.06 | 49.75 |
| Qwen2 | 1.5 | 7000 | Qwen2 | **35.75** | 65.95 | **65.51** | 5.52 | 63.61 | 36.80 | 75.19 | **65.27** | **58.30** |
| Rodimus | 1.4 | 500 | Rodimus | 36.09 | 68.01 | 62.44 | 6.33 | 59.62 | 35.60 | 74.32 | 59.75 | 56.55 |
| Rodimus+ | 1.6 | 1000 | Rodimus | **36.35** | **68.35** | 64.51 | 5.38 | 63.59 | **38.80** | **76.17** | 62.51 | **58.61** |

- Here, **bold**, underline, wavy underline indicates the best, second, and third best result, respectively. For the column "Tokens", underline indicates models using fewer tokens. The superscript £ denotes models trained from scratch using the same configuration for fair comparison.

increase of the model size. As the number of layers and parameters grows, the benefits of the two-hop residual connections may become more pronounced (Ma et al., 2024). Consequently, Rodimus+ demonstrates superior scaling potential, especially when approaching 1.3 billion parameters.

**Downstream Evaluation**: We evaluate Rodimus* across various downstream tasks, as outlined in Gu & Dao (2023). These tasks include content analysis (LAMBADA (Paperno et al., 2016)), commonsense reasoning (PiQA (Bisk et al., 2020) and HellaSwag (Zellers et al., 2019)), coreference resolution (WinoGrande (Sakaguchi et al., 2019)), reading comprehension (OpenBookQA (Mihaylov et al., 2018)), and professional examinations (ARC-Easy and ARC-Challenge (Clark et al., 2018)). The evaluation results are obtained using the `lm-evaluation-harness` (Gao et al., 2023). For comparison, we choose well-known open-source models as baselines, indicating the number of tokens and the tokenizers used, similar approaches are commonly adopted in prior works (Gu & Dao, 2023; Dao & Gu, 2024). Our experimental structure is as follows: (i) We first train both Rodimus and Mamba series with 130M parameters from scratch using the same settings on 100B tokens from the Pile dataset (denoted as Rodimus£, Mamba£ and Mamba2£) for a fair comparison. (ii) We then train Rodimus*-130M on 300B tokens from Pile and compare it to other similarly sized models also trained on 300B tokens from Pile, noting that the training corpora for these models may still differ. (iii) We evaluate Rodimus* with about 460M parameters against other open-source models like BLOOM-560M and Qwen2-500M. Since these models are trained on self-cleaned high-quality data, we also utilize curated datasets, including FineWeb (Penedo et al., 2024), Pile (Gao et al., 2020), etc., to train our Rodimus* on 150B tokens. (iv) To fully explore the model's capabilities, we train Rodimus+-1.6B on 1T tokens from the curated dataset, while Rodimus-1.4B is trained on 500B tokens due to resource constraints. Both models are compared against other open-source models of comparable sizes, such as BLOOM, RWKV6, and Qwen2, which also use self-cleaned datasets for training. Note that the tokenizers used in the last two experiments are developed based on their respective curated datasets. The specific results are presented in Table 2. More details can be found in Appendices D.5 and D.6.

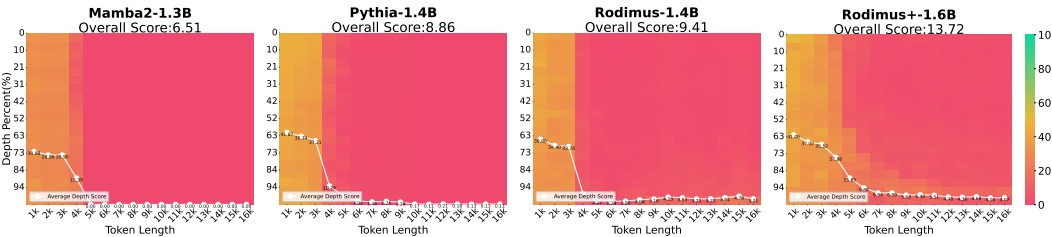

Figure 5: Results on NeedleBench. The average accuracy of each subtask of NeedleBench in contexts of different lengths. Overall Score represents the global average score.

In experiment (i), where Rodimus$^£$, Mamba$^£$, and Mamba2$^£$ are trained under identical conditions, Rodimus outperforms the Mamba series on downstream tasks, validating the effectiveness of the DDTS design. In experiment (ii), where all models are trained with the same number of tokens from Pile, Rodimus outperforms all existing models across the benchmarks. Moreover, Rodimus+ achieves even better results, indicating that the proposed hybrid structure is more effective for downstream tasks. Moving to experiment (iii), we note that the number of training tokens can impact performance. Due to a relatively small token count, Rodimus-460M underperforms compared to Qwen2-500M and Mamba2-370M but still surpasses Mamba-370M, which is trained on 300B tokens. In contrast, Rodimus+-470M delivers the best average result, even though it is trained on only 150B tokens, further validating the proposed hybrid structure's effectiveness. Lastly, in experiment (iv), training larger models on a larger corpus reinforces Rodimus*'s potential as a practical LLM. Specifically, Rodimus-1.4B outshines all existing purely recurrent models of similar sizes, including those from the Mamba and RWKV series. Furthermore, Rodimus+-1.6B surpasses the performance of both Qwen2-1.5B and Recurrent-Gemma-2B, despite being trained on only 1T tokens, in contrast to the 7T and 2T tokens used for the latter two models, respectively. In addition, we refer the readers to Appendix E.3, where we continue-pretrain Rodimus+ as a practical LLM for math and code.

## 4.2 RECALL BENCHMARKS

In this section, we assess the recall or retrieval capabilities of Rodimus* in the context of the MQAR task and NeedleBench. The MQAR task involves training a small-parameter model from scratch using a specific dataset, while NeedleBench evaluates models trained on a large-scale corpus. Due to space limitations, we offer a brief overview here and direct readers to Appendices D.7 and D.8 for more details. For MQAR, we primarily compare Rodimus* with other recurrent models and find that Rodimus* consistently outperforms existing models as the state expansion factor $n$ (see Figure 1b) or the model dimension $d$ (see Figure 6) increases, thanks to the proposed DDTS mechanism. For NeedleBench, both Rodimus and Rodimus+ even exceed the performance of Pythia, a softmax attention-based model (see Figure 5). It is noteworthy that many studies suggest pre-trained recurrent models often struggle with challenging recall-intensive tasks like NeedleBench when compared to models utilizing full attention mechanisms (Waleffe et al., 2024). The superior performance of Rodimus and Rodimus+ indicates that the Rodimus family effectively balances the accuracy-efficiency trade-off, particularly in enhancing recall ability. More results on other recall benchmarks, such as FDA, SWDE, NQ, SQuAD, TriviaQA, and DROP, can be found in Appendix D.9.

## 4.3 ABLATION STUDIES

In this section, we summarize the key conclusions from our ablation studies. A detailed analysis is provided in Appendix D.10. (i) Our assessment of the components in Rodimus reveals that the inclusion of $\boldsymbol{g}_t$, $\boldsymbol{\tau}_t$, and $\hat{\boldsymbol{\beta}}_t$ is crucial for the DDTS framework. However, the incorporation of $\boldsymbol{\beta}_t$ tends to complicate the training process, leading to suboptimal results. Additionally, the two-hop residual connection is beneficial as the model depth increases. (ii) In terms of head compression methods, we find that SKA offers a balanced compromise between GQA and MHA, delivering superior performance compared to GQA while utilizing fewer parameters relative to MHA, irrespective of the backbone models used. (iii) We have determined that the optimal hyperparameter values are $\ell = 16$ and $n = 64$. Notably, the Rodimus configuration with $n = 64$ operates with a state size that is only half that of Mamba2, yet it outperforms Mamba2 on various benchmarks.

## 5 CONCLUSION

We introduce here Rodimus and Rodimus+, two innovative models that utilize semantic, token, and head compression. We refer the readers to Appendices F and G for more discussion and future work.

ACKNOWLEDGMENTS

The work was supported by the National Natural Science Foundation of China (No. 62325109, U21B2013) and Ant Group.

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

# A   MAIN NOTATION INTRODUCTION

Table 3: Meanings of Notations

| Notation | Size | Meaning |
|---|---|---|
| $T$ | Constant | The length of the context. |
| $d$ | Constant | The model dimension. |
| $m$ | Constant | The dimensionality of the hidden state corresponding to the token mixer. |
| $n$ | Constant | The expansion factor of matrix hidden state, when is set to 1 in vector hidden state. |
| $d_h$ | Constant | The head dimension in the multi-head mechanism. |
| $\mathbf{1}_x$ | $1 \times x$ | The all-ones vector of dimension $x$. |
| $\boldsymbol{X}$ | $T \times d$ | The input sequence of module. |
| $\boldsymbol{O}$ | $T \times d$ | The output sequence of module. |
| $\boldsymbol{Q}$ | $T \times n$ | The query. |
| $\boldsymbol{K}$ | $T \times n$ | The key. |
| $\boldsymbol{V}$ | $T \times m$ | The value. |
| $\boldsymbol{W}_Q$ | $d \times n$ | The weight matrix of the query. |
| $\boldsymbol{W}_K$ | $d \times n$ | The weight matrix of the key. |
| $\boldsymbol{W}_V$ | $d \times m$ | The weight matrix of the value. |
| $\boldsymbol{S}_t$ | $n \times m$ | The hidden state at stamp $t$. |
| $\boldsymbol{A}_t$ | $n \times m$ | The transition matrix at stamp $t$. |
| $\boldsymbol{B}_t$ | $n \times m$ | The input matrix at stamp $t$. |
| $\boldsymbol{C}_t$ | $1 \times n$ | The output matrix at stamp $t$. |
| $\boldsymbol{\alpha}_t$ | $1 \times n$ | The gating mechanism along the dimension $n$ acts on the transition. |
| $\hat{\boldsymbol{\alpha}}_t$ | $1 \times n$ | The gating mechanism along the dimension $n$ acts on the input, which is negatively correlated with $\boldsymbol{\alpha}_t$. |
| $\boldsymbol{\beta}_t$ | $1 \times m$ | The gating mechanism along the dimension $m$ acts on the transition. |
| $\hat{\boldsymbol{\beta}}_t$ | $1 \times m$ | The gating mechanism along the dimension $m$ acts on the input, which is negatively correlated with $\boldsymbol{\beta}_t$. |
| $\boldsymbol{u}_t$ | $n \times m$ | The input term of recurrent calculations. |
| $\boldsymbol{x}'_t$ | $1 \times m$ | The output of ShortConv. |
| $\boldsymbol{g}_t$ | Variable | The gating mechanism of selection or decay. In Rodimus, it's a selection gate with the size of $1 \times n$. |
| $\boldsymbol{W}_g$ | $m \times n$ | The weight matrix of $\boldsymbol{g}_t$. |
| $\boldsymbol{b}_g$ | $1 \times n$ | The bias of $\boldsymbol{g}_t$. |
| $\boldsymbol{\tau}_t$ | $1 \times n$ | The temperature gate in Rodimus. |
| $\boldsymbol{W}_\tau$ | $m \times n$ | The weight matrix of $\boldsymbol{\tau}_t$ in Rodimus. |
| $\boldsymbol{b}_\tau$ | $1 \times n$ | The bias of $\boldsymbol{\tau}_t$ in Rodimus. |
| $\ell$ | Constant | The low-rank dimensionality of projections. |
| $\boldsymbol{W}_{\hat{\beta}}^1$ | $m \times \ell$ | The down-projection weight matrix of $\hat{\boldsymbol{\beta}}_t$ in Rodimus. |
| $\boldsymbol{W}_{\hat{\beta}}^2$ | $\ell \times m$ | The up-projection weight matrix of $\hat{\boldsymbol{\beta}}_t$ in Rodimus. |
| $\boldsymbol{b}_{\hat{\beta}}$ | $1 \times m$ | The bias of $\hat{\boldsymbol{\beta}}_t$ in Rodimus. |
| $\boldsymbol{d}_t$ | $1 \times m$ | The feedthrough matrix in SSMs. |

# B   METHOD DETAILS

## B.1   SIMILAR FUNCTIONAL FORMS OF OTHER METHODS

We summarize the functional forms of $\boldsymbol{A}_t$ and $\boldsymbol{B}_t$ used in state-of-the-art methods in Table 4.

As illustrated in the above table, the specific formulation of DDTS provides several notable advantages:

- **Dynamic Control of Hidden State Updates**: DDTS allows for data-dependent selection through $\boldsymbol{A}_t$ and $\boldsymbol{B}_t$ when writing and deleting information, enabling dynamic control of hidden state updates. This feature is discussed in Section 3.1.2, with a formal proof provided in Appendix C.2.
- **Efficient Parallelization**: As DDTS performs data-dependent selection solely along the dimension $n$, it supports the implementation of chunkwise parallel algorithms. This capability enables efficient parallel training while maintaining flexibility, as elaborated in Section 3.1.3 and Appendix B.3.

Table 4: Comparison of $\boldsymbol{A}_t$ and $\boldsymbol{B}_t$ across various models. For models employing the multi-head mechanism, only the gates for a single head are presented. The term "Gate Dim" represents the dimensionality of the gates $\boldsymbol{g}_t$ in $\boldsymbol{A}_t$ and $\boldsymbol{B}_t$, and "L.A." denotes linear attention without scaling (Katharopoulos et al., 2020; Qin et al., 2022a). Elements with the subscript $t$ are data-dependent gates, with most elements being learnable. Notably, the non-learnable constants include $\gamma$ in RetNet, $\tau$ in GLA and gRetNet, and $c$ in Hawk. Although the "State Size" of RWKV6 is smaller than that of Rodimus, the use of DDTS in Rodimus yields superior performance across all metrics (see Tables 2).

| Model | Gate Dim | State Size $n \times m$ | Parallelization like L.A. | Recurrent Type | $\boldsymbol{A}_t$ | $\boldsymbol{B}_t$ |
|---|---|---|---|---|---|---|
| L.A. (Katharopoulos et al., 2020) | No Gates | $256 \times d$ | ✓ | No Gates | $\mathbf{1}_n^\top \mathbf{1}_m$ | $\mathbf{1}_n^\top \mathbf{1}_m$ |
| RetNet (Sun et al., 2023a) | $1 \times 1$ | $256 \times d$ | ✓ | Decay | $g \cdot \mathbf{1}_n^\top \mathbf{1}_m$ | $\mathbf{1}_n^\top \mathbf{1}_m$ |
| gRetNet (Sun et al., 2024) | $1 \times 1$ | $256 \times d$ | ✓ | Decay | $\left(\sigma(g_t)^{1/\tau}\right) \cdot \mathbf{1}_n^\top \mathbf{1}_m$ | $\mathbf{1}_n^\top \mathbf{1}_m$ |
| GLA (Yang et al., 2024b) | $1 \times n$ | $256 \times d$ | ✓ | Decay | $\left(\sigma(\boldsymbol{g}_t)^{1/\tau}\right)^\top \mathbf{1}_m$ | $\mathbf{1}_n^\top \mathbf{1}_m$ |
| RWKV6 (Peng et al., 2024) | $1 \times n$ | $64 \times d$ | ✓ | Decay | $\exp\left(-\exp(\boldsymbol{g}_t)\right)^\top \mathbf{1}_m$ | $\mathbf{1}_n^\top \mathbf{1}_m$ |
| HGRN2 (Qin et al., 2024a) | $1 \times n$ | $128 \times d$ | ✓ | Selection | $\left(1 - \sigma(\boldsymbol{g}_t)\right)^\top \mathbf{1}_m$ | $\left(\sigma(\boldsymbol{g}_t)\right)^\top \mathbf{1}_m$ |
| Hawk (De et al., 2024) | $1 \times d$ | $1 \times d$ | ✗ | Selection | $\sqrt{1 - (a^{c \cdot \sigma(\boldsymbol{g}_t)})^2}$ | $a^{c \cdot \sigma(\boldsymbol{g}_t)}$ |
| S4D (Gu et al., 2022a) | $1 \times d$ | $16 \times d$ | ✗ | Selection | $\exp(-\boldsymbol{A} \odot \mathbf{1}_n^\top \boldsymbol{g})$ | $\mathbf{1}_n^\top \zeta(\boldsymbol{g})$ |
| Mamba (Gu & Dao, 2023) | $1 \times d$ | $16 \times 2d$ | ✗ | Selection | $\exp(-\boldsymbol{A} \odot \mathbf{1}_n^\top \boldsymbol{g}_t)$ | $\mathbf{1}_n^\top \zeta(\boldsymbol{g}_t)$ |
| Mamba2 (Dao & Gu, 2024) | $1 \times 1$ | $128 \times 2d$ | ✓ | Selection | $\exp(-a \cdot g_t) \cdot \mathbf{1}_n^\top \mathbf{1}_m$ | $\zeta(g_t) \cdot \mathbf{1}_n^\top \mathbf{1}_m$ |
| **Rodimus** | $1 \times n$ | $64 \times 2d$ | ✓ | DDTS | $\exp(-\boldsymbol{g}_t \odot \boldsymbol{\tau}_t)^\top \mathbf{1}_m$ | $(\boldsymbol{g}_t^{\boldsymbol{\tau}_t})^\top \hat{\boldsymbol{\beta}}_t$ |

- **Enhanced Flexibility through the Temperature Gate**: The temperature gate $\tau_t$ introduces additional flexibility in the selection mechanism, allowing for more precise control over information compression and representation. This aspect sets our approach apart from existing works, as noted in Section 3.1.2 and detailed in Appendix E.2.

## B.2 RELATED WORK

**Linear Attention with Gates**: Linear attention replaces the traditional exponential kernel, $\exp(\boldsymbol{q}_t^\top \boldsymbol{k}_t)$, with the dot-product feature map $\phi(\boldsymbol{q}_t)^\top \phi(\boldsymbol{k}_t)$ (Katharopoulos et al., 2020). This approach allows inference to be reformulated as a linear RNN with a matrix hidden state. However, without gating mechanisms, it often struggles to effectively capture input and forget historical information, resulting in irrelevant information within the hidden state (Van Der Westhuizen & Lasenby, 2018). To address this, gating mechanisms such as purely decay (e.g., GLA (Yang et al., 2024b)) and selection mechanisms for associating input and decay (e.g., HGRN2 (Qin et al., 2024a)) have been introduced.

These improvements, however, often lead to the issue of sacrificing hidden state size to enhance performance (Qin et al., 2024a), lacking thorough analysis and design of gates, which hinders performance even with larger hidden states. In Section 3.1.1, we analyze effective gate design and propose a solution for a more efficient gating mechanism named DDTS.

**State-Space Models**: Recent advancements in state-space models based on HiPPO (Gu et al., 2020) have led to notable results in long sequence modeling, inspiring further work like S4 (Gu et al., 2022b), S5 (Smith et al., 2023), and H3 (Fu et al., 2023). Mamba (Gu & Dao, 2023) builds on S4D (Gu et al., 2022a), the parameterized diagonal version of S4, making state-space model (SSM) parameters data-dependent. Its input-based selection mechanism arises from ZOH discretization. However, Mamba's diagonal state space structure limits calculations with larger hidden states expanded by the expansion factor $n$, restricting memory capacity improvement.

Mamba2 (Dao & Gu, 2024) addresses this by using a scalar state space structure instead of the original diagonal state space structure in Mamba. This change allows Mamba2 to be trained with chunkwise parallelization, similar to linear attention (Sun et al., 2023a; Ma et al., 2022), permitting an expansion factor of 128. However, this approach mirrors linear attention's challenge of over-relying on hidden state size while neglecting gating design. The oversimplified gating mechanism struggles to approximate the original softmax attention performance (see Section 3.1.1).

Our method features a similar architecture to Mamba2, enabling chunk-wise parallel calculations, while our gating mechanism, DDTS, better approximates softmax attention.

**Softmax Attention with Custom Attention Mask**: Several studies have enhanced the attention mechanism while retaining the softmax function. One approach involves modifying the attention scope by integrating prior knowledge to create diverse custom attention masks. During inference, these masks function similarly to token-level sequence compression, storing unmasked tokens in the KV cache (Xiao et al., 2024). Many studies observe local behavior in attention, leading to the use of local attention instead of full attention, as seen in models like LongFormer (Beltagy et al., 2020). Additionally, some methods employ learnable patterns, such as the Routing Transformer (Roy et al., 2021), to construct attention masks in a data-driven manner. However, custom attention masks only support token-level compression, which can result in information loss for masked tokens.

In Rodimus+, the Rodimus block can compress long contexts and transform token embeddings into context embeddings by dynamically fusing multiple tokens to create semantic compression, as shown in Figure 2b. This enables the local attention in SW-SKA to capture token information beyond the window.

### B.3    CHUNKWISE PARALLEL REPRESENTATION OF THE RODIMUS BLOCK

Let $\mathcal{A}_t = \prod_{j=1}^t \boldsymbol{\alpha}_j$ denote the cumulative product of gates $\boldsymbol{\alpha}_j$, and define $\mathcal{A} = \{\mathcal{A}_1, \ldots, \mathcal{A}_T\} \in \mathbb{R}^{T \times n}$, then $\boldsymbol{\Lambda}_t = \prod_{j=1}^t \boldsymbol{\beta}_j$ denote the cumulative product of gates $\boldsymbol{\beta}_j$, and define $\boldsymbol{\Lambda} = \{\boldsymbol{\Lambda}_1, \ldots, \boldsymbol{\Lambda}_T\} \in \mathbb{R}^{T \times m}$. After the following derivation, we can obtain the parallel form of the Rodimus block:

$$
\begin{aligned}
\boldsymbol{S}_t &= (\boldsymbol{\alpha}_t^\top \boldsymbol{\beta}_t) \odot \boldsymbol{S}_{t-1} + (\hat{\boldsymbol{\alpha}}_t^\top \hat{\boldsymbol{\beta}}_t) \odot (\boldsymbol{k}_t^\top \boldsymbol{v}_t) \\
&= (\boldsymbol{\alpha}_t^\top \boldsymbol{\beta}_t) \odot \boldsymbol{S}_{t-1} + (\hat{\boldsymbol{\alpha}}_t \odot \boldsymbol{k}_t)^\top (\hat{\boldsymbol{\beta}}_t \odot \boldsymbol{v}_t), \\
\boldsymbol{o}_t &= \boldsymbol{q}_t \boldsymbol{S}_t,
\end{aligned}
\tag{14}
$$

from the derivation in (Yang et al., 2024b), we can have:

$$
\begin{aligned}
\boldsymbol{o}_t &= \boldsymbol{q}_t \sum_{i=1}^t \Big( \prod_{j=i+1}^t (\boldsymbol{\alpha}_j^\top \boldsymbol{\beta}_j) \odot (\hat{\boldsymbol{\alpha}}_i \odot \boldsymbol{k}_i)^\top (\hat{\boldsymbol{\beta}}_i \odot \boldsymbol{v}_i) \Big) \\
&= \boldsymbol{q}_t \sum_{i=1}^t \Big( (\prod_{j=i+1}^t \boldsymbol{\alpha}_j)^\top (\prod_{j=i+1}^t \boldsymbol{\beta}_j) \odot (\hat{\boldsymbol{\alpha}}_i \odot \boldsymbol{k}_i)^\top (\hat{\boldsymbol{\beta}}_i \odot \boldsymbol{v}_i) \Big) \\
&= \boldsymbol{q}_t \sum_{i=1}^t \Big( (\prod_{j=i+1}^t \boldsymbol{\alpha}_j \odot \boldsymbol{k}_i \odot \hat{\boldsymbol{\alpha}}_i)^\top (\prod_{j=i+1}^t \boldsymbol{\beta}_j \odot \boldsymbol{v}_i \odot \hat{\boldsymbol{\beta}}_i) \Big) \\
&= \sum_{i=1}^t \Big[ \boldsymbol{q}_t (\prod_{j=i+1}^t \boldsymbol{\alpha}_j \odot \hat{\boldsymbol{\alpha}}_i \odot \boldsymbol{k}_i)^\top \Big] (\prod_{j=i+1}^t \boldsymbol{\beta}_j \odot \hat{\boldsymbol{\beta}}_i \odot \boldsymbol{v}_i) \\
&= \sum_{i=1}^t \Big[ (\boldsymbol{q}_t \odot \mathcal{A}_t)(\hat{\boldsymbol{\alpha}}_i \odot \boldsymbol{k}_i / \mathcal{A}_i)^\top \Big] (\hat{\boldsymbol{\beta}}_i \odot \boldsymbol{v}_i / \boldsymbol{\Lambda}_i) \odot \boldsymbol{\Lambda}_t,
\end{aligned}
\tag{15}
$$

$$
\begin{aligned}
\tilde{\boldsymbol{Q}} &= \boldsymbol{Q} \odot \mathcal{A}, \quad \tilde{\boldsymbol{K}} = \boldsymbol{K} / \mathcal{A} \odot \hat{\boldsymbol{\alpha}}, \quad \tilde{\boldsymbol{V}} = \boldsymbol{V} / \boldsymbol{\Lambda} \odot \hat{\boldsymbol{\beta}}, \\
\boldsymbol{O} &= (\tilde{\boldsymbol{Q}} \tilde{\boldsymbol{K}}^\top \odot \boldsymbol{M}) \tilde{\boldsymbol{V}}, \\
\tilde{\boldsymbol{O}} &= \boldsymbol{O} \odot \boldsymbol{\Lambda}.
\end{aligned}
\tag{16}
$$

It follows from Eq. (16) that the parallel formula can be written as:

$$
\tilde{\boldsymbol{O}} = \big( (\boldsymbol{Q} \odot \mathcal{A})(\boldsymbol{K} / \mathcal{A} \odot \hat{\boldsymbol{\alpha}})^\top \odot \boldsymbol{M} \big)(\boldsymbol{V} / \boldsymbol{\Lambda} \odot \hat{\boldsymbol{\beta}}) \odot \boldsymbol{\Lambda}.
\tag{17}
$$

Since we set $\beta_j = 1$ in Section 3.1.1, the above equation can be simplified as:

$$
\boldsymbol{O} = \big( (\boldsymbol{Q} \odot \mathcal{A})(\boldsymbol{K} / \mathcal{A} \odot \hat{\boldsymbol{\alpha}})^\top \odot \boldsymbol{M} \big)(\boldsymbol{V} \odot \hat{\boldsymbol{\beta}}),
\tag{18}
$$

which the parallel form for the proposed Rodimus block. However, the complexity of computing $\boldsymbol{O}$ via Eq. (18) is quadratic. To further reduce the training complexity, we exploit a chunkwise integration of the recurrent and parallel forms, in analogy to other linear attention approaches (Yang

et al., 2024b). Specifically, we first divide the entire sequence into $T/B$ consecutive chunks, each with length $B$. For each chunk, the parallel computation as described in Eq. (18) is employed to obtain the intra-chunk output $\boldsymbol{O}^{\text{intra}}$. Subsequently, information between chunks is integrated using a recurrence approach akin to Eq. (11):

$$\boldsymbol{S}_{[i]} = \boldsymbol{S}_{[i-1]} \odot ((\textstyle\prod_{j=(i-1)B+1}^{iB} \boldsymbol{\alpha}_j)^\top \mathbf{1}) + (\boldsymbol{K}_{[i]} \odot \hat{\boldsymbol{\alpha}}_{[i]} \odot \textstyle\prod_{j=(i-1)B+b+1}^{iB} \boldsymbol{\alpha}_j)^\top (\boldsymbol{V}_{[i]} \odot \hat{\boldsymbol{\beta}}_{[i]}),$$

$$\boldsymbol{O}^{\text{inter}}_{[i]} = (\boldsymbol{Q}_{[i]} \odot \textstyle\prod_{j=(i-1)B+1}^{(i-1)B+b} \boldsymbol{\alpha}_j)\boldsymbol{S}_{[i]}, \tag{19}$$

where $[i]$ represents the $i$-th chunk. The parallel computations within each chunk can further leverage tensor cores to speed up matrix operations (Yang et al., 2024b; Dao & Gu, 2024). Additionally, this sequential form for inter-chunk computation reduces overall computational complexity from quadratic to subquadratic and optimizes memory utilization during training. To further optimize training efficiency, we incorporate `FlashLinearAttention`[1].

### B.4 RODIMUS+'S RELATIONSHIP TO EXISTING HYBRID MODELS

Here we focus on models that integrate self-attention with recurrent architectures. One notable approach is Griffin (De et al., 2024; Botev et al., 2024), which combines sliding window attention with a linear RNN. However, unlike Rodimus+, which employs matrix hidden states, the hidden state in Griffin's linear RNN is a vector, limiting its capacity to retain historical information. Another hybrid model, Based (Arora et al., 2024b), also integrates linear attention and sliding window attention. Unfortunately, it lacks a decay or gating mechanism within its linear attention framework, making it susceptible to the attention dilution problem. On a different note, Samba (Ren et al., 2024) explores various methods to combine Mamba with self-attention. The original version of Samba inserts MLP (FFN) layers between Mamba and self-attention, which neglects Mamba's channel mixing capabilities and introduces unnecessary parameters. Another variant of Samba, Mamba-SWA-MLP, then removes the MLP between Mamba and self-attention, but it does not implement the two-hop residual connection used in Rodimus+. Different from the aforementioned works, Mamba-2-Attention (Dao & Gu, 2024) seeks to enhance retrieval (i.e., recall) capabilities by replacing some of the SSM layers with self-attention. This approach contrasts with the model architecture of Rodimus+, which regularly incorporates SWA between SSM layers. Moreover, determining the optimal number of layers to replace with attention requires further experimentation, and the introduction of full attention layers with their quadratic complexity complicates model scaling. Lastly, Jamba (Lieber et al., 2024) combines the Transformer block with Mamba and MoE layers, yet it also adds extra MLP layers after Mamba, similar to Samba. Additionally, establishing the optimal number of Transformer blocks remains an unresolved challenge.

### B.5 MODELS IN EXPERIMENTS

**Transformer** (Brown et al., 2020): The foundational architecture with a causal attention mask used in GPT-3.

**Transformer++** (Touvron et al., 2023): An enhancement of the Transformer, Transformer++ incorporates RoPE, GLU, and RMSNorm for improved performance. It is currently the most robust and widely used Transformer configuration (Gu & Dao, 2023).

**Mamba** (Gu & Dao, 2023): Mamba makes the original SSM parameters data-dependent and introduces an I/O-aware associative scan.

**Mamba2** (Dao & Gu, 2024): Mamba2 simplifies the diagonal state space structure in Mamba to the scalar state space structure, enhancing computational efficiency through chunkwise parallelism algorithms like linear attention.

**GPT-Neo** (Black et al., 2021): Similar to the Transformer, GPT-Neo employs full attention and local attention in alternating layers and uses ALiBi for positional encoding.

**OPT** (Zhang et al., 2022): OPT mirrors the Transformer architecture but utilizes learnable positional encoding.

---

[1] `https://github.com/sustcsonglin/flash-linear-attention`

**BLOOM** (Le Scao et al., 2023): BLOOM is akin to the Transformer, except it uses ALiBi for positional encoding.

**Pythia** (Biderman et al., 2023): An improvement upon the Transformer, Pythia uses RoPE for positional encoding.

**RWKV4** (Peng et al., 2023): RWKV4 integrates techniques such as token-shift to merge RNN-like states with Transformer-style attention mechanisms.

**RWKV6** (Peng et al., 2024): An enhanced version of RWKV4, RWKV6 features a more flexible gating mechanism and introduces multi-headed matrix-valued states to boost the memory capacity of the recurrent model.

**RetNet** (Sun et al., 2023a): RetNet applies RoPE with a fixed scalar decay $\gamma$ to linear attention.

**GLA** (Yang et al., 2024b): GLA omits positional encoding on linear attention, opting instead for data-dependent decay.

**HGRN** (Qin et al., 2024b): HGRN introduces a decay lower bound based on traditional linear RNNs, employs recurrent calculation in the complex domain, and uses a selection mechanism in the real domain.

**HGRN2** (Qin et al., 2024a): HGRN2 removes complex domain calculations from HGRN and introduces state expansion similar to Mamba2 and linear attention.

**Qwen2** (Yang et al., 2024a): Built on Transformer++, Qwen2 uses GQA to reduce inference costs.

**Llama3** (Dubey et al., 2024): Llama3's architecture is similar to that of Qwen2.

**RecurrentGemma** (Botev et al., 2024): RecurrentGemma is a series of open language models built upon Google's innovative Griffin architecture (De et al., 2024).

## C PROOF

### C.1 LEMMA FOR SELECTION MECHANISM

**Lemma 1.** *For the recurrent model defined below,*

$$\boldsymbol{S}_t = \phi_A(\boldsymbol{\eta}_t) \odot \boldsymbol{S}_{t-1} + \phi_B(\boldsymbol{\eta}_t) \odot \boldsymbol{u}_t, \tag{20}$$

*where $\boldsymbol{S}_t \in \mathbb{R}^{n \times m}, \boldsymbol{x}_t \in \mathbb{R}^{1 \times m}, \boldsymbol{u}_t = \phi_u(\boldsymbol{x}_t) \in \mathbb{R}^{n \times m}, \boldsymbol{\eta}_t = \phi_\eta(\boldsymbol{x}_t) \in \mathbb{R}^{n \times m}$. If the following condition is met, it is a model with the selection mechanism.*

$$\forall i \in \{0, 1, \ldots, n\} \ \forall j \in \{0, 1, \ldots, m\}, \frac{d\phi_A(\boldsymbol{\eta}_{t,i,j})}{d\boldsymbol{\eta}_{t,i,j}} \frac{d\phi_B(\boldsymbol{\eta}_{t,i,j})}{d\boldsymbol{\eta}_{t,i,j}} < 0. \tag{21}$$

*Proof.* Since the product of the gradient of $\phi_A(\boldsymbol{\eta}_{t,i,j})$ and $\phi_B(\boldsymbol{\eta}_{t,i,j})$ is negative, $\phi_A(\boldsymbol{\eta}_{t,i,j})$ is negatively correlated with $\phi_B(\boldsymbol{\eta}_{t,i,j})$, satisfying the selection mechanism.

□

### C.2 PROOF OF PROPOSITION 1

To prove Proposition 1, we first verify that Rodimus satisfies Lemma 1. In Rodimus, $\boldsymbol{u}_t = \boldsymbol{k}_t^\top \boldsymbol{v}_t$, and we define $\boldsymbol{\eta}_t = (\boldsymbol{x}_t \boldsymbol{W}_g + \boldsymbol{b}_g)^\top \boldsymbol{1}_m$. Consequently, $\boldsymbol{g}'_t = \text{softplus}(\boldsymbol{\eta}_t)$. For simplicity, we omit temperature $\boldsymbol{\tau}'_t$ in this proof. We define $\phi_A(\boldsymbol{\eta}_t) = \exp(-\boldsymbol{g}'_t)$ and $\phi_B(\boldsymbol{\eta}_t) = \boldsymbol{g}'_t$, which corresponds to Eq. (11). Finally, we find that:

$$\frac{d\phi_A(\boldsymbol{\eta}_{t,i,j})}{d\boldsymbol{\eta}_{t,i,j}} \frac{d\phi_B(\boldsymbol{\eta}_{t,i,j})}{d\boldsymbol{\eta}_{t,i,j}} = -\sigma(\boldsymbol{\eta}_{t,i,j})^2 (1 - \sigma(\boldsymbol{\eta}_{t,i,j})) < 0, \tag{22}$$

where $\sigma$ is the sigmoid function. Thus, Rodimus satisfies the selection mechanism.

# D  EXPERIMENTAL DETAILS

## D.1  LANGUAGE MODELING ON WIKITEXT-103

The training settings are listed in Table 5. For the models trained on WikiText-103, the number of layers and dimensions are set to 6 and 512, respectively. The main difference from Qin et al. (2024b) is that we increase the total batch size, which reduces the training steps to obtain results more quickly.

Table 5: Settings details for WikiText-103 and the Pile subset.

| Dataset | WikiText-103 | Pile subset |
|---|---|---|
| Tokenizer method | BPE | GPT-Neox Tokenizer |
| Vocab size | 50265 | 50277 |
| Sequence length | 512 | 2048 |
| Total Batch size | 256 | 256 |
| Training steps | 25000 | 13351 |
| Warmup steps | 2000 | 572 |
| Peak learing rate | 5e-4 | 5e-4 |
| Min learning rate | 1e-5 | 1e-5 |
| Optimizer | AdamW | AdamW |
| Adam $\beta_1$ | 0.9 | 0.9 |
| Adam $\beta_2$ | 0.95 | 0.95 |
| Weight decay | 0.1 | 0.1 |
| Gradient clipping | 1.0 | 1.0 |

## D.2  LANGUAGE MODELING ON PILE

In addition to WikiText-103, we also train various model architectures, including Transformer++ (full attention), Transformers++ with sliding window attention (sparse attention), HGRN2 (linear attention), Mamba/Mamba2 (state-space models), Rodimus, and Rodimus+, using 7B tokens in the Pile dataset. This dataset is considerably larger than WikiText-103. This training aims to investigate the language modeling performance across different model architectures, with the training settings adhering to the 350M parameter configuration outlined in Table 8. The results are summarized in Table 6.

Consistent with the findings in Table 1, Rodimus demonstrates superior language modeling performance compared to other models, including Transformer++ and Mamba2. However, unlike the results from Table 1, we find that Rodimus+ and Rodimus exhibit similar performance levels under this scenario. We attribute this similarity to the limited scale of the dataset. Note that Rodimus+ needs to distribute the tasks of global and local dependency modeling respectively to the Rodimus block and the SW-SKA, making it more challenging to train than the original Rodimus and thus requiring more data. Despite this, Rodimus+ shows greater potential for enhanced language modeling performance compared to Rodimus when deployed at a larger parameter scale, as illustrated in Figure 4.

## D.3  ACCURACY-EFFICIENCY TRADE-OFF

We first present the accuracy-efficiency trade-off among Rodimus*, recurrent models (GLA, HGRN-2, Mamba, and Mamba-2), and sparse attention models (StreamingLLM) using the WikiText-103 dataset. The experimental settings are detailed in the WikiText-103 column of Table 5, while the results are illustrated in Figure 1a. To manipulate the memory footprint (focusing solely on the cache), we adjust the scaling factor $n$ for the recurrent models. In contrast, for StreamingLLM, we modify the number of recent tokens or the size of the sliding window. It is important to note that the input history sequence length is set at 512, which is also the recent token size corresponding to the rightmost endpoint of the StreamingLLM curve (i.e., the blue curve). This configuration aligns with that of the standard Transformer, which attends to the entire historical token sequence. Consequently,

Table 6: Language modeling on the Pile dataset.

| Model | Valid PPL |
|---|---|
| Transformer++ | 6.04 |
| Transformer++-SWA | 6.13 |
| HGRN2 | 6.08 |
| Mamba | 6.19 |
| Mamba2 | 6.02 |
| Rodimus | 5.88 |
| Rodimus+ | 5.88 |

the performance of the standard Transformer can be referenced at the rightmost endpoint of the StreamingLLM curve.

As demonstrated in Figure 1a, Rodimus* exhibits superior modeling performance compared to recurrent models, even when operating under the same fixed memory footprint. Furthermore, Rodimus* also surpasses both the sparse attention model (StreamingLLM) and the standard Transformer baseline. These findings underscore Rodimus*'s capacity to effectively balance accuracy and memory efficiency, achieving a substantial trade-off between the two.

To further validate the memory efficiency of Rodimus, we conduct experiments using Mamba2, Rodimus, and StreamingLLM on a larger dataset, as shown in Table 7. The foundational experimental settings are aligned with the 125M configurations described in Table 8. Once again, we adjust the memory footprint by modifying the expansion factor $n$. The results of PPL summarized in Table 7 indicate that Rodimus continues to outperform the accuracy-efficiency trade-off when trained at a larger scale compared to WikiText-103.

Table 7: Modifying the memory footprint to test the LM task of PPL on the Pile dataset.

| Model | 1.2M Bytes | 2.3M Bytes | 4.7M Bytes | 9.4M Bytes |
|---|---|---|---|---|
| StreamingLLM | 24.76 | 17.54 | 12.57 | 9.39 |
| Mamba2 | 8.16 | 7.91 | 7.81 | 7.61 |
| Rodimus | 7.60 | 7.47 | 7.35 | 7.33 |

### D.4 SCALING LAWS

We train all models on a subset of the Pile, using the parameter settings from GPT-3 (Brown et al., 2020). The context length is 2048 tokens. The training steps and total token counts follow the Mamba settings (Gu & Dao, 2023). Detailed configurations are provided in Table 8. For methods using purely recurrent models that combine GLU and token mixers, such as Mamba series and Rodimus, the number of layers should be doubled to match the GPT-3 specifications.

It is worth noting that In Rodimus+, we replace the second Rodimus block in each layer with SW-SKA and FFN, but the overall layer count remains unchanged. To clarify the parameter count: the Rodimus Block consists of $6d^2$ parameters, where $d$ is the model dimension. Since each layer contains two Rodimus blocks, the total parameter count for Rodimus sums to $12d^2$. In contrast, the Rodimus+ Block consists of $13d^2$ parameters, comprising one Rodimus block ($6d^2$), alongside multi-head projection, multi-query projection, and output projection in the attention mechanism ($3d^2$), and the FFN ($4d^2$).

### D.5 DOWNSTREAM EVALUATION

Our default training settings align with those in Table 8 of the scaling law experiment. Unless otherwise specified, these settings remain unchanged. All our models use mixed precision and

Table 8: Model parameters and training settings used in our scaling experiments. "Steps" indicates the number of training steps, and "Batch Size" refers to the number of tokens used for each update.

| Params | n_layer | d_model | n_heads/d_head | Steps | Learing Rate | Batch Size | Tokens |
|---|---|---|---|---|---|---|---|
| 125M | 12 | 768 | 12/64 | 4800 | 6e-4 | 0.5M | 2.5B |
| 350M | 24 | 1024 | 16/64 | 13500 | 3e-4 | 0.5M | 7B |
| 760M | 24 | 1536 | 16/96 | 29000 | 2.5e-4 | 0.5M | 15B |
| 1.3B | 24 | 2048 | 32/64 | 50000 | 2e-4 | 0.5M | 26B |

FSDP (Zhao et al., 2023) to improve training speed. To maximize GPU memory usage, we adjust the batch size according to Table 9 and modify the maximum and minimum learning rates.

The Rodimus Tokenizer uses Hugging Face's byte-level BPE algorithm, featuring a vocabulary of 126,340 tokens, including special tokens, trained on a 60 billion token subset of the curated dataset.

Notably, we do not apply weight decay to the biases of linear projections and LayerNorm, nor to the weights of LayerNorm or RMSNorm. Additionally, we set the head size of SW-SKA in Rodimus+ to 128 and the window size to half the training context, as described in Samba (Ren et al., 2024).

Table 9: The extra training settings of Rodimus* in downstream task.

| Params (B) | Tokenizer | Training context | Batch size | Max learning rate | Min learning rate |
|---|---|---|---|---|---|
| 0.13 | GPT-NeoX | 2048 | 1.5M | 3e-3 | 1e-5 |
| 0.46 | Rodimus | 2048 | 0.8M | 1.5e-3 | 1e-5 |
| 1.4 | Rodimus | 2048 | 6.3M | 1e-3 | 1e-5 |

### D.6 Fair Comparison of Larger Scale Models on Downstream Tasks

In the first block of Table 2, we noted that the model size was relatively small, consisting of only 130M parameters. To further examine the performance of Rodimus in comparison to Mamba2, we conducted an experiment using larger models. Specifically, we trained both Rodimus and Mamba2 with around 1.4B parameters from scratch under identical conditions, utilizing 100B tokens from the Pile dataset. We follow the configurations outlined in the last row of Table 8. The results are presented in Table 10, where it is evident that Rodimus, trained using the same dataset and tokenizer as Mamba2, consistently outperforms Mamba2. This performance enhancement can be attributed to the innovative design of the DDTS.

Table 10: More Results of Downstream Tasks via Zero-Shot Evaluation.

| Model | Params (B) | Tokens (B) | Tokenizer | ARC-c acc_norm ↑ | ARC-e acc ↑ | HS acc_norm ↑ | LMB ppl ↓ | LMB acc ↑ | OBQA acc_norm ↑ | PIQA acc ↑ | WG acc ↑ | AVG acc* ↑ |
|---|---|---|---|---|---|---|---|---|---|---|---|---|
| Mamba[£] | 0.13 | 100 | NeoX | 23.72 | **46.84** | 33.37 | 19.72 | 40.33 | 29.80 | 65.07 | **52.17** | 41.61 |
| Mamba2[£] | 0.13 | 100 | NeoX | **23.81** | 45.50 | 34.13 | 20.42 | 39.55 | 29.20 | 64.47 | 51.85 | 41.22 |
| Rodimus[£] | 0.13 | 100 | NeoX | 23.38 | **46.84** | 34.22 | 17.95 | 42.44 | 30.00 | 65.40 | 51.46 | 41.96 |
| Mamba2[£] | 1.4 | 100 | NeoX | 26.54 | 56.06 | 48.74 | 8.22 | 55.25 | 33.40 | 70.57 | **54.14** | 49.24 |
| Rodimus[£] | 1.4 | 100 | NeoX | **28.75** | **56.61** | **49.51** | **7.78** | **56.05** | **34.00** | **70.78** | 52.41 | **49.73** |

- Here, **bold**, underline, wavy underline indicates the best, second, and third best result, respectively. For the column "Tokens", underline indicates models using fewer tokens. The superscript [£] denotes models trained from scratch using the same configuration for fair comparison.

### D.7 MQAR (Multi-Query Associative Recall)

The MQAR task (Arora et al., 2024a) is a benchmark widely used to assess the associative recall capabilities of recurrent models. Note that quadratic softmax attention based models, such as the vanilla Transformer, achieves perfect scores across all configurations, and so these models are excluded from detailed analysis (Yang et al., 2024b; Arora et al., 2024a). In this task, input sequences

consist of several key-value pairs followed by queries. When presented with a query, the model must retrieve the corresponding key-value pair from earlier in the sequence in order to accurately predict the next token.

For example, consider the input "A 3 B 2 C 1". The key-value pairs are "A 3", "B 2", and "C 1". Later, when "A ?", "B ?", and "C ?" appear as queries, the correct outputs are "3", "2", and "1", respectively.

Notably, models utilizing full softmax attention tend to achieve perfect scores on this benchmark. Consequently, our study concentrates exclusively on recurrent models, specifically Mamba (Gu & Dao, 2023), Mamba2 (Dao & Gu, 2024), RWKV (Peng et al., 2023), and RWKV6 (Peng et al., 2024).

The experiment in Figure 6 follows the settings from MQAR (Arora et al., 2024a). We use sequence lengths of 128, 256, and 512, with corresponding key-value pairs of 8, 16, and 32. The total vocabulary size is set to 8192, the number of model layers to 2, and the model dimension $d$ varies as 64, 128, 256, and 512. Learning rates of $10^{-4}$, $10^{-3.3}$, $10^{-2.7}$ and $10^{-2}$ are applied to train for 64 epochs using the cosine annealing learning rate schedule, with the best accuracy recorded as the reported accuracy for each model dimension. In Figure 1b, the sequence length is increased to 1024, the key-value pairs are set to 256, and the model dimension $d$ is fixed at 256. Only the expansion factor $n$ is varied. Learning rates of $10^{-2}$, $10^{-2.5}$ and $10^{-3.5}$ are used to train for 32 epochs with the cosine annealing learning rate schedule, and the best accuracy is recorded as the reported accuracy corresponding to the expansion factor.

We first depict the accuracy as a function of the expansion factor $n$ in Figure 1b. In this analysis, we focus on the Mamba series for comparison, as their architecture closely resembles that of Rodimus*, allowing for consistent settings across all models. Our results indicate that Rodimus* exhibits superior recall ability at the same expansion factor $n$. This enhancement can be attributed to the model's DDTS mechanism, which filters out irrelevant past information and current inputs more effectively.

Next, we present the accuracy as a function of model dimension $d$ in Figure 6, where the expansion factor $n$ is maintained at its original value across different models. Once again, Rodimus* achieves the highest performance. In comparison, Mamba and RWKV underperform, particularly when the model dimension is small. This suggests that the fixed-capacity state of these models can only deliver reliable recall results when the state size is sufficiently large to retain relevant past information.

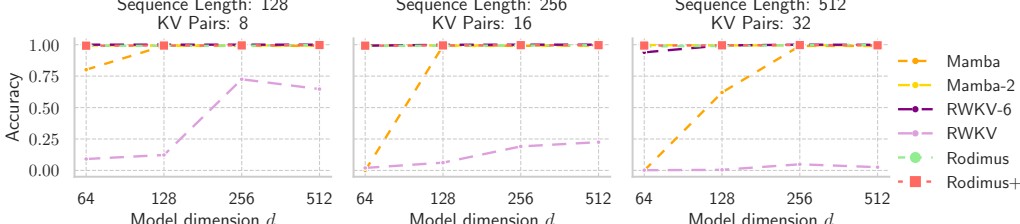

Figure 6: Recall accuracy of MQAR Task at different sequence lengths.

## D.8 NEEDLEBENCH

The NeedleBench (Li et al., 2024) is a retrieval evaluation method that randomly inserts key information into long texts to create prompts for LLMs. The primary objective of this test is to assess the ability of LLMs to effectively extract crucial information from extensive documents, thereby evaluating their proficiency in processing and comprehending long-form content. The NeedleBench comprises several tasks: the Single-Needle Retrieval Task (S-RT), the Multi-Needle Retrieval Task (M-RT), the Multi-Needle Reasoning Task (M-RS), and the Ancestry Trace Challenge (ATC). The S-RT focuses on the precise extraction of specific details, while the M-RT evaluates the retrieval of multiple related pieces of information, simulating complex queries. The M-RS task involves synthesizing and understanding interconnected information, and the ATC examines logical reasoning, requiring comprehensive engagement with all provided content to solve intricate problems.

In this study, we evaluate Mamba-1.4B (Gu & Dao, 2023), Mamba2-1.3B (Dao & Gu, 2024), Pythia-1.4B (Biderman et al., 2023), Rodimus-1.4B, and Rodimus+-1.6B on NeedleBench using `OpenCompass` [2] (Contributors, 2023), since the sequence length of all these models is 2048 tokens.

---
[2] https://github.com/open-compass/opencompass

The brief evaluation results, featuring Mamba2-1.3B, Pythia-1.4B, Rodimus-1.4B, and Rodimus+-1.6B, are presented in Figure 5. Full evaluation results, including subtasks, are detailed in Figures 9 (Mamba-1.4B), 10 (Mamba2-1.3B), 11 (Rodimus-1.4B), 12 (Pythia-1.4B), and 13 (Rodimus+-1.6B). The "Overall Score" represents the average result of these subtasks. Additionally, we present the NeedleBench results for Rodimus+ during training in Appendix E.3, where the model has trained on the 4K-length context with 2.5T tokens, and demonstrates superior long-distance retrieval capabilities, as shown in Figure 14.

Many studies have suggested that pre-trained recurrent models tend to underperform on challenging recall-intensive tasks like NeedleBench when compared to models utilizing full attention mechanisms (Fu et al., 2023; Waleffe et al., 2024). This trend is illustrated by comparisons between Pythia and Mamba2. Nevertheless, Rodimus demonstrates superior recall ability relative to Pythia. While Pythia shines in scenarios involving shorter contexts (0k-4k), Rodimus exhibits better extrapolation capabilities for longer contexts, leading to overall enhanced performance. Furthermore, Rodimus+, a hybrid model, further improves recall ability by integrating global semantic-level and local token-level representations. The results for Rodimus and Rodimus+ indicate that the Rodimus family effectively navigates the accuracy-efficiency trade-off, particularly in terms of enhancing recall ability. In addition, training with longer contexts can effectively enhance the model's recall ability in Needlebench. Rodimus+ trained with the 4K-length context performs better on Needlebench than when trained with the standard 2K-length context.

### D.9 MORE RECALL-INTENSIVE BENCHMARKS OF PRE-TRAINED LANGUAGE MODELS

The NeedleBench benchmark may be excessively demanding for small pre-trained models with a context length limitation of 2K tokens. To mitigate this issue, we introduce several additional, simpler recall benchmarks that are outlined below:

- **FDA**: This benchmark involves extracting specific details, such as device codes, classifications, and intended uses, from medical device reports submitted to the FDA.
- **SWDE**: This task focuses on retrieving particular information, such as movie directors and university tuition, from HTML pages across 14 websites in domains like Movies and Universities.
- **SQuADv2**: A question-answering task where models identify answer spans within Wikipedia articles based on given queries.
- **TriviaQA**: This benchmark challenges models to answer questions using information from Wikipedia and other broad web sources, featuring diverse question formats.
- **NQ**: Utilizing real Google search queries, this task requires models to find answers within Wikipedia, necessitating the extraction of relevant text snippets.
- **DROP**: A reasoning-intensive question-answering task where models must carry out operations—including counting, sorting, or performing arithmetic—using information derived from Wikipedia passages.

The same checkpoints used in the NeedleBench section are employed for evaluation, with the accuracy results presented in Table 11. The average scores for both Rodimus and Rodimus+ outperform those of Mamba and Mamba2. However, contrary to the trends observed in NeedleBench, Rodimus demonstrates a slight performance lag behind Pythia. This finding supports our earlier analysis in Appendix D.8, which indicates that the softmax attention-based model Pythia surpasses recurrent models in recall tasks within the non-extrapolation interval. Notably, the hybrid model Rodimus+, which integrates SW-SKA, successfully exceeds Pythia's performance, highlighting that the incorporation of this hybrid model effectively addresses the performance limitations of recurrent models.

### D.10 ABLATION STUDIES DETAILS

In this section, we present ablation studies designed to achieve four primary objectives: (i) assessing the contributions of various components—such as the gates and the two-hop residual—in Rodimus*, (ii) evaluating performance with different head compression methods, and (iii) determining the optimal hyperparameter values ($\ell$ and $n$), (iv) analyzing the ordering of the Rodimus block and SW-SKA plus FFN in Rodimus+. We begin with experiments conducted on the WikiText-103 dataset, chosen for its relatively small size, which allows for quicker results and makes the impact

Table 11: Recall results of models via zero-shot evaluation measured by accuracy. The input context length is set to 1024, and the maximum generation length is 512.

| Model | SWDE | FDA | SQUAD | Drop | NQ | TriviaQA | AVG |
|---|---|---|---|---|---|---|---|
| Pythia | 53.05 | 81.65 | 41.89 | 21.85 | 34.49 | 58.18 | 48.52 |
| Mamba | 45.36 | 59.85 | 41.28 | 20.84 | 34.43 | 60.43 | 43.70 |
| Mamba2 | 52.67 | 71.12 | 42.19 | 22.86 | 35.79 | 63.57 | 48.03 |
| Rodimus | 49.77 | 79.47 | 43.10 | 22.76 | 32.56 | 61.85 | 48.25 |
| Rodimus+ | 58.76 | 80.84 | 48.74 | 24.58 | 36.27 | 64.63 | 52.30 |

of removing specific components easily observable. The second set of experiments utilizes a 7B subset of Pile, providing a larger dataset to discern subtle differences arising from hyperparameter adjustments. Unless otherwise specified, the models trained on WikiText-103 and Pile follow the training settings detailed in the second and third columns of Table 5, respectively. Regarding model configurations, for the models trained on WikiText-103, the number of layers and dimensions are set to 6 and 512, respectively. For the models trained on Pile, the various model configurations are provided in Table 8. For the WikiText-103 dataset, we report both the test PPL and the best test PPL. The test PPL refers to the evaluation of the test set after training on the training dataset for a fixed number of epochs, while the best test PPL reflects the lowest value achieved during early stopping of training. Although the former approach is commonly employed in the literature, it does not always accurately capture model performance.

**The necessity of $g_t$, $\tau_t$, and $\hat{\beta}_t$**: As presented in Table 15, the removal of these gates results in decreased performance, indicating their essential role in DDTS.

**The necessity of $\beta_t$**: Table 15 shows that incorporating $\hat{\beta}_t$ does not reduce the PPL further; in fact, it tends to increase it slightly. As shown in Table 16, this finding suggests that excluding $\beta_t$, is preferable, as it does not provide performance improvements but may complicate the training process and hinder the training efficiency (see detailed explanations Section 3.1.1).

**The necessity of the two-hop residual**: According to Table 17, the two-hop residual in Rodimus+ can slightly enhance performance when the model deepens with more layers. This mechanism aids in stabilizing training, as noted by Ma et al. (2024). Given that modern Large Language Models (LLMs) typically consist of many layers and the two-hop residual incurs no extra computational cost, we implement it in Rodimus+.

**Comparison between SKA, GQA, and MHA**: In this experiment, the model settings refer to 125M item in the Table 8. Table 14 indicates that SKA consistently outperforms GQA in terms of PPL across both Rodimus+ or Transformer++, demonstrating the effectiveness of SKA. However, SKA performs less optimally than MHA, likely due to increased training difficulty, requiring more iterations or data for comparable performance. Nevertheless, since SKA is more parameter-efficient than MHA and surpasses GQA in performance, we opt to utilize SKA in Rodimus+. Additionally, we provide an analysis of the memory footprint of the cache, excluding the model weights. SKA demonstrates a substantial reduction in memory footprint compared to MHA, although its cache size is slightly larger than that of GQA.

**Sensitivity to $\ell$ and $n$**: Table 12 reveals that increasing the rank $\ell$ of the weight matrices in Eq. (10) generally improves performance but also raises the parameter count. We find that setting $\ell = 16$ effectively balances performance and parameter quantity, and thus we adopt this value for our experiments. Regarding the state expansion factor $n$, shown in Table 13, we observe that valid PPL decreases as $n$ increases; however, the rate of decrease slows significantly for $n \geq 64$. Consequently, we establish $n = 64$ for our experiments. Note that $n = 128$ in Mamba2. Thus, Rodimus outperforms Mamba2 in most tests with only half the state size.

**The ordering of the Rodimus block and SW-SKA plus FFN**: We investigate the impact of the ordering of the Rodimus Block, SW-SKA, and FFN on model performance. The results are detailed in Table 18. Specifically, for the Pile subset, the model settings correspond to 350 million items, as referenced in Table 8. Since SW-SKA and FFN are integrated within a two-residual hop structure,

we treat them as a single unit and focus on the relative positioning of the Rodimus Block with respect to SW-SKA and FFN. The term "Rodimus+X" denotes the original order presented in Eq. (13), while "X+Rodimus" indicates the rearrangement of Rodimus's position relative to SW-SKA and FFN in the same equation.

The findings in Table 18 reveal that for shallow models tested on the WikiText-103 dataset, it is essential to place the Rodimus Block before SW-SKA and FFN. In contrast, the results from the Pile dataset suggest that as the number of layers increases, the significance of the Rodimus Block's positioning diminishes. We hypothesize that this occurs because an increase in layers lessens the impact of module types that are closer to the word embedding and output projection on overall model performance.

Table 12: Results of the ablation experiments on the low-rank term.

| $\ell$ | Valid PPL | Params (M) | Valid PPL | Params (M) |
|---|---|---|---|---|
| 1 | 7.41 | 126.3 | 5.84 | 360.8 |
| 2 | 7.32 | 126.4 | 5.81 | 361.0 |
| 4 | 7.36 | 126.5 | 5.77 | 361.4 |
| 8 | 7.30 | 126.8 | 5.76 | 362.2 |
| 16 | 7.26 | 127.4 | 5.75 | 363.7 |
| 32 | 7.31 | 128.6 | 5.74 | 366.9 |
| 64 | 7.24 | 130.9 | 5.71 | 373.2 |
| 128 | 7.17 | 135.7 | 5.70 | 385.8 |

Table 13: Results of ablation experiments on expansion factor $n$.

| $n$ | Valid PPL | Params (M) | Valid PPL | Params (M) |
|---|---|---|---|---|
| 16 | 5.75 | 363.7 | 7.26 | 127.4 |
| 32 | 5.66 | 370.0 | 7.13 | 129.8 |
| 64 | 5.62 | 382.6 | 7.01 | 134.5 |
| 128 | 5.58 | 407.8 | 6.95 | 143.9 |
| 256 | 5.57 | 458.1 | 6.9 | 162.8 |

Table 14: Results of ablation experiment of MHA, GQA and SKA. For the Memory Footprint, we log only the cache usage, excluding the contribution of model weights.

| Model | Head Type | Valid PPL | Memory Footprint (M Bytes) |
|---|---|---|---|
| Transformer++ | MHA | 7.25 | 75.5 |
| Transformer++ | GQA | 7.38 | 25.2 |
| Transformer++ | SKA | 7.33 | 44.0 |
| Rodimus+ | MHA | 7.09 | 40.1 |
| Rodimus+ | GQA | 7.19 | 14.9 |
| Rodimus+ | SKA | 7.15 | 24.4 |

Table 15: Results of ablation experiments on gates.

| $\hat{\beta}_t$ | $g_t$ | $\tau_t$ | Test PPL | Best Test PPL |
|---|---|---|---|---|
| ✗ | ✗ | ✗ | 23.23 | 22.66 |
| ✓ | ✗ | ✗ | 23.24 | 22.20 |
| ✗ | ✓ | ✗ | 22.34 | 21.33 |
| ✗ | ✓ | ✓ | 22.14 | 21.15 |
| ✓ | ✓ | ✗ | 22.24 | 21.17 |
| ✓ | ✓ | ✓ | **21.90** | **20.84** |

Table 16: Results of ablation experiment on type of $\beta$.

| $\beta_t =?$ | Test PPL | Best Test PPL |
|---|---|---|
| $\prod_{j=i+1}^{t} \beta_j$ | 24.36 | 23.32 |
| $\mathbf{1}_m$ | 23.24 | 22.20 |

Table 17: Results of ablation experiment on models with or without two-hop residual.

| $L$ | THR | Test PPL | Best Test PPL |
|---|---|---|---|
| 6 | ✓ | 21.56 | 20.46 |
| 6 | ✗ | 21.77 | 20.37 |
| 24 | ✓ | 21.50 | 20.85 |
| 24 | ✗ | 21.52 | 20.95 |

"THR" denotes the two-hop residual, and $L$ represents the number of layers.

# E  MORE EXPERIMENTAL RESULTS

## E.1  THROUGHPUTS, FLOPS, AND MEMORY

In this section, we present the FLOPs and throughputs (words per second, wps) of Rodimus*, Mamba2, and the Transformer model, both during training and inference. Additionally, we provide an analysis of memory usage during inference, excluding model weights to ensure a fair evaluation of space complexity. The specific configurations examined correspond to the 1.3B settings outlined in Table 8. For the Transformer model, we evaluate its attention mechanism implemented in both PyTorch (denoted as "torch") and Flash Attention 2 (denoted as "F.A."). The PyTorch implementation adheres to the theoretical time and space complexity of softmax attention, while Flash Attention significantly optimizes computation speed and reduces memory usage. Our analysis of FLOPs and throughputs focuses on scenarios with a batch size of one, with results detailed in Tables 19 and 20.

Table 18: Results of ablation experiment on the ordering of Rodimus, SW-SKA, and FFN on WikiText-103 and Pile.

| Configuration | WikiText-103 PPL n_layer=6 | Pile PPL n_layer=24 |
|---|---|---|
| Rodimus + X | 21.56 | 5.9 |
| X + Rodimus | 21.95 | 5.9 |

The results reveal that the FLOPs and memory footprint of Rodimus are the smallest among all models tested, indicating the time and space efficiency of the proposed architecture. Particularly, Rodimus consumes only half the memory of Mamba2, making it an attractive option for resource-constrained environments, such as edge devices or low-power systems. Conversely, Rodimus+ exhibits a slight increase in FLOPs but a significant rise in memory consumption due to the integration of SW-SKA and FFN components. Overall, both Rodimus and Rodimus+ demonstrate greater time and space efficiency than the Transformer model, with Rodimus being more efficient than Mamba2 as well. Notably, Rodimus, Rodimus+, and Mamba2 can manage long sequences (e.g., pre-filling 24K tokens) more effectively than Transformers.

However, it is noteworthy that the throughputs of Mamba2 and the Transformer (F.A.) is significantly higher than that of Rodimus*. This discrepancy arises because Rodimus* does not incorporate the I/O-aware optimizations that are implemented in these baselines. In future work, we aim to address these throughputs limitations by integrating I/O-aware optimizations to enhance Rodimus*'s efficiency during both training and inference.

Table 19: Throughputs and FLOPs during training.

| Model | 4K-length contexts | |
|---|---|---|
| | FLOPs $\times 10^{13}$ | Throughputs (wps) |
| Transformer (torch) | 4.46 | 2726.64 |
| Transformer (F.A.) | - | 8547.29 |
| Mamba2 | 3.65 | 10147.10 |
| Rodimus | 3.60 | 8592.59 |
| Rodimus+ | 4.29 | 9410.38 |

Table 20: Throughputs, FLOPs, and Memory Footprint during inference.

| Model | 4K-length contexts | | | 24K-length contexts | | |
|---|---|---|---|---|---|---|
| | FLOPs $\times 10^9$ | Throughputs (wps) | Memory Footprint (M Bytes) | FLOPs $\times 10^9$ | Throughputs (wps) | Memory Footprint (M Bytes) |
| Transformer (torch) | 3.6 | 2.4 | 805 | 7.7 | OOM | 4831.8 |
| Transformer (F.A.) | - | 49.4 | - | - | 11.0 | - |
| Mamba2 | 3.0 | 59.3 | 0.8 | 3.0 | 59.3 | 0.8 |
| Rodimus | 2.9 | 21.4 | 0.4 | 2.9 | 21.4 | 0.4 |
| Rodimus+ | 3.5 | 19.6 | 207.8 | 3.5 | 19.6 | 207.8 |

### E.2 GATES WITH TEMPERATURE $\tau_t$

We further visualize the effect of gates influenced by the temperature gate $\tau_t$. In Figure 7, we present the curves of the decay term $\exp(\boldsymbol{g}_t \odot \boldsymbol{\tau}_t)$ and the input term $\boldsymbol{g}_t^{\tau_t}$ as the temperature varies. As $\boldsymbol{\tau}_t$ decreases, the lower bounds of both terms gradually rise, enhancing information retention.

In Figure 8, we compare the gates of a layer in models from Table 15 with and without the temperature gate $\tau_t$. By comparing Figure 8a with Figure 8b, and Figure 8c with Figure 8d, we observe that the inclusion of temperature increases the flexibility of the gates in selecting information.

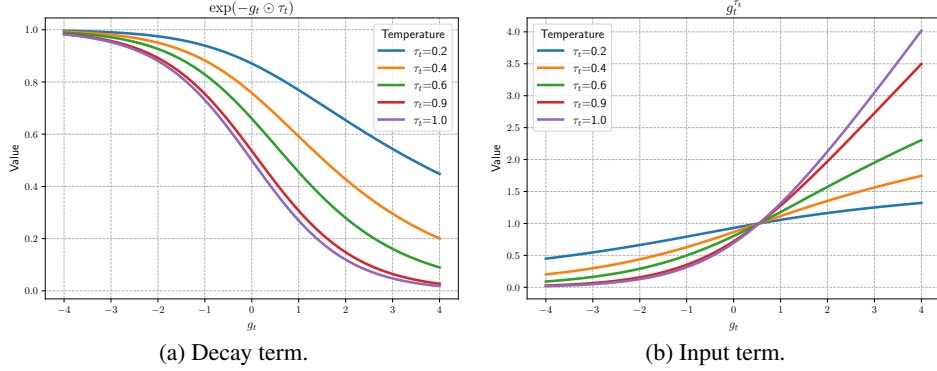

(a) Decay term.                    (b) Input term.

Figure 7: The curves of the gates with temperature gate $\boldsymbol{\tau}_t$.

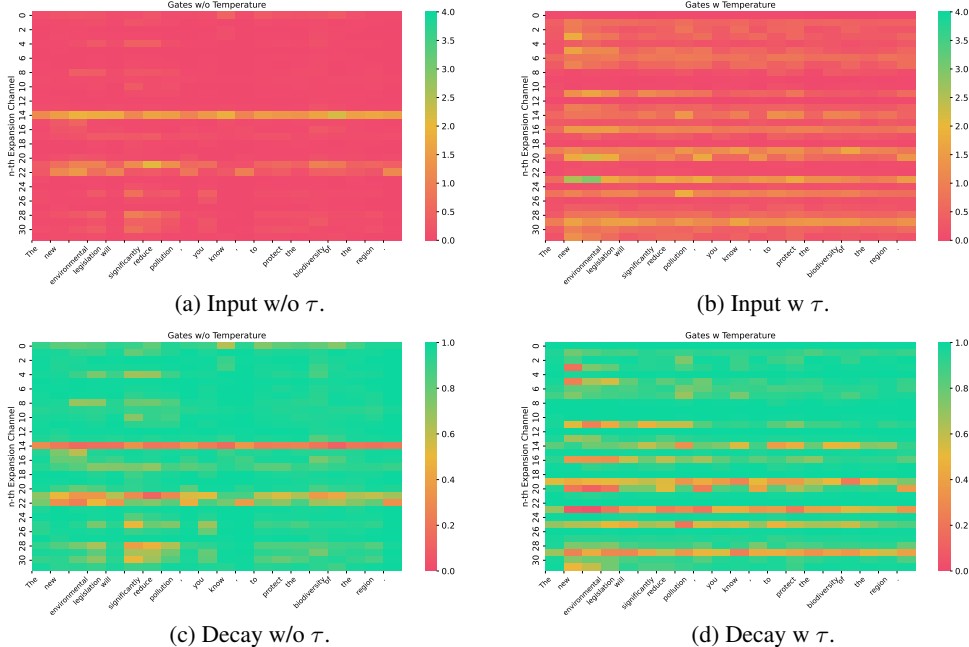

(a) Input w/o $\tau$.                    (b) Input w $\tau$.

(c) Decay w/o $\tau$.                    (d) Decay w $\tau$.

Figure 8: The visualization results of the gates are shown. The y-axis represents the n-th dimension of the gates, and the x-axis displays the sub-words after tokenization. (a) and (b) show the heatmap visualizations of the input gate $\boldsymbol{g}_t$ and $\boldsymbol{g}_t^{\tau_t}$. (c) and (d) present the corresponding results of $\exp(\boldsymbol{g}_t)$ and $\exp(\boldsymbol{g}_t \odot \boldsymbol{\tau}_t)$.

### E.3 MORE PRACTICAL LLM: MULTI-STAGE TRAINING OF RODIMUS+

To evaluate the practical capabilities of Rodimus+ and develop it into a functional LLM, we develop Rodimus+-Coder, a lightweight code generation model available in 1.6B and 4B parameter versions. Both variants have achieved SOTA performance in their respective size. The training process for both the 1.6B and 4B models begins from scratch and follows a three-stage pre-training approach:

- **Initial:** Training on 6T tokens of primarily natural language data (20% code).
- **Coding:** Continued training on 1.5T tokens of code-focused data (60% code).
- **Annealing:** Annealing with 500B tokens (more than 50% synthetic data).

Table 21: Comprehensive evaluation results on the base models.

| Datasets | Qwen2.5-Coder-1.5B | Rodimus+-Coder-1.6B | Gemma2-2B-PT | Qwen2.5-Coder-3B | Rodimus+-Coder-4B | Gemma3-4B-PT | Qwen2.5-Coder-7B |
|---|---|---|---|---|---|---|---|
| *Coding Tasks* | | | | | | | |
| HumanEval | 41.5 | 51.2 | 19.5 | 51.8 | **60.4** | 36.0 | **60.4** |
| HumanEval+ | 34.8 | 45.1 | - | 40.9 | **52.4** | - | 50.6 |
| MBPP | 57.2 | 51.2 | 31.0 | 62.6 | 64.6 | 46.0 | **70.0** |
| MBPP+ | 66.1 | 62.2 | - | 65.9 | **71.4** | - | 70.1 |
| BCB$_{COMPLETION}$ | 21.6 | 17.9 | - | 26.2 | **30.8** | - | 30.4 |
| MultiPL-E | 46.1 | 52.5 | - | 49.4 | **60.7** | - | 56.9 |
| CRUXEval | 38.5 | 45.1 | - | 44.6 | 56.4 | - | **56.8** |
| **Coding Avg.** | 43.7 | 46.5 | - | 48.8 | **56.7** | - | 56.4 |
| *General Tasks* | | | | | | | |
| C-EVAL | 55.2 | 56.7 | - | 65.3 | 70.2 | - | 69.1 |
| CMMLU | 54.5 | 52.3 | - | 65.4 | 68.3 | - | 72.7 |
| MMLU | 55.5 | 51.1 | 52.2 | 63.3 | 62.6 | 59.6 | 70.5 |
| BBH | 21.8 | 46.8 | 42.4 | 32.5 | 61.9 | 50.9 | 67.3 |
| **General Avg.** | 46.8 | 51.7 | - | 56.6 | 65.8 | - | 69.9 |
| *Mathematics Tasks* | | | | | | | |
| GSM8K | 60.4 | 68.7 | 25.0 | 72.1 | 78.5 | 38.4 | 83.4 |
| MATH | 23.7 | 29.0 | 16.4 | 31.9 | 37.0 | 24.2 | 42.2 |
| **Mathematics Avg.** | 41.9 | 48.9 | 20.7 | 52.0 | 57.8 | 31.3 | 62.8 |
| **Overall** | 44.4 | 48.4 | - | 51.7 | **59.6** | - | **61.6** |

The total pre-training dataset comprises about 8T tokens. The post training pipeline includes Supervised Fine-Tuning (SFT) and Direct Preference Optimization (DPO), utilizing the same dataset as for Ling-Coder-lite (Codefuse & Team, 2025). The model weights have been open-sourced [3].

We conduct a comprehensive evaluation of Rodimus+-Coder models on 19 diverse benchmarks, including 13 code-related tasks (HumanEval (Chen et al., 2021), HumanEval+ (Liu et al., 2023), MBPP (Austin et al., 2021), MBPP+ (Liu et al., 2023), LiveCodeBench Jain et al. (2024), Big-CodeBench (Zhuo et al., 2024), MultiPL-E (Cassano et al., 2022), MBXP-PLUS (Athiwaratkun et al., 2022), CRUXEval (Gu et al., 2024), HumanEvalFix (Muennighoff et al., 2023),Spider (Yu et al., 2019)), 4 general knowledge assessments (C-Eval (5-shot) (Huang et al., 2023), CMMLU (5-shot) (Li et al., 2023), MMLU (5-shot) (Hendrycks et al., 2021a), and BBH (3-shot) (Suzgun et al., 2023)), and 2 mathematical reasoning challenges (GSM8K (4-shot) (Cobbe et al., 2021) and MATH (4-shot) (Hendrycks et al., 2021b)).

We systematically compare performance against various scales of contemporary models, including Qwen2.5-Coder (Hui et al., 2024), Gemma (Team et al., 2024; 2025), and Phi-4-Mini (Abouelenin et al., 2025), to establish a comprehensive analysis of coding proficiency, general knowledge comprehension, and mathematical problem-solving capabilities. To ensure direct comparability, we re-evaluate Qwen series models using the same methodology as Rodimus+-Coder. Metrics for other models (Gemma, Phi-4-Mini) are taken from their original publications. All evaluation scripts are publicly available in the CodeFuse-Evaluation repository [4] for transparency and reproducibility.

### E.3.1 PERFORMANCE ON BASE MODEL

Overall assessment demonstrates that Rodimus+-Coder base models outperform similarly sized Qwen2.5-Coder and Gemma models across benchmarks. In particular, Rodimus+-Coder-4B achieves a comparable average coding performance to Qwen2.5-Coder-7B, although it has fewer parameters. Furthermore, even after extensive code-specific training, Rodimus+-Coder maintains competitive performance in general natural language understanding and mathematical reasoning tasks, indicating effective preservation of its general capabilities alongside domain-specific enhancement.

---

[3] https://huggingface.co/collections/codefuse-ai
[4] https://github.com/codefuse-ai/codefuse-evaluation

Table 22: Comprehensive evaluation results on the Chat model.

| Datasets | Qwen2.5-Coder-1.5B-Instruct | Rodimus+-Coder-1.6B-Chat | Gemma2-2B-IT | Qwen2.5-Coder-3B-Instruct | Phi-4-Mini-3.8B | Rodimus+-Coder-4B-Chat | Gemma3-4B-IT | Qwen2.5-Coder-7B-Instruct |
|---|---|---|---|---|---|---|---|---|
| *Coding Tasks* | | | | | | | | |
| HumanEval | 64.6 | 76.8 | 20.1 | 79.9 | 74.4 | 86.6 | 71.3 | 87.2 |
| HumanEval+ | 63.4 | 73.8 | - | 80.5 | 68.3 | 82.9 | - | 82.3 |
| MBPP | 51.0 | 59.0 | 36.6 | 59.2 | 65.3 | 68.0 | 63.2 | 75.8 |
| MBPP+ | 53.0 | 66.4 | - | 61.9 | 63.8 | 68.5 | - | 75.1 |
| LCB$_{(24.08-24.11)}$ | 4.0 | 10.9 | - | 13.0 | - | 13.9 | - | 22.8 |
| BCB$_{INSTRUCT}$ | 10.8 | 21.5 | - | 21.7 | 33.8 | 26.6 | - | 30.6 |
| HumanEval-Mul | 50.8 | 57.3 | - | 67.4 | - | 70.6 | - | 76.1 |
| MBPP-Mul | 43.4 | 52.4 | - | 53.4 | - | 59.6 | - | 61.4 |
| MBXP-EN | 55.8 | 75.5 | - | 76.0 | - | 87.3 | - | 87.7 |
| MBXP-CN | 48.8 | 75.0 | - | 68.7 | - | 84.3 | - | 83.5 |
| CRUXEval | 28.6 | 55.0 | - | 51.6 | - | 63.2 | - | 69.3 |
| HumanEvalFix | 38.9 | 52.6 | - | 55.5 | - | 68.8 | - | 69.3 |
| Spider | 61.2 | 71.4 | - | 71.8 | 42.2 | 73.5 | - | 82.0 |
| **Coding Avg.** | 44.2 | 57.5 | - | 58.5 | - | **65.7** | - | **69.5** |
| *General Tasks* | | | | | | | | |
| C-EVAL | 51.5 | 50.8 | - | 62.0 | - | 61.6 | - | 66.4 |
| CMMLU | 45.2 | 50.5 | - | 60.1 | - | 62.0 | - | 64.9 |
| MMLU | 52.0 | 49.3 | 56.1 | 61.7 | 67.3 | 57.5 | 58.1 | 66.1 |
| BBH | 24.2 | 58.7 | 41.4 | 57.3 | 70.4 | 63.7 | 72.2 | 59.1 |
| **General Avg.** | 43.2 | 52.3 | - | 60.3 | - | 61.2 | - | 64.1 |
| *Mathematics Tasks* | | | | | | | | |
| GSM8K | 54.4 | 68.5 | 62.6 | 73.5 | 88.6 | 79.2 | 89.2 | 79.5 |
| MATH | 38.1 | 33.5 | 27.2 | 44.1 | 64.0 | 44.1 | 75.6 | 60.8 |
| **Mathematics Avg.** | 46.2 | 51.0 | 44.9 | 58.8 | 68.8 | 61.7 | 82.4 | 70.1 |
| **Overall** | 44.2 | 55.8 | - | 58.9 | - | **64.3** | - | **68.4** |

### E.3.2 PERFORMANCE ON CHAT OR INSTRUCT MODELS.

Overall assessment reveals that Rodimus+-Coder-Chat models maintain superior performance compared to models of similar size. Remarkably, Rodimus+-Coder-1.6B-Chat and Rodimus+-Coder-4B-Chat achieve coding performance comparable to their larger counterparts, Qwen2.5-Coder-3B-Instruct and Qwen2.5-Coder-7B-Instruct respectively, demonstrating efficient parameter utilization while delivering competitive performance with larger instruction-tuned models.

## F DISCUSSION ON WIDESPREAD USAGE OF RECURRENT AND HYBRID MODELS

The technical challenges associated with our proposed models, particularly in comparison to the widespread adoption of softmax attention-based large language models (LLMs), are well-recognized. However, the evolving landscape of machine learning research suggests a shift in this dynamic. The growing interest in recurrent and hybrid models within both academic and industrial domains is beginning to address these barriers, paving the way for broader exploration and application.

Recent developments illustrate this trend, as numerous new recurrent and hybrid models have emerged. Notable examples include Mamba, Mamba2, RWKV, RWKV6, RetNet, HGRN, HGRN2, GLA, Hawk, and Griffin. Moreover, Mistral has successfully scaled Mamba to 7B parameters and has made it open-source. Similarly, Google has scaled Griffin to an impressive scale of 9B and open-sourced Recurrent-Gemma. These advancements indicate growing confidence in the performance of recurrent and hybrid models, often achieving comparable results compared to traditional softmax attention-based LLMs while maintaining lower complexity.

From an inference perspective, it is indeed worth highlighting that recurrent and hybrid models are inherently designed for efficiency. The per-token generation complexity for these models is theoretically $\mathcal{O}(1)$, in stark contrast to the $\mathcal{O}(T)$ complexity of softmax-based models, where $T$ denotes the context length. As a result, even without optimization, recurrent and hybrid models demonstrate rapid inference capabilities. Furthermore, state-of-the-art inference optimization frameworks such as vLLM, TensorRT-LLM, and OLlama have already begun to support the inference of models like Mamba and Mamba2, further paving the way for their more widespread usage.

## G  FUTURE WORK

Rodimus* has delivered excellent results across benchmarks; however, there are areas for improvement. Due to limited computing resources, we have not expanded its parameters to match open-source models like RWKV6-14B and Qwen2-72B. Additionally, Rodimus* lacks the highly I/O-aware optimization found in models such as Mamba and Mamba2. There's potential to enhance its performance by designing I/O-aware multi-head scalar decay and integrating it with Rodimus's DDTS to broaden gating mechanisms without significantly impacting training efficiency. Furthermore, for Rodimus+, the memory usage of SW-SKA can be further reduced while achieving better practical application performance. We aim to address these issues in future work.

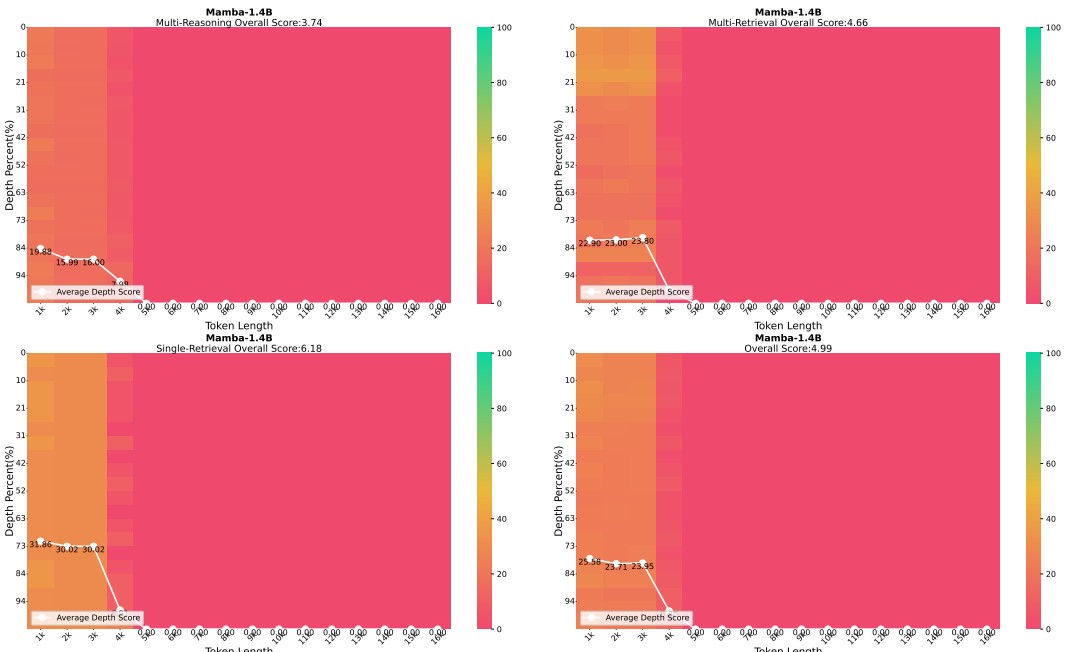

Figure 9: Mamba-1.4B

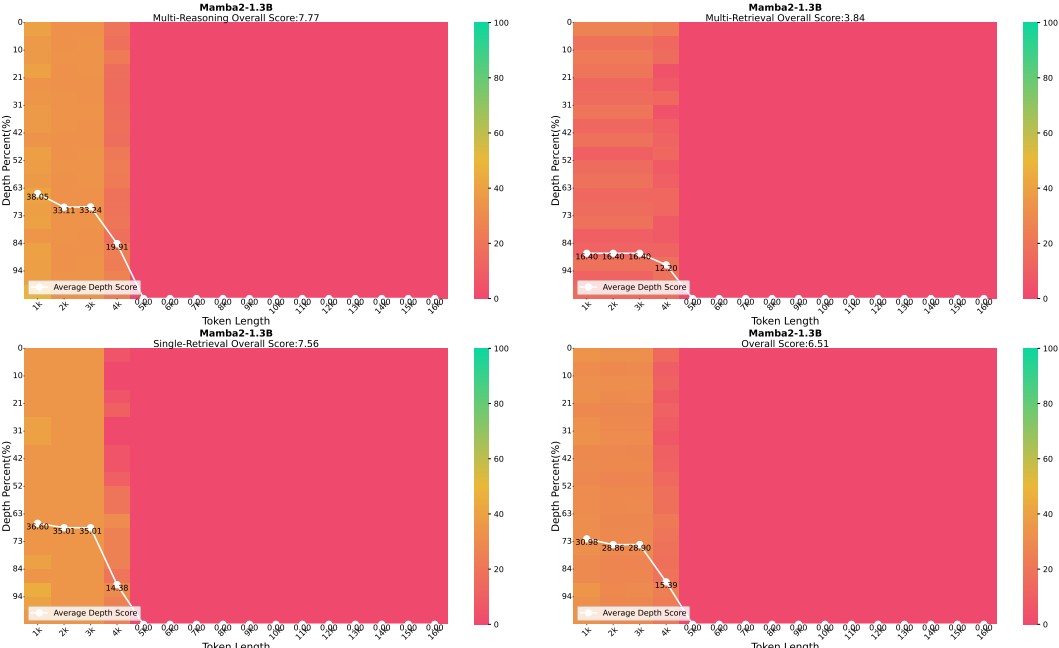

Figure 10: Mamba2-1.3B

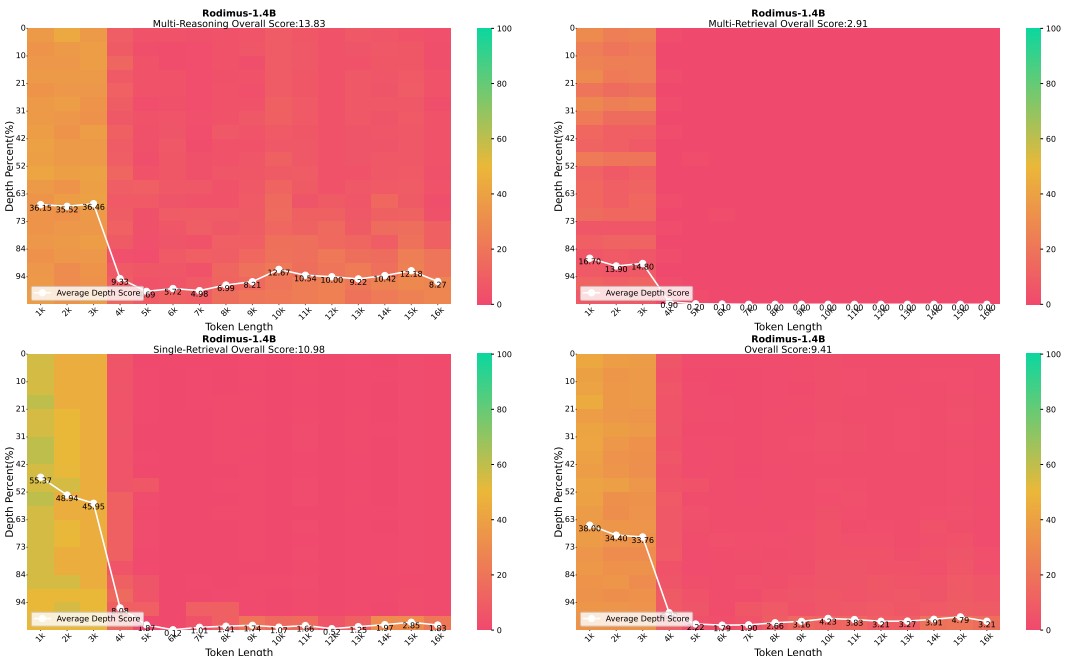

Figure 11: Rodimus-1.4B

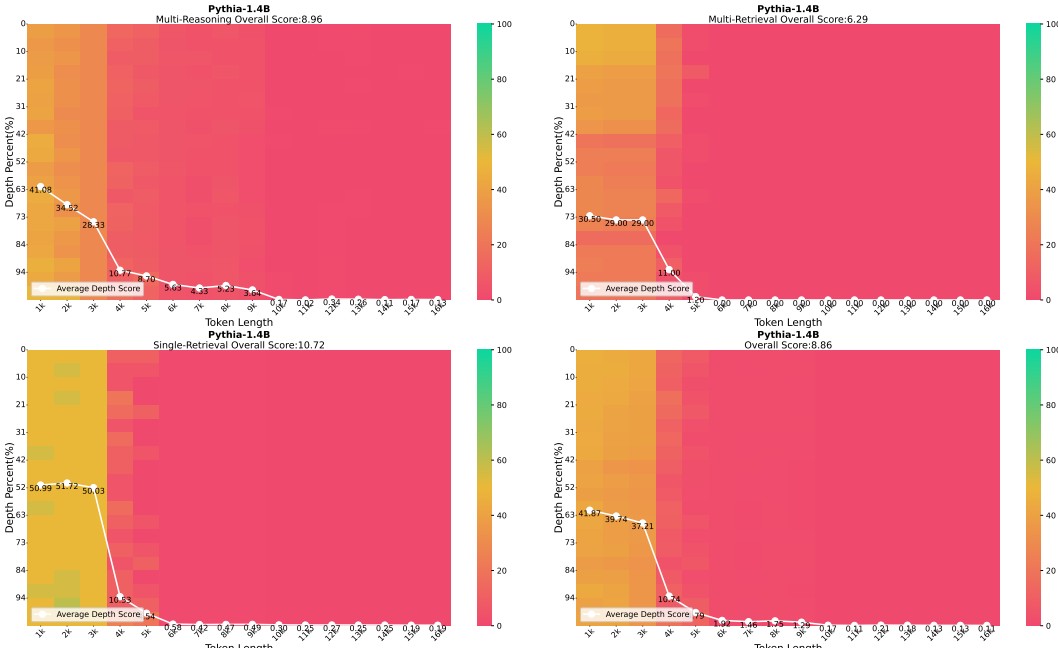

Figure 12: Pythia-1.4B

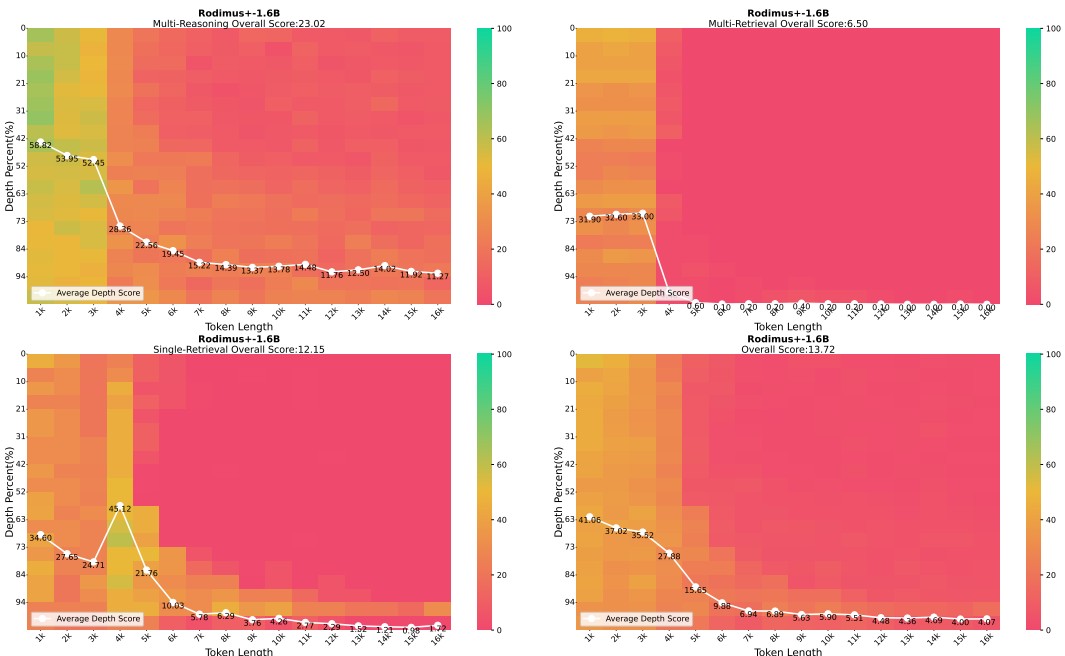

Figure 13: Rodimus+-1.6B-2K

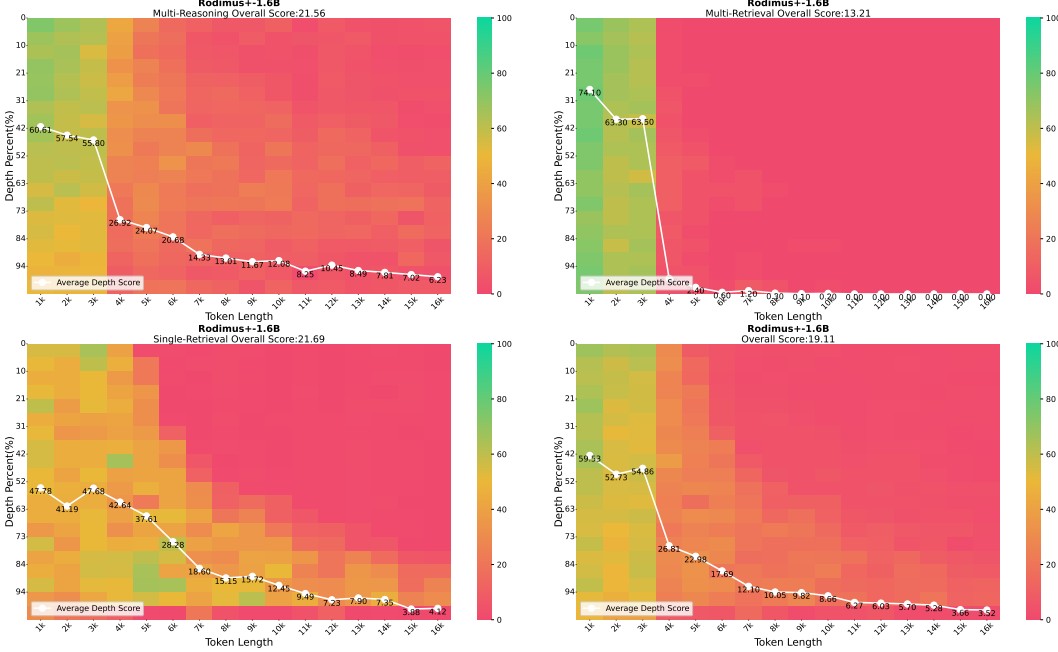

Figure 14: Rodimus+-1.6B-4K

