# OpenReview forum: "Rodimus*: Breaking the Accuracy-Efficiency Trade-Off with Efficient Attentions"
_ICLR.cc/2025/Conference — ICLR 2025 Poster_

### Official Review · Reviewer_ZtZu · 2024-10-27

**Soundness:** 3
**Presentation:** 3
**Contribution:** 2
**Rating:** 6
**Confidence:** 4

**Summary:**

The paper introduces Rodimus, a linear attention based model, purely recurrent, and and its enhanced variant Rodimus+, a hybrid model, which combines Rodimus with SW-SKA. The paper aims at reducing the computational inefficiencies of traditional softmax attention mechanisms in large language models (LLMs). Rodimus uses a linear attention approach combined with a data dependent tempered selection mechanism to achieve O(T) complexity for per-token generation. This significantly reduces memory usage while maintaining performance (a key limitation of recurrent models). Rodimus+ extends this by integrating a sliding window shared key attention (SW-SKA), which combines the strengths of semantic, token, and head compression to enhance efficiency and accuracy.

The paper is in the broader context of linear attention mechanisms such as those introduced by Katharopoulos et al. (2020), which reduce computational complexity through kernel-based approximations. Plus, it builds on the concept of sparse attention, similar to the approaches used in Longformer (Beltagy et al., 2020), but enhances it by integrating local and global context understanding. Rodimus+ also leverages ideas from Grouped-Query Attention (GQA) (Ainslie et al., 2023) by implementing shared-key attention, compressing the memory footprint while preserving performance.

The experimental results show that Rodimus+ achieves better downstream performance even against models trained on more tokens.

**Strengths:**

The proposed approach seems to achieve interesting results in complexity vs performances:
- Rodimus+ manages to achieve high performance even when trained on fewer tokens compared to other models.
- The authors provide extensive validation across multiple benchmarks, showing that Rodimus+ outperforms existing models.
- The introduction of the data dependent tempered selection mechanism shows an innovative way to manage the flow of information in recurrent architectures.
The code will be available after the review process.

**Weaknesses:**

- While the innovations presented are technically very interesting, they also introduce a level of complexity that could be a barrier for widespread usage.
- Although Rodimus improves on the traditional recurrent models, it may still inherit some of the limitations associated with recurrent architectures (like the sequential processing bottlenecks).

**Questions:**

How does the semantic compression affect the fine-tuning performance on specific downstream tasks that require retaining nuanced context?

---

> ### Author Response · Authors · 2024-11-25
> **Response to Reviewer ZtZu (Part 1)**
>
> Thank you for your valuable feedback! We believe that addressing the reviewer’s comments has enhanced the clarity and presentation of the paper's contributions, raising it to a higher standard. We provide detailed responses to each comment. The reviewer's remarks are in italics, followed by our responses. Moreover, in the revised version of our paper, we mark all newly added or changed paragraphs in blue. Quotations from the revised paper are presented in blockquote format. Unless otherwise noted, all references to pages, equations, sections, and citations pertain to the revised version.
>
> _Q1 - While the innovations presented are technically very interesting, they also introduce a level of complexity that could be a barrier for widespread usage._
>
> We appreciate your acknowledgment of the technical interest behind our work and we agree that these models may present certain barriers to widespread adoption compared to softmax attention-based LLMs. However, we believe that the landscape is changing. **Increasing recognition and exploration of recurrent and hybrid models within both academic and industrial domains are gradually mitigating these challenges. We have mentioned this point in Appendix F in the revised version of our paper.**
>
> > Recent developments illustrate this trend, as numerous new recurrent and hybrid models have emerged. Notable examples include Mamba, Mamba2, RWKV, RWKV6, RetNet, HGRN, HGRN2, GLA, Hawk, and Griffin. Moreover, Mistral has successfully scaled Mamba to 7B parameters and has made it open-source. Similarly, Google has scaled Griffin to an impressive scale of 9B and open-sourced Recurrent-Gemma. These advancements indicate growing confidence in the performance of recurrent and hybrid models, often achieving comparable results compared to traditional softmax attention-based LLMs while maintaining lower complexity.
>
> > From an inference perspective, it is indeed worth highlighting that recurrent and hybrid models are inherently designed for efficiency. The per-token generation complexity for these models is theoretically $\mathcal O(1)$, in stark contrast to the $\mathcal O(T)$ complexity of softmax-based models, where $T$ denotes the context length. As a result, even without optimization, recurrent and hybrid models demonstrate rapid inference capabilities. Furthermore, state-of-the-art inference optimization frameworks such as vLLM, TensorRT-LLM, and OLlama have already begun to support the inference of models like Mamba and Mamba2, further paving the way for their more widespread usage.
> >
>
> _Q2 - Although Rodimus improves on the traditional recurrent models, it may still inherit some of the limitations associated with recurrent architectures (like the sequential processing bottlenecks)._
>
> We would like to clarify that Rodimus is designed to train in parallel through chunkwise parallelism algorithms. During inference, it employs a sequential processing approach while caching only a fixed-size hidden state to optimize memory usage. This clarification is mentioned at the end of Section 3.1.3 and further elaborated in Appendix B3.

---

> ### Author Response · Authors · 2024-11-25
> **Response to Reviewer ZtZu (Part 2)**
>
> _Q3 - How does the semantic compression affect the fine-tuning performance on specific downstream tasks that require retaining nuanced context?_
>
>
>
> Research indicates that semantic compression can influence the recall and retrieval performance of large language models (LLMs) (see, for instance, [R1]). In our original analysis, we compared our proposed model, Rodimus*, with various architectures, including recurrent models that utilize similar semantic compression techniques and full attention models like Pythia, all of which have comparable model sizes. **These comparisons were detailed in Section 4.2 and in Appendices E4 and E5, focusing on benchmarks from both the MQAR and NeedleBench datasets.**
>
>
>
> **To further investigate how semantic compression impacts the model's ability to comprehend nuanced context, we are incorporating additional recall benchmarks in Appendix D9 (Lines 1371-1395)** as follows:
>
> > We introduce several additional, simpler recall benchmarks that are outlined below:
> >
> > • FDA: This benchmark involves extracting specific details, such as device codes, classifications, and intended uses, from medical device reports submitted to the FDA.
> >
> > • SWDE: This task focuses on retrieving particular information, such as movie directors and university tuition, from HTML pages across 14 websites in domains like Movies and Universities.
> >
> > • SQuADv2: A question-answering task where models identify answer spans within Wikipedia articles based on given queries.
> >
> > • TriviaQA: This benchmark challenges models to answer questions using information from Wikipedia and other broad web sources, featuring diverse question formats.
> >
> > • NQ: Utilizing real Google search queries, this task requires models to find answers within Wikipedia, necessitating the extraction of relevant text snippets.
> >
> > • DROP: A reasoning-intensive question-answering task where models must carry out operations—including counting, sorting, or performing arithmetic—using information derived from Wikipedia passages.
> >
>
> > The same checkpoints used in the NeedleBench section are employed for evaluation, with the accuracy results presented in Table 11. **The average scores for both Rodimus and Rodimus$+$ outperform those of Mamba and Mamba2.** However, contrary to the trends observed in NeedleBench, Rodimus demonstrates a slight performance lag behind Pythia. This finding supports our earlier analysis in Appendix D8, which indicates that the softmax attention-based model Pythia surpasses recurrent models in recall tasks within the non-extrapolation interval. **Notably, the hybrid model Rodimus$+$, which integrates SW-SKA, successfully exceeds Pythia’s performance, highlighting that the incorporation of this hybrid model effectively addresses the performance limitations of recurrent models.**
> >
>
> | Model | SWDE | FDA | SQUAD | Drop | NQ | TriviaQA | AVG |
> | --- | --- | --- | --- | --- | --- | --- | --- |
> | Pythia | 53.05 | 81.65 | 41.89 | 21.85 | 34.49 | 58.18 | 48.52 |
> | Mamba | 45.36 | 59.85 | 41.28 | 20.84 | 34.43 | 60.43 | 43.70 |
> | Mamba2 | 52.67 | 71.12 | 42.19 | 22.86 | 35.79 | 63.57 | 48.03 |
> | Rodimus | 49.77 | 79.47 | 43.10 | 22.76 | 32.56 | 61.85 | 48.25 |
> | Rodimus+ | 58.76 | 80.84 | 48.74 | 24.58 | 36.27 | 64.63 | 52.30 |
>
>
>
>
> [R1] Waleffe R, Byeon W, Riach D, et al. An Empirical Study of Mamba-based Language Models[J]. arXiv preprint arXiv:2406.07887, 2024.

---

### Official Review · Reviewer_Az8Q · 2024-11-01

**Soundness:** 3
**Presentation:** 3
**Contribution:** 3
**Rating:** 6
**Confidence:** 5

**Summary:**

The paper introduces Rodimus and Rodimus+, two efficient alternatives to traditional Transformer models that aim to balance accuracy and computational efficiency. Rodimus uses a Data-Dependent Tempered Selection (DDTS) mechanism which is an improved parameterization of gated linear attention. Rodimus+ extends this with Sliding Window Shared-Key Attention (SW-SKA) for enhanced local token focus, further optimizing efficiency without sacrificing performance. Experiments demonstrate these models outperform comparable models on language modeling and recall tasks, making Rodimus+ a scalable, efficient alternative for large language models.

**Strengths:**

- This work comprehensively examines the design space of gated linear attention (GLA), particularly focusing on an outer-product-based gating structure with input and forget gates. It addresses a gap in systematic studies on GLA's gating mechanisms, positioning itself well to bridge this gap. The proposed Data-Dependent Tempered Selection (DDTS) mechanism is both intuitively appealing and practically useful, as demonstrated by experimental results. Additionally, the definition of the selection mechanism in this context is compelling.

- The ablation study on architectural design is relatively thorough.

**Weaknesses:**

- The work appears somewhat incremental, as it largely builds upon existing concepts. Technical contribution is limited.

- Regarding language modeling experiments, the choice of the WikiText-103 dataset for language modeling may not be ideal, as it is relatively simplistic and sensitive to hyperparameter tuning. For Figure 1a, larger-scale experiments using more extensive datasets would provide a more robust evaluation. Additionally, the controlled experiment scale in Table 2 is too limited; ideally, models with around 1B parameters trained on 100B tokens should be tested.

- Regarding recall-intensive eval,  the NeedleBench benchmark may be overly challenging for smaller-scale models, as evidenced by Figure 5, where all models perform poorly. Consider testing on simpler recall-intensive benchmarks (e.g., FDA, SWDE, NQ, SQuAD, TriviaQA, DROP as used in [1, Table 1]) for more interpretable results. Additionally, it would be insightful to know if Rodimus can outperform Mamba2 in recall performance with half the state size under equivalent training conditions. Results on these tasks could be easily added using the lm-eval-harness during the rebuttal period.

- Training and inference throughputs would be reported.

- Several minor errors are present throughout. See the list of specific corrections below.

---

### Typos and Minor Corrections:

- **Table 1:** Replace “suboptimal” with “second-best.”
- **Line 279:** Incorrect citation for GLA.
- **Table 4:** “State size d” should consistently be represented as $d^2 $.
- **Table 4:** For Rodimus,  $B_t$ should likely be $ (g_t^{\tau_t})^T \hat{\beta}_t $.
- **Line 1280:** The symbol $\beta$ in the first line should be $\hat{\beta}_t $.

---

### Reference:

[1] Just Read Twice: Closing the Recall Gap for Recurrent Language Models

**Questions:**

- How significant is $ d_t $ in Eq. (11)?

- Have you investigated the ordering of Rodimus, SW-SKA, and FFN in Eq. (13)?

---

> ### Author Response · Authors · 2024-11-25
> **Response to Reviewer Az8Q (Part 1)**
>
> Thank you for your valuable feedback! We believe that addressing the reviewer’s comments has enhanced the clarity and presentation of the paper's contributions, raising it to a higher standard. We provide detailed responses to each comment. The reviewer's remarks are in italics, followed by our responses. Moreover, in the revised version of our paper, we mark all newly added or changed paragraphs in blue. Quotations from the revised paper are presented in blockquote format. Unless otherwise noted, all references to pages, equations, sections, and citations pertain to the revised version.
>
> _Q1 - The work appears somewhat incremental, as it largely builds upon existing concepts. Technical contribution is limited._
>
> We would like to argue that **small yet significant advancements can play a crucial role in the evolution of NLP.** For instance, models such as SpanBERT, RoBERTA, and ELECTRA introduced modifications to BERT's pretraining tasks and masking mechanisms, yet have become essential tools for text embeddings. Similarly, the GPT series (from GPT-1 to GPT-3) implemented minor but impactful adjustments to model architecture—transitioning from post-norm to pre-norm architectures and incorporating variations like locally banded sparse attention. These developments laid the groundwork for the state-of-the-art ChatGPT and GPT-4, which excel in a wide array of NLP tasks.
>
> In the context of our proposed gating mechanism, DDTS, we have outlined its distinctions from previous methods in **Table 4 of Appendix B1**. The specific formulation of DDTS provides several notable advantages:
>
> + **Dynamic Control of Hidden State Updates**: DDTS allows for data-dependent selection through $\boldsymbol{A}_t$ and $\boldsymbol{B}_t$ when writing and deleting information, enabling dynamic control of hidden state updates. This feature is discussed in Section 3.1.2 (Lines 293-295), with a formal proof provided in Appendix C2.
> + **Efficient Parallelization**: As DDTS performs data-dependent selection solely along the dimension $n$, it supports the implementation of chunkwise parallel algorithms. This capability enables efficient parallel training while maintaining flexibility, as elaborated in Section 3.1.3 (Lines 317-319) and Appendix B3.
> + **Enhanced Flexibility through the Temperature Gate:** The temperature gate $\boldsymbol{\tau}_t$ introduces additional flexibility in the selection mechanism, allowing for more precise control over information compression and representation. This aspect sets our approach apart from existing works, as noted in Section 3.1.2 and detailed in Appendix E2.
>
> To further emphasize the distinctions of Rodimus from previous approaches, **we have incorporated the above discussion in Appendix B1.**
>
> Empirically, Rodimus demonstrates superior performance compared to Mamba and Mamba2 when trained under identical conditions, as illustrated in the first experiment in Table 2 (Lines 434-436) and further validated in our newly added experiment detailed in Appendix D6.

---

> ### Author Response · Authors · 2024-11-25
> **Response to Reviewer Az8Q (Part 2)**
>
> _Q2 - Regarding language modeling experiments, the choice of the WikiText-103 dataset for language modeling may not be ideal, as it is relatively simplistic and sensitive to hyperparameter tuning. For Figure 1a, larger-scale experiments using more extensive datasets would provide a more robust evaluation. Additionally, the controlled experiment scale in Table 2 is too limited; ideally, models with around 1B parameters trained on 100B tokens should be tested._
>
> We greatly appreciate your insights and have implemented your suggestions by **adding three additional experiments to strengthen our work.**
>
>
>
> First, we explored the language modeling performance of larger models—specifically, **those with 350M parameters—trained on 7B tokens from the Pile dataset**, as outlined in Table 8. The results of this analysis can be found **in Appendix D2 on Page 22** as follows:
>
> > In addition to WikiText-103, we also train various model architectures, including Transformer$++$ (full attention), Transformers++ with sliding window attention (sparse attention), HGRN2 (linear attention), Mamba/Mamba2 (state-space models), Rodimus, and Rodimus$+$, using 7B tokens in the Pile dataset. This dataset is considerably larger than WikiText-103.  This training aims to investigate the language modeling performance across different model architectures, with the training settings adhering to the 350M parameter configuration outlined in Table 8. The results are summarized in the following Table.
> >
>
> | Model | Valid PPL |
> | --- | --- |
> | Transformer++ | 6.04 |
> | Transformer++-SWA | 6.13 |
> | HGRN2 | 6.08 |
> | Mamba | 6.19 |
> | Mamba2 | 6.02 |
> | Rodimus | 5.88 |
> | Rodimus+ | 5.88 |
>
>
> > Consistent with the findings in Table 1, **Rodimus demonstrates superior language modeling performance compared to other models, including Transformer++ and Mamba2.** However, unlike the results from Table 1, we find that Rodimus$+$ and Rodimus exhibit similar performance levels under this scenario. We attribute this similarity to the limited scale of the dataset. Note that Rodimus$+$ needs to distribute the tasks of global and local dependency modeling respectively to the Rodimus block and the SW-SKA, making it more challenging to train than the original Rodimus and thus requiring more data. Despite this, **Rodimus+ shows greater potential for enhanced language modeling performance compared to Rodimus when deployed at a larger parameter scale, as illustrated in Figure 4.**
> >
>
> Second, to address the accuracy-efficiency trade-off, we conducted experiments with key models (Mamba2, Rodimus, and StreamingLLM) using larger configurations of 125M parameters trained on a more extensive dataset of 2.5B tokens from Pile. **The perplexity of different models as a function of memory footprint (cache only) is summarized in the table below and discussed in detail in Appendix D3 (Lines 1209-1215)** as below:
>
> |  Model | 1.2M Bytes | 2.3M Bytes | 4.7M Bytes | 9.4M Bytes |
> | --- | --- | --- | --- | --- |
> | StreamingLLM | 24.76 | 17.54 | 12.57 | 9.39 |
> | Mamba2 | 8.16 | 7.91 | 7.81 | 7.61 |
> | Rodimus | 7.60 | 7.47 | 7.35 | 7.33 |
>
>
> > To further validate the memory efficiency of Rodimus, we conduct experiments using Mamba2, Rodimus, and StreamingLLM on a larger dataset, as shown in Table 7. The foundational experimental settings are aligned with the 125M configurations described in Table 8. Once again, we adjust the memory footprint by modifying the expansion factor $n$. **The results of PPL summarized in Table7 indicate that Rodimus continues to outperform the accuracy-efficiency trade-off when trained at a larger scale compared to WikiText-103.**
> >
>
> Third, we have scaled up the number of parameters for Mamba2 and Rodimus to 1.4B, and trained them on 100B tokens from Pile using the same configurations. **The results of these models on downstream tasks are presented in the table below and detailed in Appendix D6. Once again, Rodimus demonstrates superior average performance over Mamba2.**
>
> | Model | Params (B) | Tokens (B) | Tokenizer | ARC-c (acc_norm) | ARC-e (acc) | HS (acc_norm) | LMB (ppl) | LMB (acc) | OBQA (acc_norm) | PIQA (acc) | WG (acc) | AVG (acc*) |
> | --- | --- | --- | --- | --- | --- | --- | --- | --- | --- | --- | --- | --- |
> | Mamba2 | 0.13 | 100 | NeoX | 23.81 | 45.50 | 34.13 | 20.42 | 39.55 | 29.20 | 64.47 | 51.85 | 41.22 |
> | Rodimus | 0.13 | 100 | NeoX | 23.38 | 46.84 | 34.22 | 17.95 | 42.44 | 30.00 | 65.40 | 51.46 | 41.96 |
> | Mamba2 | 1.4 | 100 | NeoX | 26.54 | 56.06 | 48.74 | 8.22 | 55.25 | 33.40 | 70.57 | 54.14 | 49.24 |
> | Rodimus | 1.4 | 100 | NeoX | 28.75 | 56.61 | 49.51 | 7.78 | 56.05 | 34.00 | 70.78 | 52.41 | 49.73 |

---

> ### Author Response · Authors · 2024-11-25
> **Response to Reviewer Az8Q (Part 3)**
>
> _Q3 - Regarding recall-intensive eval, the NeedleBench benchmark may be overly challenging for smaller-scale models, as evidenced by Figure 5, where all models perform poorly. Consider testing on simpler recall-intensive benchmarks (e.g., FDA, SWDE, NQ, SQuAD, TriviaQA, DROP as used in [1, Table 1]) for more interpretable results. Additionally, it would be insightful to know if Rodimus can outperform Mamba2 in recall performance with half the state size under equivalent training conditions. Results on these tasks could be easily added using the lm-eval-harness during the rebuttal period._
>
> Thank you very much for your insightful feedback!
> We initially chose the NeedleBench benchmark due to its challenging nature, especially since we already have MRQA as a more accessible benchmark. However, after considering your comments, we agree that testing on simpler recall-intensive benchmarks would be beneficial for clarity. **In Appendix D9, we have decided to include several additional, less demanding recall benchmarks as follows:**
>
> > The NeedleBench benchmark may be excessively demanding for small pre-trained models with a context length limitation of 2K tokens. To mitigate this issue, we introduce several additional, simpler recall benchmarks that are outlined below:
> >
> > • **FDA**: This benchmark involves extracting specific details, such as device codes, classifications, and intended uses, from medical device reports submitted to the FDA.
> >
> > • **SWDE**: This task focuses on retrieving particular information, such as movie directors and university tuition, from HTML pages across 14 websites in domains like Movies and Universities.
> >
> > • **SQuADv2**: A question-answering task where models identify answer spans within Wikipedia articles based on given queries.
> >
> > • **TriviaQA**: This benchmark challenges models to answer questions using information from Wikipedia and other broad web sources, featuring diverse question formats.
> >
> > • **NQ**: Utilizing real Google search queries, this task requires models to find answers within Wikipedia, necessitating the extraction of relevant text snippets.
> >
> > • **DROP**: A reasoning-intensive question-answering task where models must carry out operations—including counting, sorting, or performing arithmetic—using information derived from Wikipedia passages.
> >
>
>
> > The same checkpoints used in the NeedleBench section are employed for evaluation, with the accuracy results presented in Table 11. **The average scores for both Rodimus and Rodimus$+$ outperform those of Mamba and Mamba2.** However, contrary to the trends observed in NeedleBench, Rodimus demonstrates a slight performance lag behind Pythia. This finding supports our earlier analysis in Appendix D8, which indicates that the softmax attention-based model Pythia surpasses recurrent models in recall tasks within the non-extrapolation interval. **Notably, the hybrid model Rodimus$+$, which integrates SW-SKA, successfully exceeds Pythia’s performance, highlighting that the incorporation of this hybrid model effectively addresses the performance limitations of recurrent models.**
> >
>
> | Model | SWDE | FDA | SQUAD | Drop | NQ | TriviaQA | AVG |
> | --- | --- | --- | --- | --- | --- | --- | --- |
> | Pythia | 53.05 | 81.65 | 41.89 | 21.85 | 34.49 | 58.18 | 48.52 |
> | Mamba | 45.36 | 59.85 | 41.28 | 20.84 | 34.43 | 60.43 | 43.70 |
> | Mamba2 | 52.67 | 71.12 | 42.19 | 22.86 | 35.79 | 63.57 | 48.03 |
> | Rodimus | 49.77 | 79.47 | 43.10 | 22.76 | 32.56 | 61.85 | 48.25 |
> | Rodimus+ | 58.76 | 80.84 | 48.74 | 24.58 | 36.27 | 64.63 | 52.30 |

---

> ### Author Response · Authors · 2024-11-25
> **Response to Reviewer Az8Q (Part 4)**
>
> _Q4 - Training and inference throughputs would be reported._
>
> Following your comments, **we have included a comprehensive analysis of FLOPs, memory usage, and throughput for Rodimus\*, comparing it with Transformer and Mamba2 models in Appendix E1** as follows:
>
> > In this section, we present the FLops and throughputs (words per second, wps) of Rodimus*, Mamba2, and the Transformer model, both during training and inference. Additionally, we provide an analysis of memory usage during inference, excluding model weights to ensure a fair evaluation of space complexity. The specific configurations examined correspond to the 1.3B settings outlined in Table 8. For the Transformer model, we evaluate its attention mechanism implemented in both PyTorch (denoted as "torch") and Flash Attention 2 (denoted as "F.A."). The PyTorch implementation adheres to the theoretical time and space complexity of softmax attention, while Flash Attention significantly optimizes computation speed and reduces memory usage. Our analysis of FLOPs and throughputs focuses on scenarios with a batch size of one, with results detailed in Tables 19 and 20.
> >
>
>
> > The results reveal that the FLOPs and memory footprint of Rodimus are the smallest among all models tested, indicating the time and space efficiency of the proposed architecture. **Particularly, Rodimus consumes only half the memory of Mamba2,** making it an attractive option for resource-constrained environments, such as edge devices or low-power systems. Conversely, Rodimus$+$ exhibits a slight increase in FLOPs but a significant rise in memory consumption due to the integration of SW-SKA and FFN components. Overall, both Rodimus and Rodimus$+$ demonstrate greater time and space efficiency than the Transformer model, with Rodimus being more efficient than Mamba2 as well. **Notably, Rodimus, Rodimus$+$, and Mamba2 can manage long sequences (e.g., pre-filling 24K tokens) more effectively than Transformers.**
> >
>
>
> > However, it is noteworthy that the throughput of Mamba2 and the Transformer (F.A.) is significantly higher than that of Rodimus*. This discrepancy arises because Rodimus* does not incorporate the I/O-aware optimizations that are implemented in these baselines. **In future work, we aim to address these throughput limitations by integrating I/O-aware optimizations to enhance Rodimus*'s efficiency during both training and inference.**
> >
>
>
>                                     Throughputs and FLOPs during training.
>
> | Model | FLOPs $ \times 10^{13} $ (4K-length contexts) | Throughputs(wps) (4K-length contexts) |
> | --- | --- | --- |
> | Transformer (torch) | 4.46 | 2726.64 |
> | Transformer (F.A.) | - | 8547.29 |
> | Mamba2 | 3.65 | 10147.10 |
> | Rodimus | 3.60 | 8592.59 |
> | Rodimus+ | 4.29 | 9410.38 |
>
>
>                                     Throughputs, FLOPs, and Memory Footprint during inference.
>
> | Model | FLOPs $ \times 10^{9} $(4K-length contexts) | Throughputs(wps) (4K-length contexts) | Memory footprint (M Bytes) (4K-length contexts) | FLOPs$ \times 10^{9} $(24K-length contexts) | Throughputs(wps) (24K-length contexts) | Memory footprint (M Bytes) (24K-length contexts) |
> | --- | --- | --- | --- | --- | --- | --- |
> | Transformer (torch) | 3.6 | 2.4 | 805 | 7.7 | OOM | 4831.8 |
> | Transformer (F.A.) | - | 49.4 | - | - | 11.0 | - |
> | Mamba2 | 3.0 | 59.3 | 0.8 | 3.0 | 59.3 | 0.8 |
> | Rodimus | 2.9 | 21.4 | 0.4 | 2.9 | 21.4 | 0.4 |
> | Rodimus+ | 3.5 | 19.6 | 207.8 | 3.5 | 19.6 | 207.8 |
>
> _Q5 - Several minor errors are present throughout. See the list of specific corrections below._
>
> Thank you for pointing out the writing issues, which we have addressed in the revised version.
>
> _Q5-3 Table 4: “State size d” should consistently be represented as _$ d^2 $_._
>
> We would like to clarify the "State Size $n\times m$" column in Table 4. Here, $n$ represents the expansion factor, which is a fixed constant in models, while $m$ is related to the original model dimension $d$. Due to differences in model architecture design, the actual $m$ varies across models. For instance, it is $m=2d$ in Mamba and Mamba2, and $m=d$ in Linear Attention. To better differentiate the actual state size of these models with fixed hidden state, we avoid writing it in the form of $d^2$ or $d$, providing a more intuitive representation for readers.

---

> ### Author Response · Authors · 2024-11-25
> **Response to Reviewer Az8Q (Part 5)**
>
> _Q6 - How significant is _$ d_t $_ in Eq. (11)?_
>
> We have given the definition of $\boldsymbol{d}_t$ in Section 3.1.3.
>
> > The learnable weight $\boldsymbol{d}_t$ functions as the feedthrough matrix in SSMs.
> >
>
> In this paper, our primary focus is on the design of DDTS, particularly the gates used in Eq. 11. Regarding $\boldsymbol{d}_t$, we adopt the design used in SSMs such as Mamba and Mamba2. As it is not the main focus of our work, we did not conduct an ablation study to show its importance. However, relevant examples are available in the literature (e.g., [R1]), where this design has been shown to be significant.
>
> [R1] Chen C T. Linear system theory and design[M]. Saunders college publishing, 1984.
>
> _Q7 - Have you investigated the ordering of Rodimus, SW-SKA, and FFN in Eq. (13)?_
>
> Following your comments, **we have check the impact of the ordering of the blocks in Eq. (13) in Appendix D10 (Lines 1454-1467) as follows:**
>
> > We investigate the impact of the ordering of the Rodimus Block, SW-SKA, and FFN on model performance. The results are detailed in Table 18. Specifically, for the Pile subset, the model settings correspond to 350 million items, as referenced in Table 8. Since SW-SKA and FFN are integrated within a two-residual hop structure, we treat them as a single unit and focus on the relative positioning of the Rodimus Block with respect to SW-SKA and FFN. The term "Rodimus+X" denotes the original order presented in Eq. (13), while "X+Rodimus" indicates the rearrangement of Rodimus's position relative to SW-SKA and FFN in the same equation.
> >
>
>
>
> > The findings in Table 18 reveal that for shallow models tested on the WikiText-103 dataset, **it is essential to place the Rodimus Block before SW-SKA and FFN.** In contrast, the results from the Pile dataset suggest that as the number of layers increases, the significance of the Rodimus Block's positioning diminishes. We hypothesize that this occurs because an increase in layers lessens the impact of module types that are closer to the word embedding and output projection on overall model performance.
> >
>
> | Configuration | WikiText-103 PPL (n_layer=6) | Pile PPL (n_layer=24) |
> | --- | --- | --- |
> | Rodimus + X | 21.56 | 5.9 |
> | X + Rodimus | 21.95 | 5.9 |

---

> > ### Comment · Reviewer_Az8Q · 2024-11-25
> >
> > Thanks for your response. My concern is mostly address and I've increased my score to 6.

---

> ### Author Response · Authors · 2024-11-26
> **Acknowledgment of your feedback and contributions**
>
> We sincerely appreciate your acknowledgment that our additional experiments and clarifications have effectively addressed your concerns.
> Once again, we extend our heartfelt gratitude for your positive feedback and invaluable contributions to enhancing the quality of our work.

---

### Official Review · Reviewer_kK5a · 2024-11-03

**Soundness:** 3
**Presentation:** 3
**Contribution:** 3
**Rating:** 6
**Confidence:** 4

**Summary:**

Attention is the most memory-costly operation in transformer architecture. It scales quadratically in sequence length, as it can correlates each position to each other. To overcome this bottleneck, state space models were introduced, however usually degenerated in downstream performance.
The authors show that their particular gating mechanics of state space models "Rodimus/DDTS" improve downstream performance as they may model language more naturally. It in particular comprises a low-rank auto-encoder approximation and a softmax based temperature parameter.
Furthermore the authors introduce a shared key attention "SKA", that reduces some memory footprint and can be shown to have the same expressiveness to standard attention. Both methods are "Rodimus+".
Lots of efforts was done with evaluations, albeit, i'm afraid, not in a comparable fashion..

**Strengths:**

the paper is generally well written and presented. ideas appear thought-through and are (somewhat) mathematically reasoned.
more reasonable interplay of gating mechanics / ssm's in general and attention is pretty important to iterate towards more robust language models.

**Weaknesses:**

1) i understood the change to Rodimus+ is the two norms, SW-SKA and FFN per layer. how can this sum up to those negligible changes in parameters in Tab 2 - or do you use entirely different architecture configs?
2) i do not find anything conclusive about runtime estimates (only a vague big-O estimation and some chunking approach for training). This should not be optional!
3) my major concern is that only Fig 1 / Tab 1 / Fig 4 appear to be "somewhat fair" comparisons but in a ridiculous low scale. i cannot follow steps in section 4 most of the time. in particular i could not reconstruct the experiment design due to missing descriptions partially found in appendix.
4) table 2 appears pretty delusive. variances in the dataset and tokenizer alone are able to cause significant variations already, which is not discussed at all and not negligible! the only comparable lines appear to be the first 3. the only thing i can derive here is that rodimus can  model language. a sentence like line 478 is pretty off standard to me. you pretend to have a better method so do a fair 1:1 training/ scaling analysis (on less models). tab 14 should be sufficient for such a case study, but is also pretty uncomparable?

**Questions:**

1) line 120 : "matrices"
2) Tab 1 only ~2M parameters increase to Rodimus+ is the SKA + FFM? other hyperparameters are the same? the changes appear to be pretty random in tab 2

---

> ### Author Response · Authors · 2024-11-25
> **Response to Reviewer kK5a (Part 1)**
>
> Thank you for your valuable feedback! We believe that addressing the reviewer’s comments has enhanced the clarity and presentation of the paper's contributions, raising it to a higher standard. We provide detailed responses to each comment. The reviewer's remarks are in italics, followed by our responses. Moreover, in the revised version of our paper, we mark all newly added or changed paragraphs in blue. Quotations from the revised paper are presented in blockquote format. Unless otherwise noted, all references to pages, equations, sections, and citations pertain to the revised version.
>
> _Q1 - i understood the change to Rodimus+ is the two norms, SW-SKA and FFN per layer. how can this sum up to those negligible changes in parameters in Tab 2 - or do you use entirely different architecture configs? Tab 1 only ~2M parameters increase to Rodimus+ is the SKA + FFM? other hyperparameters are the same? the changes appear to be pretty random in tab 2_
>
> **We would like to clarify that the number of layers in both Rodimus and Rodimus+ are the same; the distinction in model size is primarily attributed to the variations in parameters between the original Rodimus block and the SW-SKA plus the FFN in Rodimus+.**
>
>
>
> In Rodimus, each layer consists of two Rodimus Blocks to preserve the parameter count of the original Transformer architecture. This design is similar to that employed in the Mamba and Mamba2 models. Consequently, the effective number of layers is doubled compared to the original n_layer configuration. We have elaborated on this point in Appendix D4 (Lines 1228-1230) of our paper, stating:
>
> > For methods using purely recurrent models that combine GLU and token mixers, such as Mamba series and Rodimus, the number of layers **should be doubled** to match the GPT-3 specifications.
> >
>
>
>
> In Rodimus+, we replace the second Rodimus block in each layer with SW-SKA and FFN, but the overall layer count remains unchanged. To clarify the parameter count: the Rodimus Block consists of $6d^2$ parameters, where $d$ is the model dimension. Since each layer contains two Rodimus blocks, the total parameter count for Rodimus sums to $12d^2$. In contrast, the Rodimus+ Block consists of $13d^2$ parameters, comprising one Rodimus block ($6d^2$), alongside multi-head projection, multi-query projection, and output projection in the attention mechanism ($3d^2$), and the FFN ($4d^2$).
>
>
>
> As a result, there are slight differences between Rodimus and Rodimus+, with the differences becoming more pronounced as $d$ increases. **We have mentioned this point in Appendix D4 (Lines 1231-1237) in the revised version of our paper.**

---

> ### Author Response · Authors · 2024-11-25
> **Response to Reviewer kK5a (Part 2)**
>
> _Q2 - i do not find anything conclusive about runtime estimates (only a vague big-O estimation and some chunking approach for training). This should not be optional!_
>
>
>
> Thanks for pointing this out! Following your comments, **we have included a comprehensive analysis of FLOPs, memory usage, and throughput for Rodimus\*, comparing it with Transformer and Mamba2 models in Appendix E1** as follows:
>
> > In this section, we present the FLOPs and throughputs (words per second, wps) of Rodimus*, Mamba2, and the Transformer model, both during training and inference. Additionally, we provide an analysis of memory usage during inference, excluding model weights to ensure a fair evaluation of space complexity. The specific configurations examined correspond to the 1.3B settings outlined in Table 8. For the Transformer model, we evaluate its attention mechanism implemented in both PyTorch (denoted as "torch") and Flash Attention 2 (denoted as "F.A."). The PyTorch implementation adheres to the theoretical time and space complexity of softmax attention, while Flash Attention significantly optimizes computation speed and reduces memory usage. Our analysis of FLOPs and throughputs focuses on scenarios with a batch size of one, with results detailed in Tables 19 and 20.
> >
>
>
>
> > The results reveal that the FLOPs and memory footprint of Rodimus are the smallest among all models tested, indicating the time and space efficiency of the proposed architecture. **Particularly, Rodimus consumes only half the memory of Mamba2, making it an attractive option for resource-constrained environments, such as edge devices or low-power systems.** Conversely, Rodimus$+$ exhibits a slight increase in FLOPs but a significant rise in memory consumption due to the integration of SW-SKA and FFN components. Overall, both Rodimus and Rodimus$+$ demonstrate greater time and space efficiency than the Transformer model, with Rodimus being more efficient than Mamba2 as well. **Notably, Rodimus, Rodimus$+$, and Mamba2 can manage long sequences (e.g., pre-filling 24K tokens) more effectively than Transformers.**
> >
>
>
>
> > However, it is noteworthy that the throughput of Mamba2 and the Transformer (F.A.) is significantly higher than that of Rodimus*. This discrepancy arises because Rodimus* does not incorporate the I/O-aware optimizations that are implemented in these baselines. **In future work, we aim to address these throughput limitations** by integrating I/O-aware optimizations to enhance Rodimus*'s efficiency during both training and inference.
> >
>
>
>                                 Throughputs and FLOPs during training.
>
> | Model | FLOPs $ \times 10^{13} $(4K-length contexts) | Throughputs(wps) (4K-length contexts) |
> | --- | --- | --- |
> | Transformer (torch) | 4.46 | 2726.64 |
> | Transformer (F.A.) | - | 8547.29 |
> | Mamba2 | 3.65 | 10147.10 |
> | Rodimus | 3.60 | 8592.59 |
> | Rodimus+ | 4.29 | 9410.38 |
>
>                                 Throughputs, FLOPs, and Memory Footprint during inference.
>
> | Model | FLOPs $ \times 10^{9} $(4K-length contexts) | Throughputs(wps) (4K-length contexts) | Memory footprint (M Bytes) (4K-length contexts) | FLOPs$ \times 10^{9} $(24K-length contexts) | Throughputs(wps) (24K-length contexts) | Memory footprint (M Bytes) (24K-length contexts) |
> | --- | --- | --- | --- | --- | --- | --- |
> | Transformer (torch) | 3.6 | 2.4 | 805 | 7.7 | OOM | 4831.8 |
> | Transformer (F.A.) | - | 49.4 | - | - | 11.0 | - |
> | Mamba2 | 3.0 | 59.3 | 0.8 | 3.0 | 59.3 | 0.8 |
> | Rodimus | 2.9 | 21.4 | 0.4 | 2.9 | 21.4 | 0.4 |
> | Rodimus+ | 3.5 | 19.6 | 207.8 | 3.5 | 19.6 | 207.8 |

---

> ### Author Response · Authors · 2024-11-25
> **Response to Reviewer kK5a (Part 3)**
>
> _Q3 - my major concern is that only Fig 1 / Tab 1 / Fig 4 appear to be "somewhat fair" comparisons but in a ridiculous low scale. i cannot follow steps in section 4 most of the time. in particular i could not reconstruct the experiment design due to missing descriptions partially found in appendix._
>
> We have listed all detailed experimental settings in Appendix D. Specifically,
>
> + The setups of language modeling on Wikitext-103 can be found in Appendix D1 (Page 22, Lines 1138-1160) and Table 5.
> + The setups of the scaling law experiment can be found in Appendix D4 (Page 23, Lines 1224-1237) and Table 6.
> + The setups of the downstream tasks can be found in Section 4.1 (Page 9, Lines 465-485) and Appendix D5 (Page 23-24, Lines 1239-1268).
> + The setups of the MRQA experiment can be found in Appendix D7 (Page 24-25, Lines 1291-1314).
> + The setups of the NeedleBench experiment can be found in Appendix D8.
>
>
>
> After checking your comments, we also think it is better to check the language modeling performance on larger models. Specifically, we train benchmark models with 350M parameters following the 350M parameter configuration outlined in Table 8.  The results of this analysis are detailed in Appendix D2 on Page 22.
>
>
>
> > In addition to WikiText-103, we also train various model architectures, including Transformer$++$ (full attention), Transformers++ with sliding window attention (sparse attention), HGRN2 (linear attention), Mamba/Mamba2 (state-space models), Rodimus, and Rodimus$+$, using 7B tokens in the Pile dataset. This dataset is considerably larger than WikiText-103.  This training aims to investigate the language modeling performance across different model architectures, with the training settings adhering to the 350M parameter configuration outlined in Table 8. The results are summarized in Table 6.
> >
>
> | Model | Valid PPL |
> | --- | --- |
> | Transformer++ | 6.04 |
> | Transformer++-SWA | 6.13 |
> | HGRN2 | 6.08 |
> | Mamba | 6.19 |
> | Mamba2 | 6.02 |
> | Rodimus | 5.88 |
> | Rodimus+ | 5.88 |
>
>
> > Consistent with the findings in Table 1, **Rodimus demonstrates superior language modeling performance compared to other models, including Transformer++ and Mamba2.** However, unlike the results from Table 1, we find that Rodimus$+$ and Rodimus exhibit similar performance levels under this scenario. We attribute this similarity to the limited scale of the dataset. Note that Rodimus$+$ needs to distribute the tasks of global and local dependency modeling respectively to the Rodimus block and the SW-SKA, making it more challenging to train than the original Rodimus and thus requiring more data. Despite this, **Rodimus+ shows greater potential for enhanced language modeling performance compared to Rodimus when deployed at a larger parameter scale, as illustrated in Figure 4.**
> >

---

> ### Author Response · Authors · 2024-11-25
> **Response to Reviewer kK5a (Part 4)**
>
> _Q4 - table 2 appears pretty delusive. variances in the dataset and tokenizer alone are able to cause significant variations already, which is not discussed at all and not negligible! the only comparable lines appear to be the first 3. the only thing i can derive here is that rodimus can model language. a sentence like line 478 is pretty off standard to me. you pretend to have a better method so do a fair 1:1 training/ scaling analysis (on less models). tab 14 should be sufficient for such a case study, but is also pretty uncomparable?_
>
>
> First of all, we would like to clarify that we have already provided a rigorous comparison between Rodimus and other recurrent models in the first block of Table 2. Specifically, we trained Rodimus, Mamba, and Mamba2—each with 130M parameters—under identical settings on 100 billion tokens from the Pile dataset, and assessed their performance on downstream tasks. Our findings illustrate that Rodimus consistently outperforms both Mamba and Mamba2 in terms of average performance.
>
>
>
> In response to your concerns, **we conducted additional experiments using larger models, training Rodimus and Mamba2 with 1.4B parameters from scratch under the same conditions on 100B tokens from the Pile dataset. The results of these models on downstream tasks are presented in the table below and detailed in Appendix D6. Once again, Rodimus demonstrates superior average performance over Mamba2.**
>
> | Model | Params (B) | Tokens (B) | Tokenizer | ARC-c (acc_norm) | ARC-e (acc) | HS (acc_norm) | LMB (ppl) | LMB (acc) | OBQA (acc_norm) | PIQA (acc) | WG (acc) | AVG (acc*) |
> | --- | --- | --- | --- | --- | --- | --- | --- | --- | --- | --- | --- | --- |
> | Mamba2 | 0.13 | 100 | NeoX | 23.81 | 45.50 | 34.13 | 20.42 | 39.55 | 29.20 | 64.47 | 51.85 | 41.22 |
> | Rodimus | 0.13 | 100 | NeoX | 23.38 | 46.84 | 34.22 | 17.95 | 42.44 | 30.00 | 65.40 | 51.46 | 41.96 |
> | Mamba2 | 1.4 | 100 | NeoX | 26.54 | 56.06 | 48.74 | 8.22 | 55.25 | 33.40 | 70.57 | 54.14 | 49.24 |
> | Rodimus | 1.4 | 100 | NeoX | 28.75 | 56.61 | 49.51 | 7.78 | 56.05 | 34.00 | 70.78 | 52.41 | 49.73 |
>
>
>
>
> On the other hand, we acknowledge that the comparisons in Blocks 2-4 of Table 2 may not be perfectly equitable. We utilized open-source checkpoints of benchmark models instead of training them identically from scratch. While we recognize that this may introduce some inconsistencies, such comparisons are commonly practiced in the literature, including Mamba, Mamba2 and Griffin. **Due to resource limitations, we, like the authors of those papers, were unable to train all benchmark models under uniform conditions.** We have noted these differences in our manuscript on Lines 474-483 on Page 9.
>
>
>
> In particular for Blocks 3 and 4, we aim to **demonstrate Rodimus's potential as a practical small LLM, akin to models like Qwen, RWKV6, and recurrent Gemma.** This involved curating our own training data and developing tokenization strategies, followed by a comparative analysis with other open-source small LLMs that have undertaken similar efforts in data preparation and tokenizer creation. In other words, **all of these benchmark models strive for optimal performance in terms of data preparation, tokenization, and model architecture design, making the comparisons reasonably fair.** Along this direction, **we have continued pretraining Rodimus+-1.6B on 2.5T tokens and show its superiority over other pratical small LLMs including Gemma2-2B, Qwen2-1.5B, RWKV6-1.6B, and Llama3.2-1.2B in Table 21 in Appendix E3.** We plan to release our checkpoints and model code upon the paper's publication. Additionally, we have detailed the various tokenizers employed in the benchmark models within Table 2 and **referenced this information on Pages 9, Lines 484-485.**
>
> _Q5 - line 120 : "matrices"_
>
> Thanks for pointing out this typo, which we have addressed in the revised version.

---

> > ### Comment · Reviewer_kK5a · 2024-11-25
> >
> > thank you for this thorough update, i increased my scores

---

> ### Author Response · Authors · 2024-11-26
> **Acknowledgment of your feedback and contributions**
>
> We greatly appreciate your thoughtful feedback and are glad to know that our additional experiments and clarifications have sufficiently addressed your concerns.
> Thank you once again for your valuable insights and constructive suggestions, which have been instrumental in improving the quality of our work.

---

### Official Review · Reviewer_Cz2E · 2024-11-04

**Soundness:** 3
**Presentation:** 3
**Contribution:** 2
**Rating:** 6
**Confidence:** 4

**Summary:**

This paper introduces two efficient Transformer architectures that incorporate semantic, token, and head compression: Rodimus, which is purely based on linear recurrence and, Rodimus+, which combines linear recurrence with sliding window attention to improve the quality and efficiency tradeoff in LLMs. The key components proposed include a data-dependent tempered selection mechanism (DDTS) and sliding window shared-key attention (SW-SKA). Experiments with models between 130M to 1.6B on language modeling, various downstream tasks, and retrieval tasks show that Rodimus+ achieves better performance than models based on recurrence such as Mamba and based on standard attention such as Pythia.

**Strengths:**

1. The paper introduces two new techniques, data-dependent tempered selection (DDTS) and shared-key attention (SKA). DDTS is a new gating-based recurrent type for state-space models which enhances performance and increases parameter efficiency through reducing hidden state size. SKA shares a single key representation to reduce memory footprint while maintaining the multi-value setting of the original multi-head attention.
2. These techniques are evaluated extensively in general tasks (content analysis, commonsense reasoning, reading comprehension, etc), long-context retrieval tasks (NeedleBench) and show improved performance compared to various efficient and standard transformer models.
3. Overall, the paper is satisfactorily written, the proposed techniques are clearly described, and the prior work moderately well covered. The experiment section is thorough and has an organization that is easy to follow.

**Weaknesses:**

1. While the main focus of the paper is to improve the trade-off in terms of accuracy and efficiency, the experiments do not discuss the efficiency aspects such as memory cost and latency or times for  training/inference.
2. Even though the specific gates proposed haven't been used in prior work, the novelty in the design can be considered rather incremental as the formulation is very similar to previous ones (Mamba, S4D).
3. Direct comparisons with alternative gating functions in SSMs using the exact same architecture and model size are lacking. This prevents the comprehensive understanding of the benefits brought by the specific gating techniques.
4. The evidence provided in this paper to support the claim of "breaking" the accuracy-efficiency tradeoff is not comprehensive.
   - Several dimensions are lacking in the experimental setup: comparison with a wide range of efficient attention models (sparse, linear, hybrid), head-to-head comparisons (equal number of params, HPO budget), and rigorous hyperparameter optimization.
   - In addition, the performance of Rodimus compared to Mamba when using the same number of parameters is actually the same or worse.
   - Only when combining with SWA and increasing number of parameters performance improves further with Rodimus+ but this is not a valuable comparison since Mamba2 is not a hybrid method.

**Questions:**

1. Was there any specific reason for not using the same number of parameters for Rodimus and Mamba models? For instance, in the third block of Table 2 Mamba has 0.37B parameters while Rodimus has 0.46+B parameters and in the fourth block Mamba has 1.3/1.4 while Rodimus 1.4/1.6. Such differences make it difficult to understand if the improvement comes from the proposed gating design or from pure increase in capacity. To strengthen the claims made in the paper, I'd suggest to the authors to provide a head-to-head comparison with that model.
2. There is lack of comparison with hybrid models, even though Rodimus+ is in that category of models. Have you considered comparing with such models? A few examples include: Jamba, Griffin, Recurrent-Gemma.
3. What is the memory consumption and training/inference times for the models compared in the experiments? Even though the efficiency aspect is emphasized a lot in the beginning of the paper, it is under explored in the later sections.
4. Can the authors report language modeling scores with the larger model sizes? The ones reported on Wikitext-103 are only with very small models and the conclusions that we can draw from it are limited.
5. In Figures 1a and 1b, what is the performance of a standard Transformer?
6. What is the memory benefit of SKA compared to alternative attention mechanisms such as MQA and GQA? It would be useful to include an ablation study that highlights their differences.

---

> ### Author Response · Authors · 2024-11-25
> **Response to Reviewer Cz2E (Part 1)**
>
> Thank you for your valuable feedback! We believe that addressing the reviewer’s comments has enhanced the clarity and presentation of the paper's contributions, raising it to a higher standard. We provide detailed responses to each comment. The reviewer's remarks are in italics, followed by our responses. Moreover, in the revised version of our paper, we mark all newly added or changed paragraphs in blue. Quotations from the revised paper are presented in blockquote format. Unless otherwise noted, all references to pages, equations, sections, and citations pertain to the revised version.
>
> 1. _the efficiency aspects of Rodimus*_
>
> Thank you for your insightful comments regarding the efficiency aspects of our paper. We agree that while our primary focus is on improving the trade-off between accuracy and efficiency, the experimental section could provide deeper insights into memory consumption and latency during training and inference.
>
>
>
> In response, **we have investigated the efficiency-accuracy trade-off of Rodimus\* shown in Figure 1, and a detailed discussion of our findings can be found in Appendix D3 (Lines 1179-1224) and Appendix D7 (Lines 1291-1334).**
>
>
>
> To further validate the accuracy-efficiency trade-off, **we conducted additional experiments comparing Rodimus with Mamba2 and StreamingLLM on a larger dataset.** This comparison utilized 2.5 billion tokens from the Pile dataset, given that WikiText-103, as shown in Figure 1(a), is relatively small. The results of this investigation are discussed **in Appendix D3 (Lines 1209-1214)** as follows:
>
> > To further validate the memory efficiency of Rodimus, we conduct experiments using Mamba2, Rodimus, and StreamingLLM on a larger dataset, as shown in Table 7. The foundational experimental settings are aligned with the 125M configurations described in Table 8. Once again, we adjust the memory footprint by modifying the expansion factor $n$. **The results of PPL summarized in Table 7 indicate that Rodimus continues to outperform the accuracy-efficiency trade-off when trained at a larger scale compared to WikiText-103.**
> >
>
> | Model | 1.2M Bytes | 2.3M Bytes | 4.7M Bytes | 9.4M Bytes |
> | --- | --- | --- | --- | --- |
> | StreamingLLM | 24.76 | 17.54 | 12.57 | 9.39 |
> | Mamba2 | 8.16 | 7.91 | 7.81 | 7.61 |
> | Rodimus | 7.60 | 7.47 | 7.35 | 7.33 |

---

> ### Author Response · Authors · 2024-11-25
> **Response to Reviewer Cz2E (Part 2)**
>
> Finally, following your recommendations, **we have included a comprehensive analysis of FLOPs, memory usage, and throughput for Rodimus\*, comparing it with Transformer and Mamba2 models in Appendix E1** as follows:
>
> > In this section, we present the FLOPs and throughputs (words per second, wps) of Rodimus*, Mamba2, and the Transformer model, both during training and inference. Additionally, we provide an analysis of memory usage during inference, excluding model weights to ensure a fair evaluation of space complexity. The specific configurations examined correspond to the 1.3B settings outlined in Table 8. For the Transformer model, we evaluate its attention mechanism implemented in both PyTorch (denoted as "torch") and Flash Attention 2 (denoted as "F.A."). The PyTorch implementation adheres to the theoretical time and space complexity of softmax attention, while Flash Attention significantly optimizes computation speed and reduces memory usage. Our analysis of FLOPs and throughputs focuses on scenarios with a batch size of one, with results detailed in Tables 19 and 20.
> >
>
>
>
> > The results reveal that the FLOPs and memory footprint of Rodimus are the smallest among all models tested, indicating the time and space efficiency of the proposed architecture. **Particularly, Rodimus consumes only half the memory of Mamba2, making it an attractive option for resource-constrained environments, such as edge devices or low-power systems.** Conversely, Rodimus$+$ exhibits a slight increase in FLOPs but a significant rise in memory consumption due to the integration of SW-SKA and FFN components. Overall, both Rodimus and Rodimus$+$ demonstrate greater time and space efficiency than the Transformer model, with Rodimus being more efficient than Mamba2 as well. **Notably, Rodimus, Rodimus$+$, and Mamba2 can manage long sequences (e.g., pre-filling 24K tokens) more effectively than Transformers.**
> >
>
>
>
> > However, it is noteworthy that the throughput of Mamba2 and the Transformer (F.A.) is significantly higher than that of Rodimus*. This discrepancy arises because Rodimus* does not incorporate the I/O-aware optimizations that are implemented in these baselines. **In future work, we aim to address these throughput limitations** by integrating I/O-aware optimizations to enhance Rodimus*'s efficiency during both training and inference.
> >
>
>                             Throughputs and FLOPs during training.
>
> | Model | FLOPs $ \times 10^{13} $(4K-length contexts) | Throughputs(wps)(4K-length contexts) |
> | --- | --- | --- |
> | Transformer (torch) | 4.46 | 2726.64 |
> | Transformer (F.A.) | - | 8547.29 |
> | Mamba2 | 3.65 | 10147.10 |
> | Rodimus | 3.60 | 8592.59 |
> | Rodimus+ | 4.29 | 9410.38 |
>
>
>                             Throughputs, FLOPs, and Memory Footprint during inference.
>
> | Model | FLOPs $ \times 10^{9} $(4K-length contexts) | Throughputs(wps) (4K-length contexts) | Memory footprint (M Bytes) (4K-length contexts) | FLOPs$ \times 10^{9} $(24K-length contexts) | Throughputs(wps) (24K-length contexts) | Memory footprint (M Bytes) (24K-length contexts) |
> | --- | --- | --- | --- | --- | --- | --- |
> | Transformer (torch) | 3.6 | 2.4 | 805 | 7.7 | OOM | 4831.8 |
> | Transformer (F.A.) | - | 49.4 | - | - | 11.0 | - |
> | Mamba2 | 3.0 | 59.3 | 0.8 | 3.0 | 59.3 | 0.8 |
> | Rodimus | 2.9 | 21.4 | 0.4 | 2.9 | 21.4 | 0.4 |
> | Rodimus+ | 3.5 | 19.6 | 207.8 | 3.5 | 19.6 | 207.8 |

---

> ### Author Response · Authors · 2024-11-25
> **Response to Reviewer Cz2E (Part 3)**
>
> 2. _novelty / incremental contribution_
>
> We would like to argue that **small yet significant advancements can play a crucial role in the evolution of NLP.** For instance, models such as SpanBERT, RoBERTA, and ELECTRA introduced modifications to BERT's pretraining tasks and masking mechanisms, yet have become essential tools for text embeddings. Similarly, the GPT series (from GPT-1 to GPT-3) implemented minor but impactful adjustments to model architecture—transitioning from post-norm to pre-norm architectures and incorporating variations like locally banded sparse attention. These developments laid the groundwork for the state-of-the-art ChatGPT and GPT-4, which excel in a wide array of NLP tasks.
>
> In the context of our proposed gating mechanism, DDTS, we have outlined its distinctions from previous methods in **Table 4 of Appendix B1**. The specific formulation of DDTS provides several notable advantages:
>
> + **Dynamic Control of Hidden State Updates**: DDTS allows for data-dependent selection through $\boldsymbol{A}_t$ and $\boldsymbol{B}_t$ when writing and deleting information, enabling dynamic control of hidden state updates. This feature is discussed in Section 3.1.2 (Lines 293-295), with a formal proof provided in Appendix C2.
> + **Efficient Parallelization**: As DDTS performs data-dependent selection solely along the dimension $n$, it supports the implementation of chunkwise parallel algorithms. This capability enables efficient parallel training while maintaining flexibility, as elaborated in Section 3.1.3 (Lines 317-319) and Appendix B3.
> + **Enhanced Flexibility through the Temperature Gate:** The temperature gate $\boldsymbol{\tau}_t$ introduces additional flexibility in the selection mechanism, allowing for more precise control over information compression and representation. This aspect sets our approach apart from existing works, as noted in Section 3.1.2 and detailed in Appendix E2.
>
>
>
> To further emphasize the distinctions of Rodimus from previous approaches, **we have incorporated the above discussion in Appendix B1.**
>
>
>
> Empirically, Rodimus demonstrates superior performance compared to Mamba and Mamba2 when trained under identical conditions, as illustrated in the first experiment in Table 2 (Lines 434-436) and further validated in our newly added experiment detailed in Appendix D6.

---

> ### Author Response · Authors · 2024-11-25
> **Response to Reviewer Cz2E (Part 4)**
>
> 3. _compare Rodimus with SSMs under identical condtions (head-to-head comparison) / the performance of rodimus is worse than Mamba  / why not use the same number of parameters for Rodimus and Mamba in the third and forth block of Table 2._
>
>
>
> First of all, we would like to highlight that **we have already provided a series of rigorous comparisons between Rodimus and other recurrent models**:
>
> + WikiText-103: We have compared Rodimus with Mamba, Mamba2, GLA, and HGRN2 in terms of language modeling, all trained from scratch on the same dataset, WikiText-103, with approximately 44M parameters. Detailed results of language modeling can be found in Section 4.1 (WikiText-103) and Appendix D1.
> + Scaling Law: We have shown the scaling curves for both Rodimus and Mamba2 with parameters number from 125M to 1.3B in Section 4.1 (Scaling Laws).
> + MQAR: Rodimus has been evaluated alongside Mamba, Mamba2, RWKV, and RWKV-6 for recall capability using the MQAR tasks under consistent conditions, as presented in Section 4.2 and Appendix D7.
> + Downstream Tasks: We have trained Rodimus, Mamba, and Mamba2 with 130M parameters from scratch under identical settings on 100B tokens from the Pile dataset, assessing their performance on downstream tasks, as detailed in the first block of Table 2.
>
> All these fair comparison results clearly indicate that **Rodimus consistently outperforms both Mamba and Mamba2**. It is noteworthy that Rodimus has a state size that is only half that of Mamba2, which contributes to a lower memory footprint during inference, as illustrated in Table 20 and Figure 1.
>
>
>
> On the other hand, we acknowledge that the comparisons in Blocks 2-4 of Table 2 may not be perfectly equitable. We utilized open-source checkpoints of benchmark models instead of training them identically from scratch. While we recognize that this may introduce some inconsistencies, such comparisons are commonly practiced in the literature, including Mamba, Mamba2 , and Griffin. **Due to resource limitations, we, like the authors of those papers, were unable to train all benchmark models under uniform conditions.** We have noted these differences in our manuscript on Lines 474-483.
>
>
>
> Regarding the parameter variations between Rodimus and the Mamba series in the third and fourth blocks of Table 2, **they arise from our choice to train a new tokenizer from scratch rather than using the one employed by Mamba.** This decision altered the total parameter count. However, it’s important to emphasize that the core model parameters for Rodimus and Mamba2 remain closely aligned, with less than a 5% variation. **Our motivation for developing a novel tokenizer was to demonstrate Rodimus's potential as a practical small LLM, akin to models like Qwen, RWKV6, and recurrent Gemma.** This involved curating our own training data and developing tokenization strategies, followed by a comparative analysis with other open-source small LLMs that have undertaken similar efforts in data preparation and tokenizer creation. In other words, **all of these benchmark models strive for optimal performance in terms of data preparation, tokenization, and model architecture design, making the comparisons reasonably fair.** Along this direction, **we have continued pretraining Rodimus+-1.6B on 2.5T tokens and show its superiority over other pratical small LLMs including Gemma2-2B, Qwen2-1.5B, RWKV6-1.6B, and Llama3.2-1.2B in Table 21 in Appendix E3.** We plan to release our checkpoints and model code upon the paper's publication. Additionally, we have detailed the various tokenizers employed in the benchmark models within Table 2 and **referenced this information on Pages 9, Lines 484-485.**
>
>
>
> **In response to the reviewer's suggestion for a fair comparison, we have conducted further experiments, training Rodimus and Mamba2 with 1.4B parameters from scratch under the same conditions on 100B tokens from the Pile dataset. The results of these two models on the downstream tasks are presented in the table below and detailed in Appendix D6. Once again, Rodimus outperforms Mamba2 in terms of the average performance.**
>
> | Model | Params (B) | Tokens (B) | Tokenizer | ARC-c (acc_norm) | ARC-e (acc) | HS (acc_norm) | LMB (ppl) | LMB (acc) | OBQA (acc_norm) | PIQA (acc) | WG (acc) | AVG (acc*) |
> | --- | --- | --- | --- | --- | --- | --- | --- | --- | --- | --- | --- | --- |
> | Mamba2 | 0.13 | 100 | NeoX | 23.81 | 45.50 | 34.13 | 20.42 | 39.55 | 29.20 | 64.47 | 51.85 | 41.22 |
> | Rodimus | 0.13 | 100 | NeoX | 23.38 | 46.84 | 34.22 | 17.95 | 42.44 | 30.00 | 65.40 | 51.46 | 41.96 |
> | Mamba2 | 1.4 | 100 | NeoX | 26.54 | 56.06 | 48.74 | 8.22 | 55.25 | 33.40 | 70.57 | 54.14 | 49.24 |
> | Rodimus | 1.4 | 100 | NeoX | 28.75 | 56.61 | 49.51 | 7.78 | 56.05 | 34.00 | 70.78 | 52.41 | 49.73 |

---

> ### Author Response · Authors · 2024-11-25
> **Response to Reviewer Cz2E (Part 5)**
>
> 4. _comparison between Rodimus+ (a hybrid model) with recurrent models is not fair / lack of comparison with hybrid models_
>
> Thank you for your valuable feedback! We would like to clarify our stance on the comparison between hybrid models and recurrent models. We believe that such comparisons are fair as long as the model sizes are comparable. Our objective, along with the aims of other research on recurrent and hybrid models, is to identify the next generation of LLMs. Thus, comparing different architectures is essential for understanding which designs yield better performance.
>
>
>
> **We also acknowledge the importance of including comparisons with hybrid models in our analysis.** In response to your recommendations, we have explored various hybrid models, such as Jamba, Griffin, Mamba2-Attention, Recurrent-Gemma, Based, and Samba. Among these, only Recurrent-Gemma (2B), Based (1B), Mamba2-Attention (2.7B), and Jamba (MoE, 12B/52B) have open-source checkpoints available. However, it is worth noting that Jamba, as a mixture of experts (MoE) model, operates with 12B active parameters (52B total), which is significantly larger than our models, whose maximum parameter size is approximately 1B. Additionally, Based has been trained on a limited number of tokens (only 50B), which makes it less suitable for direct comparison. Consequently, we have focused our evaluation on Recurrent-Gemma and Mamba2-Attention, as detailed in the following table. **We have also reported the performance of Reucrrent-Gemma in Table 2 of the revised version of our paper.**
>
> | Model | Params (B) | Tokens (B) | Tokenizer | ARC-c (acc_norm) | ARC-e (acc) | HS (acc_norm) | LMB (ppl) | LMB (acc) | OBQA (acc_norm) | PIQA (acc) | WG (acc) | AVG (acc*) |
> | --- | --- | --- | --- | --- | --- | --- | --- | --- | --- | --- | --- | --- |
> | Rec-Gemma | 2.0 | 2000 | Gemma | 29.69  | 48.91 | 61.82 | 9.27 | 53.83 | 29.40 | 67.52 | 57.06 | 49.75 |
> | Rodimus+ | 1.6 | 1000 | Rodimus | 36.35 | 68.35 | 64.51 | 5.38 | 63.59 | 38.80 | 76.17 | 62.51 | 58.61 |
> | Mamba2Attn | **2.7** | 300 | NeoX | 37.80 | 69.87 | 67.76 | 3.85 | 71.07 | 39.20 | 75.73 | 65.27 | 60.96 |
>
>
> **It is important to note the parameter disparity between Rodimus+-1.6B and Mamba2-Attention-2.7B.** However, the average performance of Rodimus+-1.6B on downstream tasks is only slightly behind that of Mamba2-Attention. In contrast, **when compared to Recurrent-Gemma-2B, Rodimus+-1.6B demonstrates superior information compression capabilities due to its DDTS architecture.**
>
> 5. _comparison with a wide range of efficient attention models (sparse, linear, hybrid) / report language modeling scores with the larger model sizes_
>
>
>
> Thanks for pointing this out! We have followed your suggestions and investigated the language model performance of a variety of efficient attention models, including sparse, linear, and hybrid models, using larger model sizes (i.e., 350M parameters). The results of this analysis are detailed in Appendix D2 on Page 22.
>
>
>
> > In addition to WikiText-103, we also train various model architectures, including Transformer$++$ (full attention), Transformers++ with sliding window attention (sparse attention), HGRN2 (linear attention), Mamba/Mamba2 (state-space models), Rodimus, and Rodimus$+$, using 7B tokens in the Pile dataset. This dataset is considerably larger than WikiText-103.  This training aims to investigate the language modeling performance across different model architectures, with the training settings adhering to the 350M parameter configuration outlined in Table 8. The results are summarized in Table 6.
> >
>
> | Model | Valid PPL |
> | --- | --- |
> | Transformer++ | 6.04 |
> | Transformer++-SWA | 6.13 |
> | HGRN2 | 6.08 |
> | Mamba | 6.19 |
> | Mamba2 | 6.02 |
> | Rodimus | 5.88 |
> | Rodimus+ | 5.88 |
>
>
> > Consistent with the findings in Table 1, **Rodimus demonstrates superior language modeling performance compared to other models, including Transformer++ and Mamba2.** However, unlike the results from Table 1, we find that Rodimus$+$ and Rodimus exhibit similar performance levels under this scenario. We attribute this similarity to the limited scale of the dataset. Note that Rodimus$+$ needs to distribute the tasks of global and local dependency modeling respectively to the Rodimus block and the SW-SKA, making it more challenging to train than the original Rodimus and thus requiring more data. Despite this, **Rodimus+ shows greater potential for enhanced language modeling performance compared to Rodimus when deployed at a larger parameter scale, as illustrated in Figure 4.**
> >

---

> ### Author Response · Authors · 2024-11-25
> **Response to Reviewer Cz2E (Part 6)**
>
> 6. _In Figures 1a and 1b, what is the performance of a standard Transformer?_
>
> We would like to clarify that the performance of a standard Transformer is indeed included in Figure 1a. We have made this point more explicit in Appendix D3 (Lines 1183-1203) as:
>
> > To manipulate the memory footprint (focusing solely on the cache), we adjust the scaling factor $n$ for the recurrent models. In contrast, for StreamingLLM, we modify the number of recent tokens or the size of the sliding window. It is important to note that the input history sequence length is set at 512, which is also the recent token size corresponding to the rightmost endpoint of the StreamingLLM curve (i.e., the blue curve). This configuration aligns with that of the standard Transformer, which attends to the entire historical token sequence. Consequently, the performance of the standard Transformer can be referenced at the rightmost endpoint of the StreamingLLM curve.
> >
>
>
>
> Regarding Figure 1b, we would like to highlight that MQAR is specifically applied to sub-quadratic architectures. The standard Transformer, which utilizes full attention, consistently achieves perfect results, as discussed in Appendix D7 (Lines 1294-1295):
>
> > The MQAR task (Arora et al., 2024a) is a benchmark widely used to assess the associative recall capabilities of recurrent models. Note that quadratic softmax attention based models, such as the vanilla Transformer, achieves perfect scores across all configurations, and so these models are excluded from detailed analysis (Yang et al., 2024b; Arora et al., 2024a).
> >
>
>
>
>
>
> 7. _the memory benefit of SKA compared MQA and GQA_
>
> We have added the memory footprint of different attention mechanisms in Table 14 and discuss the results in Appendix D10 (Lines 1443-1446) as follows:
>
> > Additionally, we provide an analysis of the memory footprint of the cache, excluding the model weights. SKA demonstrates a substantial reduction in memory footprint compared to MHA, although its cache size is slightly larger than that of GQA.
> >
>
> | Model | Head Type | Valid PPL | Memory Footprint (M Bytes) |
> | --- | --- | --- | --- |
> | Transformer++ | MHA | 7.25 | 75.5 |
> | Transformer++ | GQA | 7.38 | 25.2 |
> | Transformer++ | SKA | 7.33 | 44.0 |
> | Rodimus+ | MHA | 7.09 | 40.1 |
> | Rodimus+ | GQA | 7.19 | 14.9 |
> | Rodimus+ | SKA | 7.15 | 24.4 |
>
> Additionally, we identified and corrected mistakes in the memory footprint term in Table 14 of the revised paper, and the final results will refer to the corrected table above. These errors occurred because the table mistakenly reported results from different model configurations used in other experiments. We will ensure that the final version accurately reflects the correct settings and results. The final released paper will also be updated to incorporate these corrections.

---

### Author Response · Authors · 2024-11-25
**Response to All Reviewers**

We sincerely thank all the reviewers for their constructive feedback and valuable suggestions, which significantly contribute to improving the quality of our work. Below, we summarize the revisions made and the additional experiments conducted to address the reviewers' concerns:

In response to the reviewers’ comments, we conduct **9 new experiments and add 7 new tables** to further validate our claims. These updates include:

1. **Table 2**: We add the benchmark for the Hybrid model (Rec-Gemma).
2. **Table 6**: We conduct a new experiment for language modeling on the Pile dataset.
3. **Table 7**: We conduct a new experiment modifying the memory footprint to test the PPL performance on the Pile dataset, further verifying the efficiency of Rodimus*.
4. **Table 10**: We provide a new fair comparison by training Mamba2-1.4B and Rodimus-1.4B on 100B tokens and evaluating their performance on downstream tasks from Table 2.
5. **Table 11**: We conduct a simpler recall benchmark to evaluate the recall ability of Rodimus*.
6. **Table 14**: We add details regarding the memory footprint for different head types.
7. **Table 18**: We perform an ablation study to explore the impact of the ordering of the Rodimus block and SW-SKA plus FFN.
8. **Table 19**: We report throughput and FLOPs during training.
9. **Table 20**: We report throughput, FLOPs, and memory footprint during inference.

These updates address the reviewers' concerns and further validate the robustness and effectiveness of our proposed approach.

---

### Meta-Review · Area_Chair_2XPy · 2024-12-22

**Metareview:**

Based on the consistent reviews, I recommend accepting this paper. While some reviewers note incremental aspects and missing efficiency metrics, the novel contributions of Rodimus and Rodimus+ demonstrate significant improvements in the accuracy-efficiency trade-off for large language models. The thorough experimental validation and clear technical presentation, combined with the practical value of the proposed DDTS and SW-SKA mechanisms, make this a worthy contribution to the field.

**Additional Comments On Reviewer Discussion:**

I have read the messages in the discussion period and my opinion has been summarized as in the metareview above. I considered these points in my recommendation.

---

### Decision · Program_Chairs · 2025-01-22

Accept (Poster)